# Restoring bone marrow niche function rejuvenates aged hematopoietic stem cells by reactivating the DNA Damage Response

Pradeep Ramalingam[1,4], Michael C. Gutkin[2,4], Michael G. Poulos[1,4], Taylor Tillery[2], Chelsea Doughty[2], Agatha Winiarski[1], Ana G. Freire[2], Shahin Rafii [3], David Redmond [3] & Jason M. Butler [1,2,3] ✉

Aging associated defects within stem cell-supportive niches contribute towards age-related decline in stem cell activity. However, mechanisms underlying age-related niche defects, and whether restoring niche function can improve stem cell fitness, remain unclear. Here, we sought to determine whether aged blood stem cell function can be restored by rejuvenating their supportive niches within the bone marrow (BM). We identify Netrin-1 as a critical regulator of BM niche cell aging. Niche-specific deletion of Netrin-1 induces premature aging phenotypes within the BM microenvironment, while supplementation of aged mice with Netrin-1 rejuvenates aged niche cells and restores competitive fitness of aged blood stem cells to youthful levels. We show that Netrin-1 plays an essential role in maintaining active DNA damage responses (DDR), and that aging-associated decline in niche-derived Netrin-1 results in DNA damage accumulation within the BM microenvironment. We show that Netrin-1 supplementation is sufficient to resolve DNA damage and restore regenerative potential of the aged BM niche and blood stem cells to endure serial chemotherapy regimens.

Aging is associated with an increased risk for hematologic malignancies that frequently require hematopoietic stem cell transplantation (HSCT) therapies. Reduced-intensity conditioning (RIC) regimens for achieving myelosuppression essential for HSCTs have greatly expanded the eligibility for HSCT in older patients and it is now estimated that ~39% of allogenic and ~55% of autologous HSCTs are performed in adults >60 years of age[1,2]. While RIC regimens decrease transplant-related mortality (TRM), they are associated with an increased risk of relapse[2]. Contrarily, higher-intensity conditioning regimens that improve disease-free survival are associated with increased TRM in older patients. One of the primary causes that predisposes older patients to a higher risk of negative HSCT outcomes/failures is that aging diminishes the regenerative ability of the

hematopoietic system to recover from cytotoxic side effects of myelosuppression, arising in part from an impairment in the fitness and functionality of the HSC-supportive bone marrow (BM) niche[3–6]. While the critical role played by niche cells in maintaining HSC fitness has been well-established[7], there is little insight into the mechanisms underlying age-related deterioration in the HSC-supportive niche activity[4,8].

To identify candidate factors that can improve niche function and HSC fitness during aging, we surveyed the transcriptomic data of a recently described murine model that manifests premature aging of the hematopoietic system[9]. Our analysis revealed a putative role for Netrin-1 (NTN1) in regulating aging of the BM niche. NTN1 is an established axon-guidance cue that has recently been shown to

[1]Department of Medicine, University of Florida Health Cancer Center, Gainesville, FL, USA. [2]Center for Discovery and Innovation, Hackensack University Medical Center, Nutley, NJ 07110, USA. [3]Ansary Stem Cell Institute, Division of Regenerative Medicine, Department of Medicine, Weill Cornell Medicine, New York, NY 10065, USA. [4]These authors contributed equally: Pradeep Ramalingam, Michael C. Gutkin, Michael G. Poulos. ✉e-mail: jason.butler@medicine.ufl.edu

regulate diverse processes including angiogenesis and osteogenesis, suggesting that NTN1 could potentially regulate BM niche acitivity[10–12]. In the hematopoietic system, NTN1 was recently identified as a ligand for Neogenin-1 (NEO1) on HSCs and it has been proposed that BM niche-derived NTN1 engages NEO1 on HSCs to preserve HSC dormancy in young mice[13,14]. However, whether NTN1 regulates BM niche function is unknown. Here we identify NTN1 as a critical regulator of niche cell fitness during homeostasis, regeneration, and aging. We define BM mesenchymal stromal cells (MSCs) and BM endothelial cells (BMECs) as critical sources of niche-derived NTN1 within the BM and demonstrate that NTN1 regulates both niche-HSC and niche-niche interactions. Conditional deletion of NTN1, either in Leptin Receptor+ (LepR+) MSCs or in ECs of young mice, induces functional defects in their BM niche and hematopoietic system. We show that niche cells and HSCs require NTN1 for maintaining an active DNA damage response (DDR), preventing DNA damage accrual, and preserving functional potential during aging. We identify that a dampened DDR is a fundamentally conserved attribute of EC, MSC, and HSC aging and demonstrate that NTN1 supplementation reactivates the dampened DDR and resolves the accrued DNA damage within niche cells and HSCs. We show that treatment of aged mice with recombinant NTN1 is sufficient to reverse phenotypic defects of the aged BM vascular niche, including improved BM vascular integrity and suppression of age-associated BM adiposity. Additionally, we demonstrate that NTN1 mediated niche rejuvenation is associated with a restoration in the self-renewal capacity of aged HSCs to youthful levels. The beneficial effects of NTN1 on the aged hematopoietic system translate to an accelerated hematopoietic recovery, increased survival, and preservation of body weight during serial myelosuppressive chemotherapy. In summary, our findings indicate that NTN1 supplementation can serve as a therapeutic strategy to enhance integrity of the aged BM vascular niche, restore the functional potential of aged HSCs, and improve survival following myelosuppressive regimens in older patients.

## Results

### Loss of NTN1 induces functional defects in the BM niche

To identify cellular sources of NTN1 within the BM niche, we analyzed published RNA-Seq datasets on BM niche cell subpopulations[15,16] which revealed that NTN1 was primarily expressed in BM MSCs and BMECs (Fig. 1a, b). We confirmed NTN1 expression in BM MSCs via immunofluorescence analysis of femoral sections in LepR-Cre; Rosa26-Tdtomato mice wherein TdTomato expression marks MSCs within the BM[17,18] (Fig. 1c). RT-PCR analysis of FACS sorted BM-derived LepR+ cells and VEcadherin+ BMECs (Fig. 1d, f), and ex vivo cultures of BM MSCs and BMECs (Fig. 1e, g), confirmed NTN1 expression in these BM niche cell populations. Collectively, these findings suggested that BM MSCs and BMECs likely represent the principal sources of NTN1 within the BM niche. To determine whether NTN1 derived from BM MSCs and BMECs regulates BM niche function, we generated murine models with conditional deletion of NTN1 specifically within MSCs or ECs. Given that NTN1 was expressed in a diverse array of BM MSC subsets including PDGFRa+ cells, CAR cells, LepR+ cells and osteoblasts (Fig. 1a, b), we utilized LepR-Cre mice that have been shown to mark these BM MSC subsets and utilized to identify niche derived regulatory factors within the BM[17–19]. To delete NTN1 in MSCs, Netrin^fl/fl mice[20] were crossed with LepR-Cre mice[19] to generate *LepR-NTN1* mice. To assess the role of EC-derived NTN1 on niche function, we crossed Netrin^fl/fl mice with an EC-specific *cre* line (*Cdh5(PAC)-creERT2*)[21] to

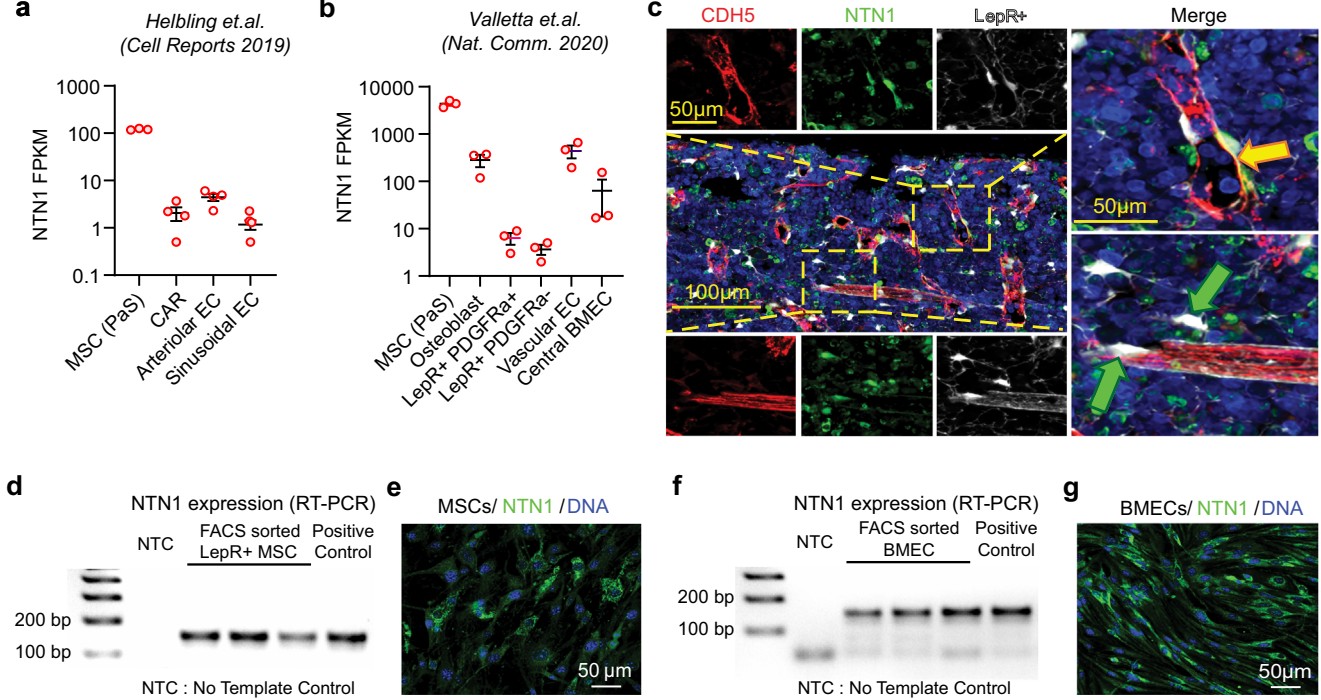

**Fig. 1 | NTN1 is expressed by LepR + MSCs and BMECs within the BM niche.** **a**, **b** RNA-Seq analyses of the BM niche demonstrating NTN1 expression within MSCs and BMECs (Helbling et al[15], Valletta et al[16]). PaS: PDGFRa+Sca1+, CAR: CXCL12 Abundant Reticular cells. Error bars denote sample mean ± standard error of the mean (SEM). **c** Representative immunofluorescence (IF) images of femoral sections showing NTN1 expression in LepR+ MSCs tightly associated with BMECs (Yellow arrows) and perivascular LepR+ MSCs (Green arrows). Blue: Lineage⁺ (TER119/CD11B/GR1/B220/CD3) hematopoietic cells. Red: BMECs intravitally labeled with anti-VE-CADHERIN antibody. White: LepR+ cells marked by LepR-Cre+/tdTomato+.

Green: NTN1 expression detected by an anti-NTN1 antibody. (*N* = 3 mice). **d** RT-PCR analysis demonstrating NTN1 expression within FACS purified LepR+ MSCs (defined as CD45⁻TER119⁻VECAD⁻CD31⁻LepR⁺) within the BM (*N* = 3 mice/group). **e** Representative IF images of cultured BM MSCs showing NTN1 expression. (*N* = 1 cell line/group). **f** RT-PCR analysis demonstrating NTN1 expression within FACS-purified BMECs (defined as CD45⁻TER119⁻LepR⁻VECAD⁺CD31⁺) within the BM. (*N* = 3 mice/group). **g** Representative IF images of cultured BMECs showing NTN1 expression. (*N* = 1 cell line/group).

generate *CDH5-NTN1* mice. *CDH5-NTN1* mice and their littermate controls were treated with Tamoxifen to induce EC-specific NTN1 deletion. Immunofluorescence analysis of femoral sections revealed that *CDH5-NTN1* mice displayed no gross morphological alterations in their vasculature (Fig. 2a); however, *LepR-NTN1* mice manifested distinct changes in vascular morphology including vessel dilation and discontinuity suggestive of impaired vascular integrity (Fig. 2a). Analysis of vascular permeability using the modified Evans Blue Dye (EBD) extravasation assay, and vessel diameter using immunofluorescence, revealed that deletion of NTN1 in BMECs did not alter vascular integrity or vessel diameter (Fig. 2b, d), while NTN1 deletion in LepR+ cells increased BM vascular leakiness and vessel diameter (Fig. 2c, e), indicating that BMECs require NTN1 from perivascular LepR+ cells to maintain vascular integrity within the BM. Immunofluorescence analysis also revealed that *LepR-NTN1* mice and *CDH5-NTN1* mice displayed adipocyte accumulation within their BM (Fig. 2f, i). Accordingly, LepR+ MSCs isolated from *LepR-NTN1* mice and *CDH5-NTN1* mice demonstrated an increase in adipogenic differentiation potential (Fig. 2g, h, j, k). Notably, impaired BM vascular integrity and adipocyte accumulation represent hallmark features of an aged BM niche. Collectively, these findings indicate that deletion of niche-derived NTN1 induces phenotypes reminiscent of an aged BM niche[22].

## Loss of NTN1 disrupts HSC function

To determine whether phenotypic niche alterations observed upon deletion of niche-derived NTN1 impairs their ability to support HSC function, we analyzed hematopoietic parameters in both *LepR-NTN1* mice and *CDH5-NTN1* mice (Fig. 3). While young *LepR-NTN1* mice displayed normal peripheral blood lineage composition (Supplementary Fig. 1a), they demonstrated a decrease in BM cellularity and a decline in frequency of phenotypic HSCs (Lineage[Neg] cKIT[+]SCA1[+]CD150[+]CD48[Neg]) (Fig. 3a–c), as well as a decrease in multipotent progenitors (MPPs; Lineage[Neg] cKIT[+]SCA1[+]CD150 [Neg]CD48[Neg]) (Supplementary Fig. 1b) when compared to their littermate controls. Cell-cycle analysis revealed that HSCs from *LepR-NTN1* mice had an increase in percentage of cells in the G0 phase (Supplementary Fig. 1c). The decline in progenitor frequency in *LepR-NTN1* mice manifested as a functional loss of progenitor activity by methylcellulose-based colony forming unit (CFU) assays (Supplementary Fig. 1d). Furthermore, competitive HSC transplantation assays demonstrated that HSCs from *LepR-NTN1* mice displayed an impairment in their long-term engraftment without significant changes in lineage distribution within the engrafted cells (Fig. 3d, e & Supplementary Fig. 1e). Young *CDH5-NTN1* mice, unlike *LepR-NTN1* mice, did not demonstrate changes in their BM cellularity, phenotypic HSPC frequency or progenitor activity, and displayed no alterations in their peripheral blood lineage composition (Fig. 3g–i & Supplementary Fig. 1f, g, i). However, similar to *LepR-NTN1* mice, HSCs of *CDH5-NTN1* mice manifested a modest increase in their quiescence (%G0) (Supplementary Fig. 1h). Competitive HSC transplantation assays revealed a decline in long-term HSC engraftment and a myeloid-biased reconstitution at the expense of lymphoid output, which represent hallmark characteristics of aged HSCs (Fig. 3j, k & Supplementary Fig. 1j). RT-qPCR analysis confirmed a significant downregulation of NTN1 expression within BM MSCs of *LepR-NTN1* mice (Fig. 3f) and BMECs of *CDH5-NTN1* mice (Fig. 3l) confirming that loss of BM niche-derived NTN1 results in HSC and niche defects observed in these model systems (Figs. 2, 3). Collectively, these findings demonstrate that niche defects arising from niche-specific NTN1 deletion in young mice are sufficient to recapitulate HSC aging phenotypes including diminished engraftment potential, myeloid-skewed output, and an increase in HSC quiescence.

To determine whether niche-derived NTN1 is essential for preserving HSC fitness during aging, we physiologically aged *LepR-NTN1* mice and *CDH5-NTN1* mice for 16 months. Notably, aged *LepR-NTN1* mice, unlike aged *CDH5-NTN1* mice, manifested marked hair greying

typically observed following exposure to radiation[23] (Supplementary Fig. 2a, b). While HSC frequencies were not significantly altered, HSCs derived from both aged *LepR-NTN1* mice and *CDH5-NTN1* mice manifested a significant decrease in long-term HSC engraftment along with alterations in lineage composition (Supplementary Fig. 2c–j), as compared to their respective littermate controls. Taken together, these data confirm that NTN1 derived from the BM niche is essential to maintain HSC function.

## NTN1 regulates DDR within the BM niche

To elucidate mechanisms underlying niche defects observed upon NTN1 deletion, we performed transcriptomic analysis (RNA-Seq) on BMECs and BM LepR+ MSCs in *LepR-NTN1* and *CDH5-NTN1* mice (Figs. 4, 5 & Supplementary Data 1–3). Gene Set Enrichment Analysis (GSEA) of the LepR+ cell transcriptomes revealed an upregulation of the ADIPOGENESIS pathway in both *LepR-NTN1* and *CDH5-NTN1* mice (Fig. 4c, f), correlating with their BM adipocyte accumulation. GSEA also revealed a significant upregulation of pathways that regulate cell-cycle and DNA damage responses (DDR) including E2F_TARGETS, G2M_CHECKPOINT, MYC_TARGETS and DNA_REPAIR within LepR+ MSCs of both *LepR-NTN1* and *CDH5-NTN1* mice (Fig. 4a–f & Supplementary Data 2). Notably, E2F_TARGETS play crucial roles in regulation of cell-cycle and DNA Damage Responses (DDR)[24].While cell-cycle analysis of LepR+ cells did not reveal any significant differences when compared to controls, alkaline comet assays revealed a significant increase in DNA damage in LepR+ MSCs of both *LepR-NTN1* and *CDH5-NTN1* mice (Fig. 4g–j). Similar to LepR+ MSCs, BMECs also revealed an upregulation of DDR pathways and accumulation of DNA damage without significant cell-cycle alterations, following NTN1 deletion in either LepR+ MSCs or BMECs (Fig. 5a–j & Supplementary Data 3). Notably, BMECs demonstrated a downregulation of the ANGIOGEN-ESIS pathway in *LepR-NTN1* mice (Supplementary Data 3) but not in *CDH5-NTN1* mice, that correlates with the loss of vascular integrity observed in *LepR-NTN1* mice. Taken together, these data illustrate that loss of NTN1 in either LepR+ cells or BMECs disrupts BMEC-LepR+ niche-cell interactions within the BM and illuminates a role for niche-derived NTN1 in preventing DNA damage accumulation within the BM niche.

## NTN1 supplementation rejuvenates the aged BM niche

The underlying mechanisms that drive BM niche defects during physiological aging remain largely unknown. Given that DDR disruption within niche cells of young *LepR-NTN1* mice and *CDH5-NTN1* mice induces premature niche aging phenotypes (Figs. 2 & 3), we sought to determine whether aging is associated with DDR dysregulation within the BM niche. To this end, we performed transcriptomic analysis on BMECs and BM LepR+ MSCs derived from young (3 month old) and aged (18-month-old) mice (Fig. 6a–d & Supplementary Data 4–7). GSEA of both LepR+ MSC and BMEC transcriptomes revealed an over-representation of DDR pathways that were significantly downregulated during aging, raising the possibility that aging results in DNA damage accumulation within the BM niche (Fig. 6a–d & Supplementary Data 5, 7). To confirm whether aging is associated with DNA damage within the niche and to test whether NTN1 supplementation could reverse these defects, we treated aged (18-month-old) mice with recombinant murine NTN1 over a 2-week period and assessed the DNA damage within the BM niche (Fig. 6e–g). As indicated by the transcriptional analysis, comet assays confirmed that both LepR+ MSCs and BMECs derived from aged mice manifested significantly increased DNA damage, as compared to young niche cells (Fig. 6f, g). Importantly, NTN1 treatment was sufficient to ameliorate DNA damage within both LepR+ MSCs and BMECs of aged mice (Fig. 6f, g). Transcriptional analysis of niche cells derived from aged mice treated with NTN1 demonstrated that NTN1 upregulates DDR pathways within aged BMECs but not LepR+ MSCs (Fig. 6h–k & Supplementary Data 8, 9),

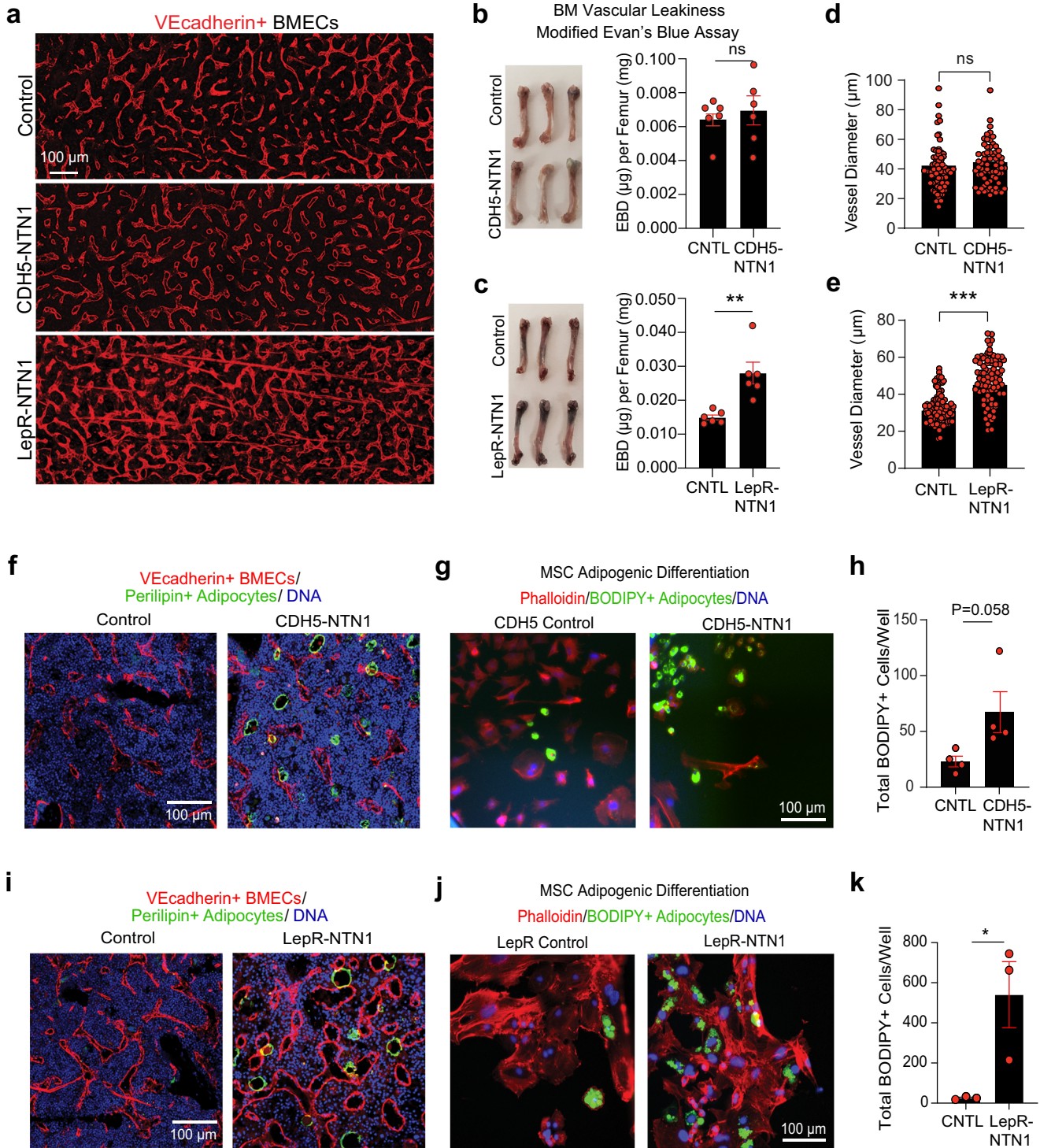

**Fig. 2 | Loss of NTN1 induces premature aging phenotypes within the BM niche.** **a** Representative immunofluorescence (IF) images of femurs intravitally-labeled with a vascular-specific CD144/VEcadherin antibody (red) demonstrating vascular disruption in *LepR-NTN1* mice (N = 3 mice/group). **b, c** Representative images of femurs isolated from mice injected with Evans Blue Dye (EBD) and quantification of vascular leakiness by EBD extravasation in **b** *CDH5-NTN1* mice, and **c** *LepR-NTN1* mice (n = 6 mice/group). **d, e** Quantification of vessel diameter within the BM of **d** *CDH5-NTN1* mice and **e** *LepR-NTN1* mice (n = 3 mice/group). **f, i** Representative IF images of femurs stained with α-Perilipin1 (PLIN1) antibody demonstrating an increase in PLIN1 + adipocytes in **f** *CDH5-NTN1* mice, and **i** *LepR-NTN1* mice, as compared to their littermate controls (n = 3 mice/group). **g, h** Representative IF images **g**, and quantification **h**, of LepR+ MSC adipogenic differentiation ex vivo in *CDH5-NTN1* mice (N = 3 mice/group). **j, k** Representative IF images **j**, and quantification **k**, of LepR+ MSC adipogenic differentiation ex vivo in *LepR-NTN1* mice (N = 3 mice/group). Statistical significance determined using two-tailed unpaired t-test. *P < 0.05; **P < 0.01; ***P < 0.001. Error bars represent sample mean ± standard error of the mean (SEM).

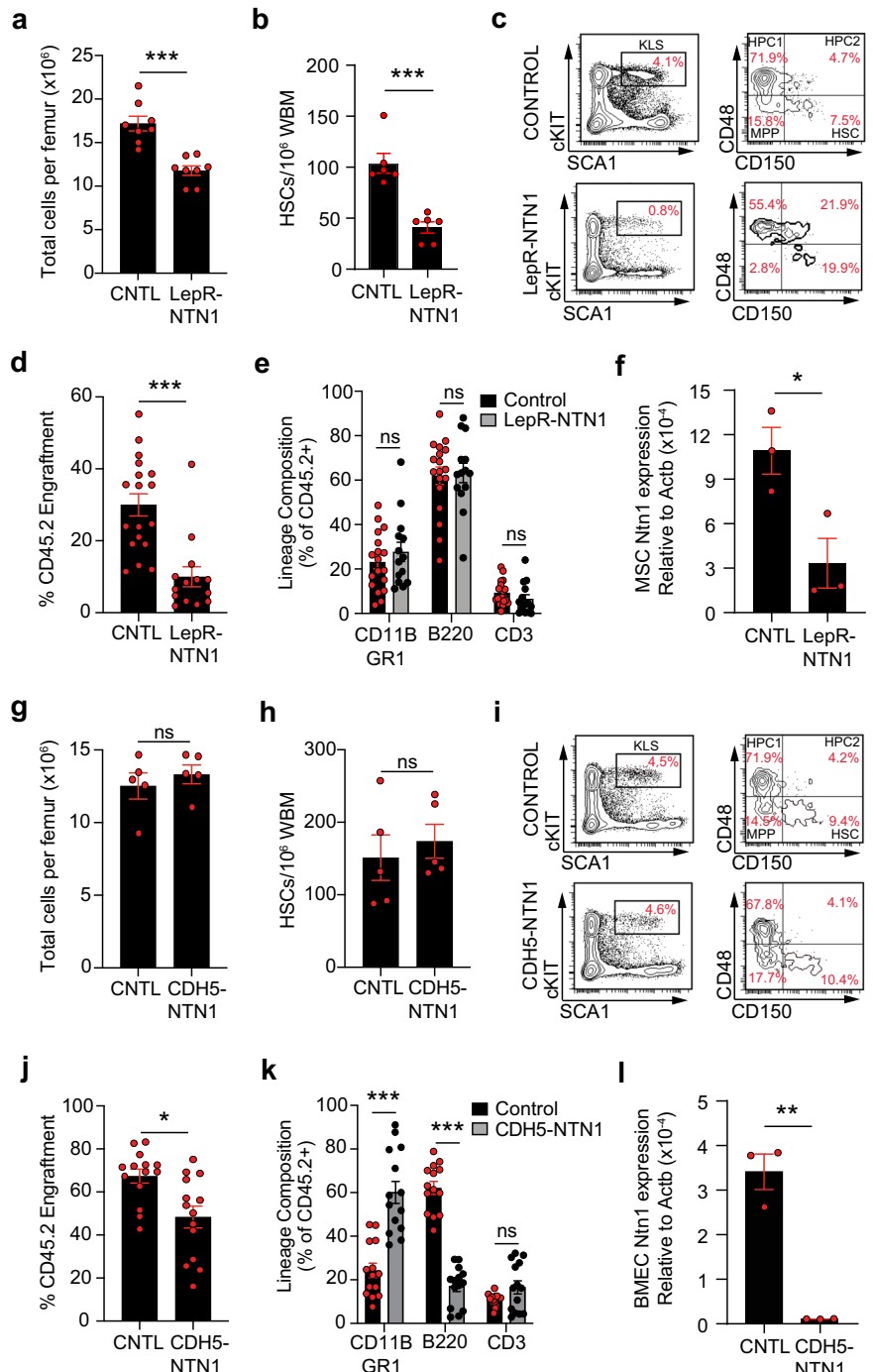

**Fig. 3 | BM niche derived NTN1 is essential for maintaining HSC homeostasis.**
**a–c** BM analysis in young (5-6 month old) *LepR-NTN1* mice demonstrating a decrease in BM cellularity **a** (N = 8 mice/group), and HSC frequency **b**, **c** (N = 6 mice/group), as compared to littermate controls. **d**, **e** Competitive HSC transplantations (250 CD45.2+ donor HSCs with $10^6$ CD45.1 WBM competitor/recipient) demonstrating a loss of long-term (>6 months post-transplant) HSC engraftment potential **d** without lineage alterations **e** in HSCs derived from *LepR-NTN1* mice (N = 6 donors/group; N = 18 Recipients for Control donors, N = 14 Recipients for *LepR-NTN1* donors). **f** RT-qPCR analysis of FACS purified LepR+ MSCs demonstrating decreased NTN1 expression in *LepR-NTN1* mice (N = 3 mice/group). **g–i** BM analysis of *CDH5-NTN1* mice demonstrating no gross changes in BM cellularity **g**, and HSC frequency **h**, **i**, as compared to littermate controls (N = 5/group). **j**, **k** Competitive HSC transplantations (250 CD45.2+ donor HSCs with $10^6$ CD45.1 WBM competitor/recipient) demonstrating a loss of long-term engraftment potential **j**, along with a myeloid-biased output **k**, in HSCs derived from *CDH5-NTN1* mice (N = 6 donors/group; N = 14 recipients/group). **l** RT-qPCR analysis of FACS purified BMECs demonstrating decreased NTN1 expression in *CDH5-NTN1* mice (N = 3 mice/group). Error bars represent sample mean ± standard error of the mean (SEM). Statistical significance determined using two-tailed unpaired t-test. *$P < 0.05$; **$P < 0.01$; ***$P < 0.001$. ns denotes statistically not significant.

indicating that DNA damage resolution within aged MSCs following NTN1 treatment likely results from an improvement in BM vascular function. Collectively, these data reaffirm the role of NTN1 in regulating DDR within the BM niche wherein deletion of NTN1 in young mice results in increased DNA damage within the BM niche (Figs. 4, 5),

whereas supplementation of aged mice with NTN1 resolves DNA damage within the aged BM niche (Fig. 6).

We next sought to determine whether NTN1 mediated DNA damage resolution is sufficient to improve the functionality of an aged BM niche. To evaluate BM vascular leakiness and oxygenation

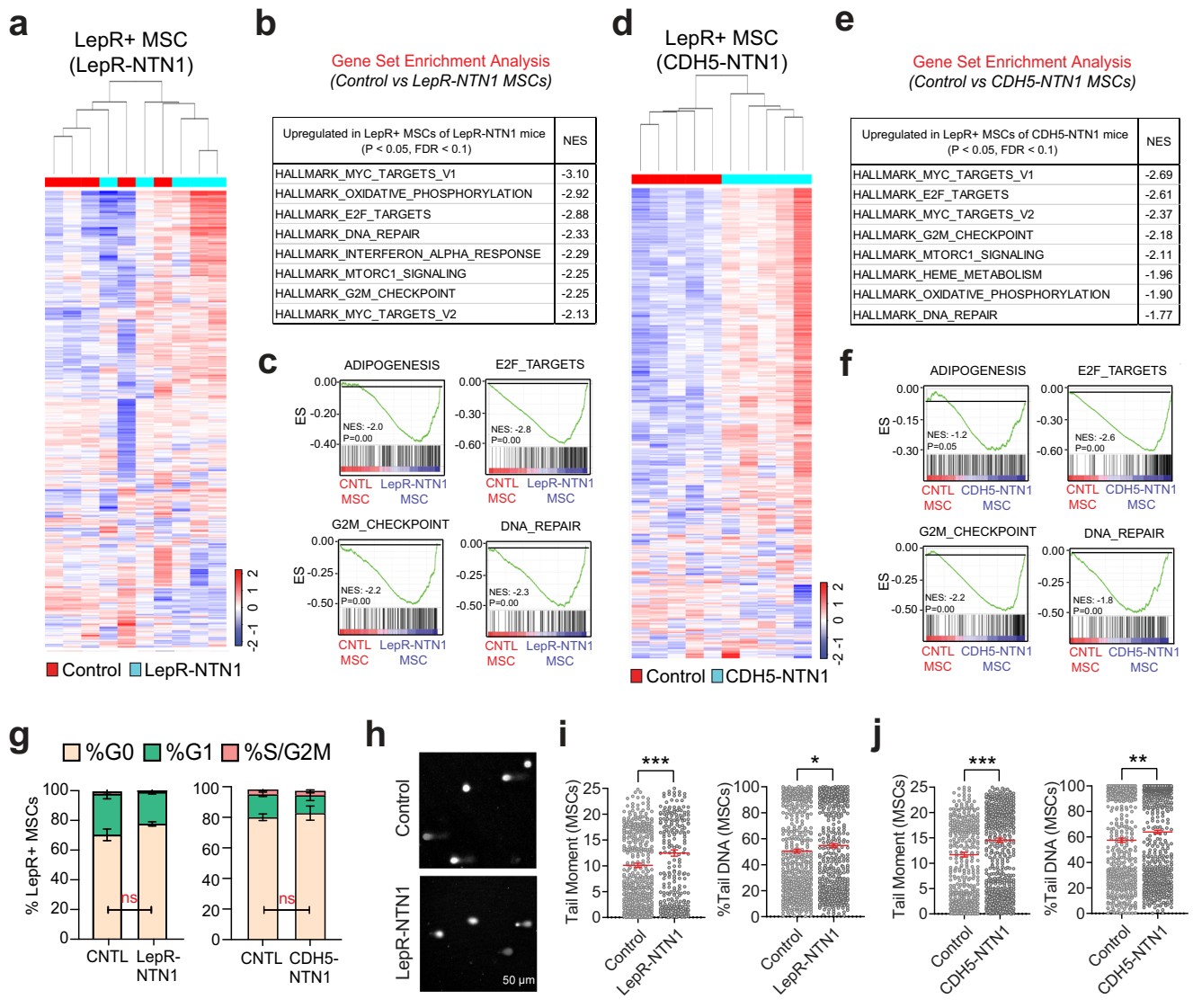

**Fig. 4 | Loss of NTN1 causes DNA damage within LepR + MSCs. a–c** RNA-Seq analysis of LepR+ MSCs derived from young (5-6 month-old) *LepR-NTN1* mice. **a** Heatmap depicting hierarchical clustering of differentially expressed genes. **b** GSEA demonstrating activation of DDR pathways in LepR+ cells of *LepR-NTN1* mice. **c** GSEA Enrichment plots demonstrating activation of ADIPOGENESIS and DNA REPAIR pathways in LepR+ cells of *LepR-NTN1* mice. **d–f** RNA-Seq analysis of LepR+ MSCs derived from *CDH5-NTN1* mice. **d** Heatmap depicting hierarchical clustering of differentially expressed genes. **e** GSEA demonstrating activation of DDR pathways in LepR+ cells of *CDH5-NTN1* mice. **f** GSEA Enrichment plots demonstrating activation of ADIPOGENESIS and DNA REPAIR pathways in LepR+ cells of

*CDH5-NTN1* mice. **g** Cell-cycle analysis of LepR+ MSCs derived from *LepR-NTN1* mice and *CDH5-NTN1* mice (N = 5 mice/group). **h** Representative IF images of single-cell gel electrophoresis (alkaline comet assays) performed on BM LepR+ MSCs. Alkaline comet analysis demonstrating an increase in average tail-moment and % Tail DNA in MSCs derived from both *LepR-NTN1* mice **i**, and *CDH5-NTN1* mice **j**, as compared to their littermate controls (N = 3 mice/group). Note that deletion of NTN1 within either MSCs or BMECs results in increased DNA damage within MSCs. Data is presented as the mean ± standard error of the mean (SEM). Statistical significance determined using two-tailed unpaired t-test. *P < 0.05; **P < 0.01; ***P < 0.001. ns denotes statistically not significant.

status simultaneously, aged mice were injected with 10 kDa Dextran and Hypoxyprobe following vehicle (PBS) or NTN1 treatment. We observed that treatment of aged mice with NTN1 resulted in significant reduction in their BM vascular leakiness assessed by immunofluorescence analysis of dextran extravasation and vessel diameter, a marked improvement in BM perfusion evaluated by Hypoxyprobe staining, and a near complete resolution of Perilipin+ adipocyte accumulation in the marrow (Fig. 7a–c & Supplementary Fig. 3a–e). Quantification of BM cytokines by ELISA did not reveal significant changes in the levels of SCF, CXCL12, IL1A and IFNG following NTN1 treatment of aged mice (Supplementary Fig. 3f). Taken together, these data demonstrate that NTN1 mediated DNA damage resolution is sufficient to restore the phenotypic defects of an aged

BM niche including restoration of BM vascular integrity and oxygenation, and a suppression of LepR+ MSC differentiation towards adipocytes within the aged BM microenvironment.

## NTN1 restores the regenerative capacity of an aged hematopoietic system

The resolution of phenotypic niche defects raised the possibility that NTN1 treatment could rejuvenate the functionality of the aged niche in their ability to maintain HSC fitness. To test this, we performed hematopoietic analysis of aged mice treated with NTN1 (Fig. 8a). While phenotypic HSC and MPP frequency remained unchanged, NTN1 treatment resulted in a modest decrease in BM cellularity, along with a reduction in myeloid cells and a concomitant improvement in B cells,

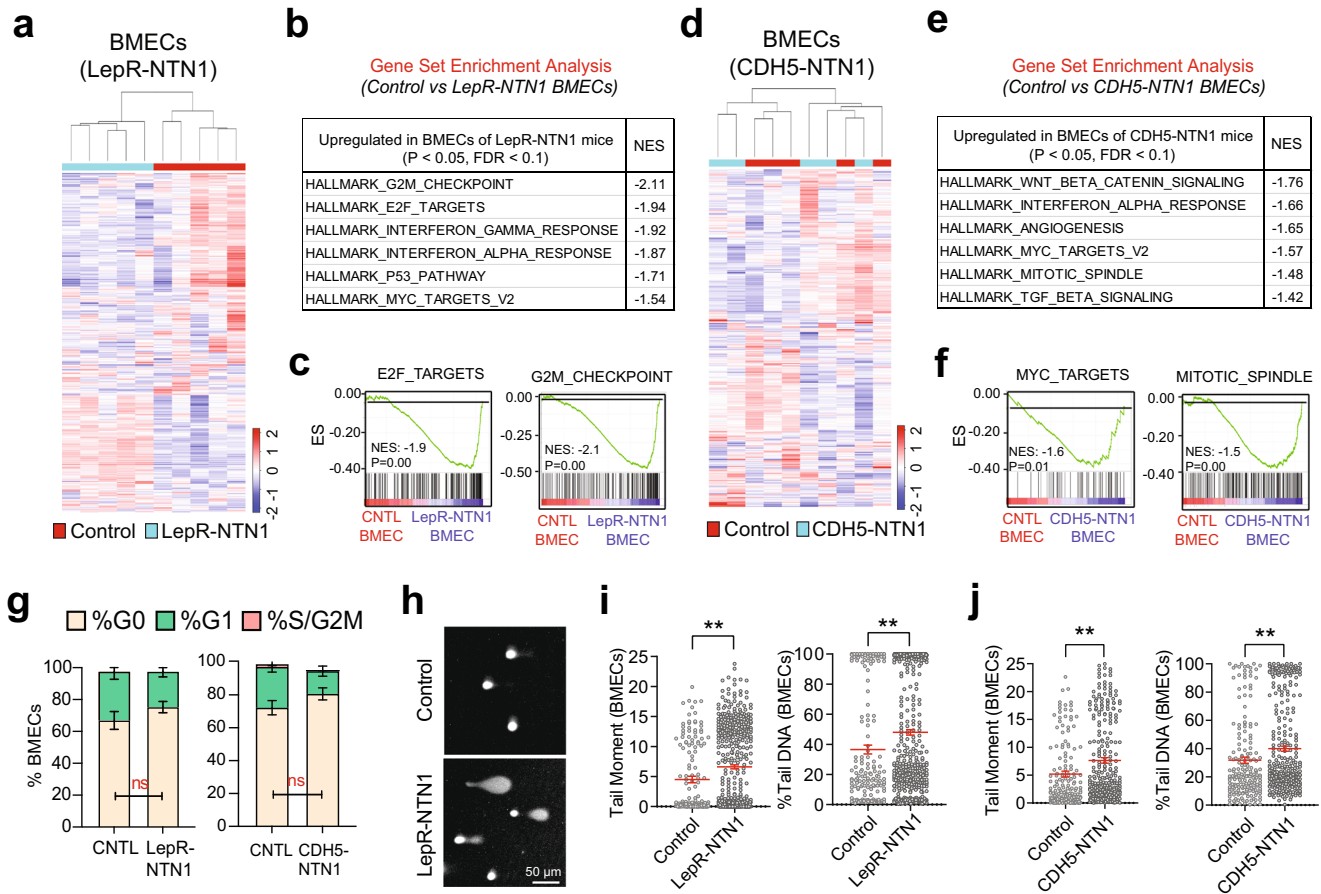

**Fig. 5 | Loss of NTN1 causes DNA damage within BMECs. a–c** RNA-Seq analysis of BMECs derived from young (5-6 month-old) *LepR-NTN1* mice. **a** Heatmap depicting hierarchical clustering of differentially expressed genes. **b** GSEA demonstrating activation of DDR pathways in BMECs of *LepR-NTN1* mice. **c** GSEA Enrichment plots demonstrating activation of E2F_TARGETS and G2M_CHECKPOINT pathways in BMECs of *LepR-NTN1* mice. **d–f** RNA-Seq analysis of BMECs derived from *CDH5-NTN1* mice. **d** Heatmap depicting hierarchical clustering of differentially expressed genes. **e** GSEA demonstrating activation of DDR pathways in BMECs of *CDH5-NTN1* mice. **f** GSEA Enrichment plots demonstrating activation of MYC_TARGETS and MITOTIC_SPINDLE pathways in BMECs of *CDH5-NTN1* mice. **g** Cell-cycle analysis of

BMECs derived from *LepR-NTN1* mice and *CDH5-NTN1* mice (*N* = 5 mice/group). **h** Representative IF images of single cell gel electrophoresis (alkaline comet assays) performed on BMECs. Alkaline comet analysis demonstrating an increase in average tail-moment and % Tail DNA in BMECs derived from both *LepR-NTN1* mice **i**, and *CDH5-NTN1* mice **j**, as compared to their littermate controls (*N* = 3 mice/group). Note that deletion of NTN1 within either MSCs or BMECs results in increased DNA damage within BMECs. Data is presented as the mean ± standard error of the mean (SEM). Statistical significance determined using two-tailed unpaired t-test. *P* < 0.05; **P* < 0.01; ***P* < 0.001. ns denotes statistically not significant.

as compared to PBS treated littermate controls (Supplementary Fig. 4a–c). Limiting dilution WBM transplantation analysis revealed that NTN1 treatment resulted in a ~4-fold increase in functional HSC numbers, based on number of recipient mice displaying long-term multi-lineage reconstitution (LTMR; >1% CD45.2+ engraftment) 16 weeks following transplantation (Fig. 8b, c). To evaluate HSC fitness, we performed competitive HSC transplantations utilizing donor HSCs derived from aged mice treated with NTN1 or vehicle (PBS). 2500 HSCs (CD45.2+) were FACS purified from each donor (*N* = 10 PBS-treated and *N* = 10 NTN1-treated aged mice), and were competitively transplanted into 5 recipient (CD45.1+) mice (500 CD45.2+ HSCs with 1.5 × 10⁶ CD45.1+ WBM competitor per recipient; a total of 50 recipients for each treatment group). Simultaneously, HSCs derived from *N* = 12 young (3 month-old, CD45.2+) mice were pooled and transplanted into 20 recipients (500 CD45.2+ HSCs with 1.5 × 10⁶ CD45.1+ WBM competitor per recipient), and served as young controls for comparison. As expected, HSCs from PBS-treated aged mice demonstrated the hallmark characteristics of an aged HSC including diminished long-term engraftment along with altered lineage reconstitution including increased myeloid-skewing and decreased T cell reconstitution, as compared to HSCs derived from young controls (Fig. 8d, e). Remarkably, HSCs derived from NTN1-treated aged mice exhibited long-term

engraftment and balanced lineage reconstitution indistinguishable from young controls (Fig. 8d, e). To assess whether NTN1 restores self-renewal potential of aged HSCs, we performed secondary transplantation assays utilizing WBM derived from primary recipients, which revealed that hematopoietic cells derived from NTN1-treated aged mice robustly maintain their long-term engraftment (~ 75% of young HSC engraftment levels), and exhibit balanced multilineage reconstitution equivalent to young HSCs (Fig. 8f, g). On the other hand, lineage analysis of hematopoietic cells derived from PBS-treated aged mice demonstrated significantly higher variance in lineage reconstitution with a near complete loss of myeloid or lymphoid output in a subset of recipients that reflect an exhaustion of HSC functional potential (Fig. 8g).

Collectively, these data suggest that the rejuvenating properties of NTN1 treatment on aged HSCs could arise from its beneficial effects on the aging microenvironment, a direct effect on the HSC itself, or a combination thereof. To determine whether direct effects of NTN1 on aged HSCs are sufficient for therapeutic rejuvenation, we took advantage of a recently described ex vivo HSC expansion platform utilizing polyvinyl alcohol (PVA) in a BSA-free medium with low dose KitL (10 ng/ml) and TPO (100 ng/ml) that expands transplantable HSCs ex vivo[25]. Using the PVA expansion protocol, we cultured HSCs derived

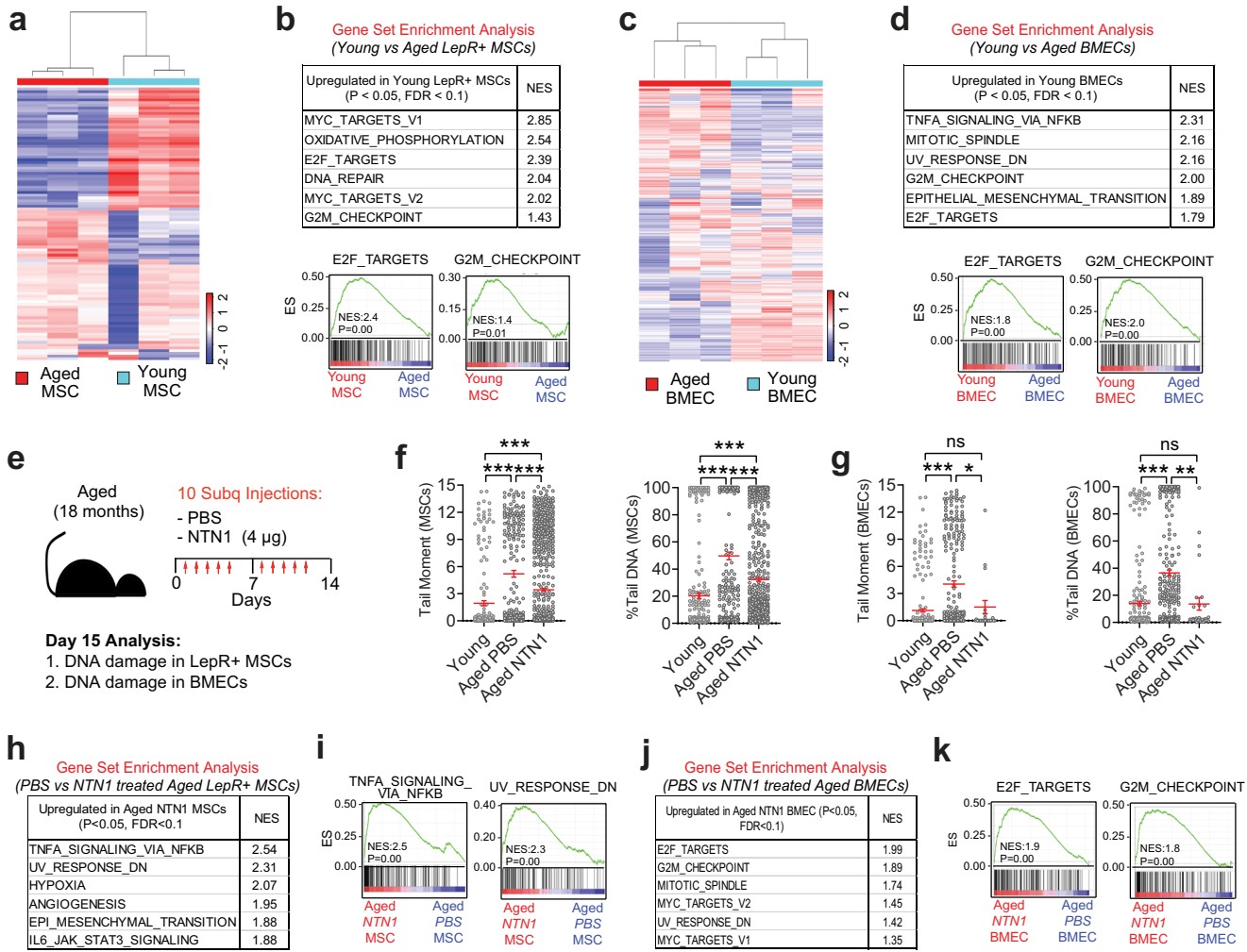

**Fig. 6 | NTN1 supplementation restores DNA damage within the aged BM niche.**
**a**, **b** RNA-Seq analysis of LepR+ MSCs derived from young (3 month) and aged (18 month) mice. **a** Heatmap depicting hierarchical clustering of differentially expressed genes. **b** GSEA demonstrating downregulation of DDR pathways in LepR+ cells of aged mice. **c**, **d** RNA-Seq analysis of BMECs derived from young (3 month) and aged (18 month) mice. **c** Heatmap depicting hierarchical clustering of differentially expressed genes. **d** GSEA demonstrating downregulation of DDR pathways in BMECs of aged mice. **e** Experimental design for NTN1 treatment. **f**, **g** Alkaline comet assays demonstrating an increase in average tail-moment and % Tail DNA in both LepR+ MSCs **f**, and BMECs **g**, of aged mice as compared to

young mice (N = 3 mice/group). Note that NTN1 treatment results in a significant reduction of DNA damage within aged MSCs and BMECs. Data is presented as the mean ± standard error of the mean (SEM). **h–k** RNA-Seq analysis of LepR+ MSCs and BMECs derived from aged (18 month) mice treated with PBS or NTN1. **h**, **i** GSEA demonstrating no significant upregulation of DDR pathways in LepR+ cells of aged mice treated with NTN1. **j**, **k** GSEA demonstrating an upregulation of DDR pathways in BMECs of aged mice treated with NTN1. Statistical significance determined using One-Way ANOVA with Tukey's correction for multiple comparisons. *P < 0.05; **P < 0.01; ***P < 0.001. ns denotes statistically not significant.

from aged mice (18 month-old male donors, N = 2 independent donors) for 11 days, in the presence of recombinant NTN1 or vehicle (PBS) (Supplementary Fig. 4d). Additionally, HSCs derived from young mice (3 month-old males) cultured in PVA medium with vehicle (PBS) served as controls for comparison (Supplementary Fig. 4d). While the total number of expanded cells following 11 days of ex vivo culture did not reveal significant differences across all 3 groups (Supplementary Fig. 4e), competitive transplantation of equal numbers of expansion cells (10,000 FACS sorted CD45.2+ expansion cells plus 1 × 10⁶ CD45.1 WBM competitor per recipient) into preconditioned CD45.1 recipients (N = 18–20 recipients/group) showed stark differences in their long-term (>6 month) engraftment ability (Fig. 8h, i). While expansion cells derived from PBS-treated aged HSCs manifested a complete loss of long-term engraftment potential, aged HSCs cultured in the presence of NTN1 demonstrated robust long-term engraftment and balanced multi-lineage reconstitution, outperforming expansion cells derived from young HSCs (Fig. 8i). To test the self-renewal ability of expanded HSCs, we performed secondary transplantation assays (Fig. 8h, j)

utilizing FACS sorted HSCs from primary donors (CD45.2+), and transplanting them into secondary recipients (CD45.1+), along with freshly isolated WBM competitor cells (1000 CD45.2+ HSC plus 1 × 10⁶ CD45.1 WBM competitor per recipient). While HSCs derived from both young control and NTN1 treated aged primary donors displayed similar levels of long-term (>4 months) engraftment, HSCs derived from NTN1 treated aged mice had more recipients with long term multilineage reconstitution (LTMR), while maintaining balanced lineage reconstitution when compared to young controls (Fig. 8j). Taken together, these data suggest that direct effects of NTN1 on aged HSCs are sufficient to rejuvenate their self-renewal potential to youthful levels.

We next sought to determine whether beneficial effects of NTN1 on the aged hematopoietic system could improve hematopoietic recovery following myelosuppressive injuries. Aged mice (18 month-old) were given a myelosuppressive dose of chemotherapy (150 mg/kg of 5 fluorouracil (5-FU)), and injected every other day with either recombinant NTN1 or PBS, for a total of 5 injections starting at Day 1

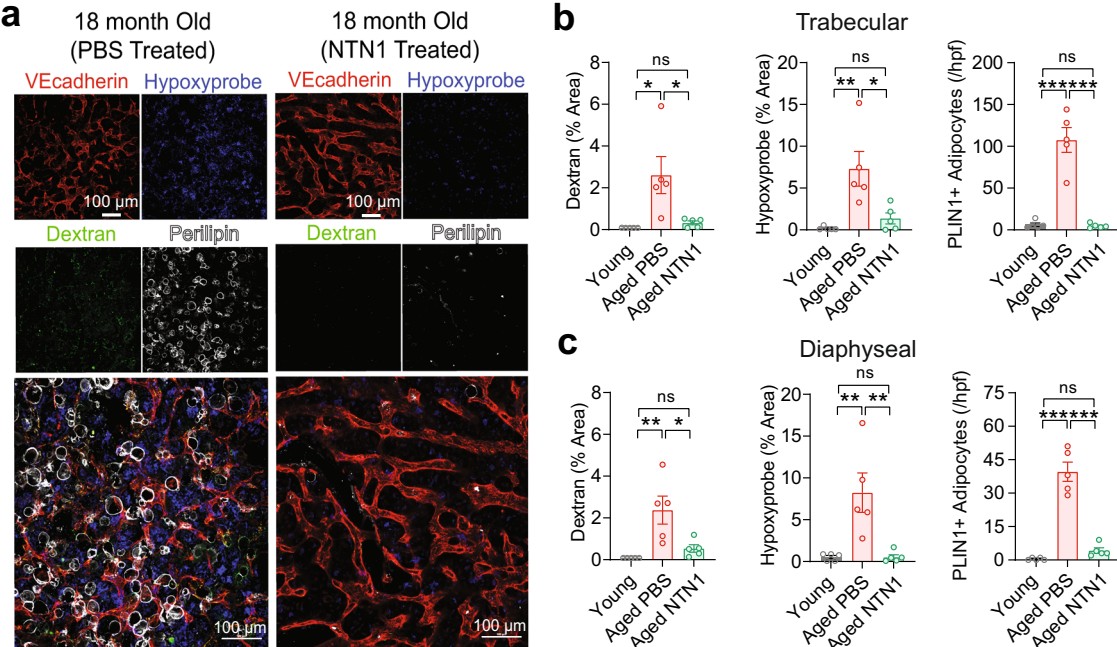

**Fig. 7 | NTN1 supplementation rejuvenates the aged BM niche. a** Representative immunofluorescence images of trabecular regions of femurs showing an improvement of aging-related BM niche defects following NTN1 treatment. **b, c** Analysis of BM niche function following NTN1 treatment of aged mice by determining vascular leakiness (10 kDa dextran extravasation), vascular perfusion (Hypoxyprobe staining), and BM adiposity (PLIN1 + adipocytes/high power field) in both trabecular **b**, and diaphyseal **c** regions of femoral BM, as compared to young controls ($N = 5$ mice/group). Data is presented as the mean ± standard error of the mean (SEM). Statistical significance determined using One-Way ANOVA with Tukey's correction for multiple comparisons. *$P < 0.05$; **$P < 0.01$; ***$P < 0.001$. ns denotes statistically not significant.

post-myelosuppression (Fig. 9a) and monitored for hematopoietic recovery (Fig. 9b–d). NTN1 treatment improved hematopoietic recovery after single dose 5-FU treatment, with platelet recovery starting at Day 7 (Fig. 9c) and white and red blood cells showing significant recovery by Day 10 (Fig. 9b, d). At Day 10 post 5-FU, NTN1 treated aged mice demonstrated a significant preservation of body weight, and a recovery of their peripheral blood lymphocytes and BM cellularity (Supplementary Fig. 4f–h). 5-FU treatment results in a significant expansion of phenotypic HSCs and HSPCs (~50 fold increase in HSC frequency at Day 10 post 5-FU as compared to steady state) to replenish the BM cellularity and blood cell counts following which they eventually return to homeostatic levels. Consistent with their accelerated recovery of BM cellularity and the hematopoietic system, NTN1 treated aged mice demonstrated a decrease in frequency of HSCs and progenitors at Day 10 post 5-FU, as well as an overall decrease in progenitor activity (Supplementary Fig. 4f–j).

To determine whether beneficial effects of NTN1 on hematopoietic regeneration extends to reducing the severe myelosuppressive stress associated with serial chemotherapy regimens, we performed serial 5-FU (150 mg/kg) treatments on aged mice every 35 days for a total of four treatments, and assessed their survival for 140 days (Fig. 9e–g). Following the first injection of 5-FU, aged mice were treated with either PBS or NTN1 every other day, for a total of 5 injections (Fig. 9e) and did not receive additional NTN1 treatment following subsequent 3 doses of 5-FU. In line with the known loss of HSC self-renewal and regenerative capacity during aging, ~75% of the PBS treated aged mice succumbed to hematopoietic failure following the first two doses of 5-FU, with only 1 out 18 mice surviving all four doses (Fig. 9f). Conversely, NTN1 treated aged mice demonstrated 100% survival (12/12) (Fig. 9f). Additionally, serial 5-FU treatment resulted in significant loss of body weight in surviving PBS treated aged mice following every injection of 5-FU, while aged mice treated with NTN1 demonstrated a better preservation of their body weight (<10% average weight loss) throughout the duration of serial 5-FU regimen

(Fig. 9g). These data indicate that NTN1 treatment can enhance self-renewal and regenerative capacity of aged HSCs, as well as protect the hematopoietic system from progressive loss of stem cell reserves and preserve body weight; toxicities commonly associated with chemotherapy in older patients.

## NTN1 restores the dampened DNA damage response within aged HSCs

To elucidate mechanisms underlying NTN1-mediated HSC rejuvenation, we performed RNA-Seq analysis on HSCs derived from aged mice treated with PBS or NTN1 (herein referred to as aged-PBS HSCs or aged-NTN1 HSCs, respectively), as well as HSCs derived from young mice (referred to as young HSCs) (Fig. 10 & Supplementary Data 10, 11). To determine whether NTN1 restores transcriptional alterations within aged HSCs, we compared the transcriptomes of aged-NTN1 HSCs with a meta-analysis of 10 previously published HSC aging transcriptomic datasets[26]. GSEA revealed that aged-NTN1 HSCs, similar to young HSCs, aligned with the 'young HSC signature', indicating that treatment with NTN1 is sufficient to restore the principal age-deregulated transcriptional networks within HSCs (Fig. 10a, b). To identify molecular pathways modulated by NTN1 that promote HSC rejuvenation, we analyzed pathways that were differentially expressed during aging (young versus aged-PBS HSCs) as well as following NTN1 supplementation (aged-PBS versus aged-NTN1 HSCs). GSEA revealed a marked overlap of pathways upregulated in young HSCs and aged-NTN1 HSCs as compared to aged-PBS HSCs (Fig. 10c, d & Supplementary Data 11). As observed in aged BM niche cells, there was an over-representation of DDR related pathways including E2F_TARGETS, G2M_CHECKPOINT, MYC_TARGETS, and DNA_REPAIR that were upregulated in young HSCs and aged-NTN1 HSCs as compared to aged-PBS HSCs (Fig. 10c, d & Supplementary Data 11). Evaluation of E2F_TARGETS and DNA_REPAIR pathways demonstrated downregulation of key DDR genes within aged-PBS HSCs (Fig. 10e, f). Cell-cycle analysis demonstrated that

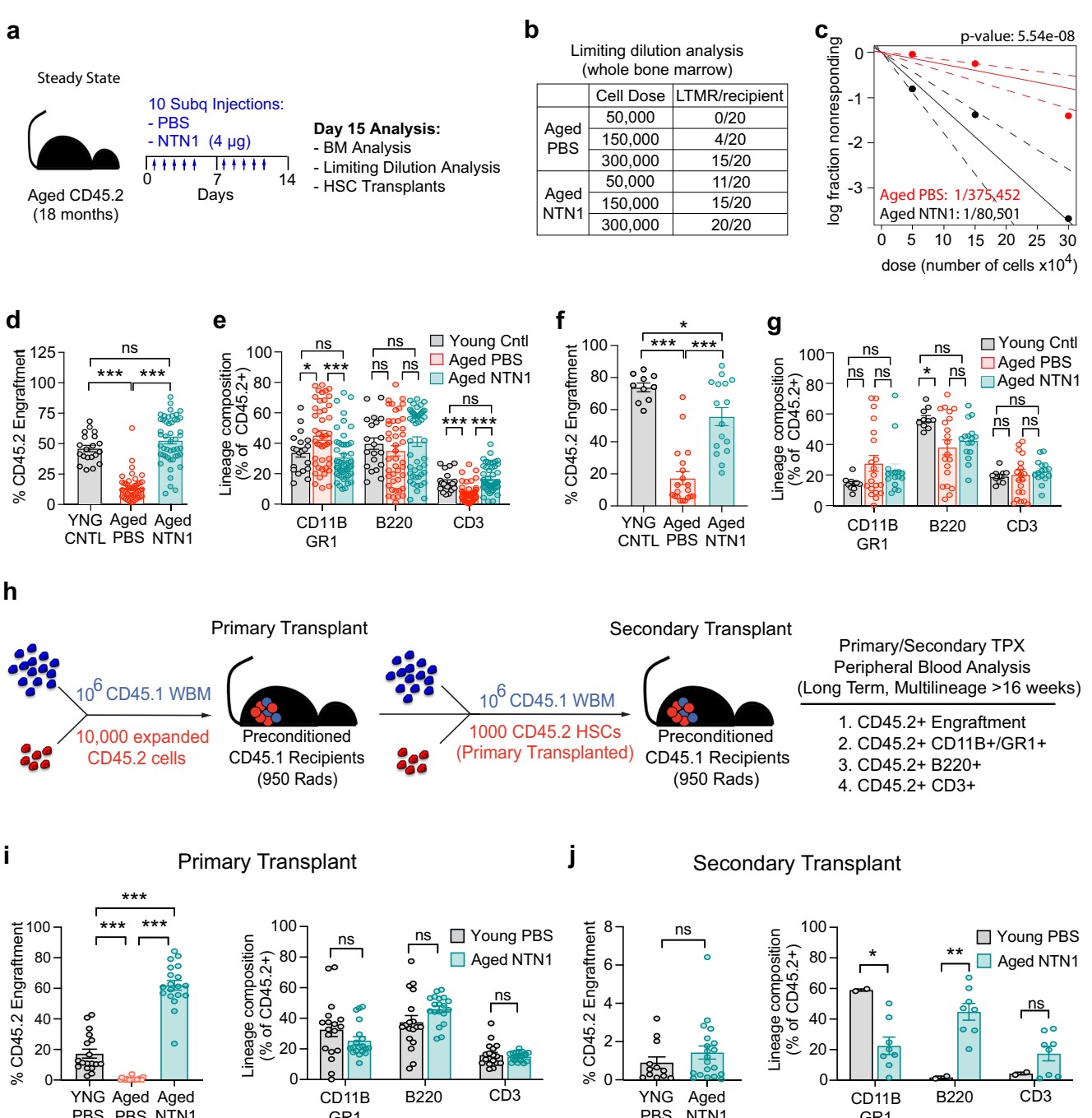

**Fig. 8 | NTN1 rejuvenates aged HSC functionality. a** Experimental design for NTN1 treatment in steady state aged mice. **b, c** Limiting dilution WBM transplantation analysis demonstrating ~4 fold increase in HSC numbers in aged mice treated with NTN1 as compared to PBS treated aged mice, evaluated by number of recipient mice demonstrating long-term multi-lineage reconstitution (LTMR) in their peripheral blood after 16 weeks following transplantation ($N = 3$ donors/group; $N = 20$ recipients/group per cell dose). Line graph displaying estimates of HSC frequency in the indicated groups with dashed lines representing 95% confidence intervals. Stem cell frequency and significance were determined using Extreme Limiting Dilution Analysis (ELDA)[61]. **d, e** Competitive HSC transplantation demonstrating that NTN1 treatment of aged mice restores long-term (>4 months) HSC engraftment potential **d** and myeloid-biased output **e**, similar to HSCs from young donors ($N = 10$ donors/group; Recipients $N = 20$ (Young), $N = 44$ (Aged-PBS), $N = 48$ (Aged-NTN1)). **f, g** Secondary transplantation demonstrating a preservation of serial repopulation **f** and balanced lineage reconstitution abilities **g** in donor cells derived from NTN1 treated aged mice ($N = 10$ donors/group; Recipients $N = 10$ (Young), $N = 20$ (Aged-PBS), $N = 16$ (Aged-NTN1)). **h** Experimental design describing HSC

transplantation strategy to assess the direct effects of NTN1 treatment on aged HSC function (Supplementary Fig. 4d). Following an 11 day ex vivo expansion, 10,000 FACS sorted expansion cells (DAPI⁻CD45.2⁺) from each donor were transplanted along with 10⁶ WBM competitor cells (CD45.1) into preconditioned CD45.1 recipients. Following long-term (>6 month) engraftment, CD45.2+ HSCs were FACS purified from primary recipients, and transplanted along with fresh CD45.1 WBM competitor cells into preconditioned CD45.1 secondary recipients. **i** Bar graphs demonstrating long-term engraftment potential in ex vivo cultured aged HSCs, as compared to HSCs derived from young mice. Note that NTN1 treatment restores LTMR ability of aged HSCs ($N = 2$ donors/group; Recipients N = 18 (Young), $N = 20$ (Aged-PBS), $N = 19$ (Aged-NTN1)). **j** Bar graphs demonstrating serial repopulation and LTMR (>4 months) (Recipients $N = 12$ (Young), $N = 20$ (Aged-NTN1)). Data is presented as the mean ± standard error of the mean (SEM). Statistical significance determined using two-tailed unpaired $t$-test (for pairwise comparisons), and One-Way ANOVA with Tukey's correction for multiple comparisons. *$P < 0.05$; **$P < 0.01$; ***$P < 0.001$. ns denotes statistically not significant.

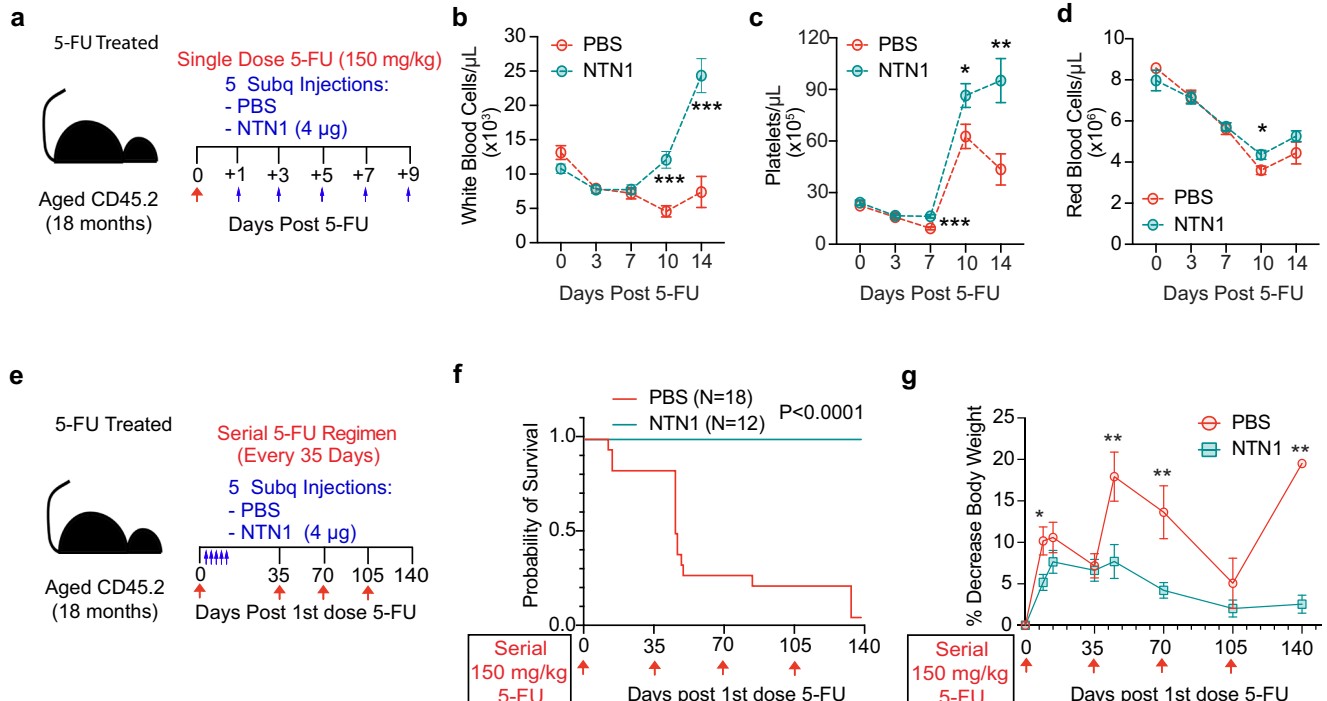

**Fig. 9 | NTN1 restores the regenerative capacity of aged HSCs. a** Experimental design for NTN1 treatment following a single dose of 5-FU. **b–d** Complete blood counts quantifying WBCs **b**, Platelets **c**, and RBCs **d** (PBS *N* = 19 (Days 0 & 3), *N* = 17 (Days 7 &10), *N* = 6 (Day 14), NTN1 *N* = 20 (Days 0, 3 & 7), *N* = 18 (Day 10), *N* = 7 (Day 14)). **e** Experimental design for NTN1 treatment following serial 5-FU treatment. **f** Kaplan-Meier survival curve (PBS *N* = 18, NTN1 *N* = 12). Statistical significance determined using Log-rank (Mantel-Cox) test. **g** Analysis of total body weight loss

following serial 5FU injections (PBS *N* = 25 (Days 0, 9), *N* = 15 (Day 14), *N* = 15 (Day 35), *N* = 14 (Day 45), *N* = 4 (Day 70), *N* = 4 (Day 105), *N* = 1 (Day 140); NTN1 *N* = 20 (Days 0, 9), *N* = 12 (Days 14, 35, 45, 70, 105 & 140)). Data is presented as the mean ± standard error of the mean (SEM). Statistical significance determined using two-tailed unpaired t-test. *P* < 0.05; **P* < 0.01; ***P* < 0.001. ns denotes statistically not significant.

NTN1 treatment did not result in gross changes to the G0/G1/SG2M fractions in HSCs of aged mice (Fig. 10g, h). Besides cell-cycle and DDR pathways, aged-NTN1 HSCs also demonstrated an upregulation of the OXIDATIVE_PHOSPHORYLATION pathway recently shown to regulate HSC function during aging[27] (Fig. 10d). However, NTN1 treatment did not alter mitochondrial membrane potential or reactive oxygen species (ROS) levels within HSCs of aged mice (Fig. 10i–k), indicating that NTN1 mediated aged HSC rejuvenation likely results from reactivation of dampened DDR within aged HSCs[28,29]. Supporting this, GSEA utilizing REACTOME and KEGG databases confirmed that aging is associated with a global down-regulation of DNA repair genes and pathways in HSCs including Base Excision Repair (BER), Nucleotide Excision Repair (NER), Homologous Recombination (HR) and Mismatch Repair (MMR) (Supplementary Fig. 5a, b & Supplementary Data 11). Notably, treatment with NTN1 restored the expression of DDR genes and pathways within aged HSCs (Fig. 10d–f, Supplementary Fig. 5a, b & Supplementary Data 11). To ascertain whether DDR downregulation is a conserved feature of HSC aging, we performed GSEA of genes dysregulated within aged HSCs identified in the meta-analysis of published HSC aging transcriptomes (Supplementary Data 12)[26]. GSEA revealed that downregulation of DDR pathways and genes represents a consistent feature of HSC aging (Supplementary Fig. 5c–f) that explains the accumulation of DNA damage within aged HSCs[30,31]. To verify whether NTN1 mediated DDR activation is sufficient to resolve DNA damage within aged HSCs, we quantified the levels of γ-H2AX which revealed that NTN1 treatment resulted in a reduction in their nuclear γ-H2AX foci to basal levels (Fig. 10l, m). Alkaline comet assays confirmed that NTN1-treatment resolves DNA damage in aged HSCs to levels observed in young mice (Fig. 10n, o), demonstrating that NTN1 supplementation reactivates DDR and resolves DNA damage

within aged HSCs. Given that NTN1 supplementation promoted hematopoietic recovery in aged mice following myelosuppressive injury (Fig. 9), we next assessed whether NTN1 treatment can accelerate DNA damage resolution following 5-FU treatment (Fig. 10p, q). Comet analysis confirmed that NTN1 supplementation was able to significantly reduce DNA damage within HSCs of aged mice treated with 5-FU (Fig. 10q). Additionally, cell-cycle analysis revealed no significant changes in HSCs of aged mice supplemented with NTN1 following 5-FU exposure, as compared to PBS treated aged mice (Fig. 10p, r and Supplementary Fig 6a, b). Notably, HSCs derived from *LepR-NTN1* and *CDH5-NTN1* mice displayed increased DNA damage, confirming that niche-derived NTN1 directly regulates DDR within HSCs (Supplementary Fig. 6c–h). Taken together, these data demonstrate that while a young niche provides adequate NTN1 to support niche integrity and HSC self-renewal by maintaining a robust DDR, age-related decline in niche-derived NTN1 contributes towards a dysfunctional niche and damaged HSCs resulting from dampened DDR and DNA damage accumulation. Supplementation of aged mice with NTN1 resolves DNA damage within the BM niche and HSCs, and rejuvenates an aging hematopoietic system (Supplementary Fig. 7).

## Discussion
An unresolved issue regarding aging is whether the regenerative potential of aged HSCs and their supportive niches can be restored to youthful levels. It was recently shown that aged HSCs are refractory to systemic anti-aging interventions that have shown improvements in other stem cell systems[32]. Contrarily, few studies have demonstrated that some defects of an aged HSC can be mitigated by targeting HSC-intrinsic mechanisms[27,33–36] or their supportive niche cells[22,37–40], evident by improvements in HSC

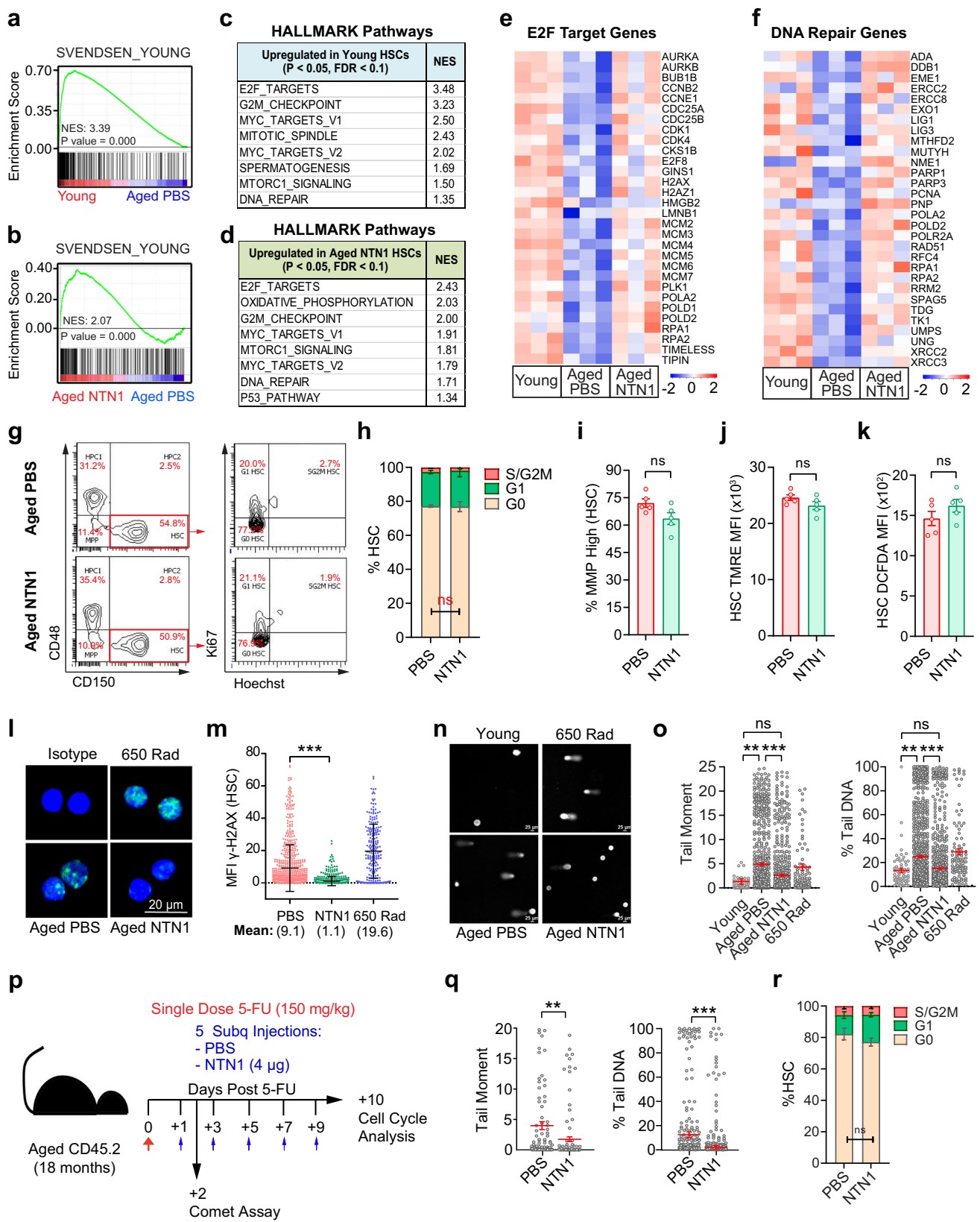

engraftment potential and reductions in their myeloid skewing. However, whether age-associated decline of HSC self-renewal ability, the defining property underlying their regenerative potential, can be functionally restored remained unknown. Here, we demonstrate that NTN1 treatment of aged mice rejuvenates their BM niche and restores HSC regenerative potential. The beneficial effects of

NTN1 on the aged hematopoietic system translate into a strong survival benefit and preservation of body weight during serial myelosuppressive chemotherapy. These findings have therapeutic implications for improving hematopoietic healthspan and improving outcomes following chemotherapeutic regimens in older patients.

**Fig. 10 | NTN1 reactivates the dampened DDR within aged HSCs. a–f** RNA-Seq analysis of HSCs derived from young (3 month), and aged (18 month) mice treated with PBS or NTN1. **a, b** Comparison with an HSC aging meta-analysis (Svendsen et al)[26] demonstrating alignment of young HSCs **a**, and aged NTN1 HSCs **b**, with the 'young HSC gene signature'. **c, d** GSEA showing upregulation of DDR pathways in both young HSCs **c** and aged NTN1 HSCs **d**, when compared with aged PBS HSCs. **e, f** Heatmaps of E2F-Targets **e**, and DNA repair genes **f**, downregulated in aged HSCs, and restored after NTN1 treatment. **g** Representative contour plots (flow cytometry) for HSC cell-cycle analysis. **h** Bar graphs showing HSC cell-cycle distribution (N = 5 mice/group). **i, j** Assessment of mitochondrial membrane potential (MMP) by flow cytometric quantification of Tetramethyl rhodamine ethyl ester (TMRE) in HSCs of aged mice following PBS/NTN1 treatment. Bar graphs demonstrating no significant changes in % of HSCs with high MMP **i**, or average TMRE per HSC **j**, following NTN1 treatment (N = 5 mice/group). **k** Assessment of ROS by flow cytometric quantification of dichlorodihydrofluorescein diacetate (DCFDA) in HSCs of aged mice following PBS/NTN1 treatment (N = 5 mice/group). **l, m** Representative IF images for γH2AX staining within HSCs **l**, showing a decrease in γH2AX immunoreactivity **m**, following NTN1 treatment of aged mice (N = 4 mice/group). HSCs from irradiated young mice were utilized as positive controls. **n, o** Representative IF images of alkaline comet analysis of HSCs **n**, showing decrease in average Tail-moment **o**, and a reduction in % Tail DNA within aged HSCs following NTN1 treatment (N = 5 mice/group). **p** Experimental design for assessment of DNA damage and cell cycle status within HSCs following a single dose of 5-FU. **q** Alkaline comet analysis showing significantly reduced DNA damage within HSCs of aged mice treated with NTN1 (N = 2 mice/group). **r** HSC cell cycle analysis in aged mice following 5-FU treatment (N = 5 mice/group). Data is presented as the mean ± standard error of the mean (SEM). Significance determined using two-tailed unpaired t-test (pairwise comparisons), and One-Way ANOVA (multiple comparisons). *P < 0.05; **P < 0.01; ***P < 0.001. ns denotes statistically not significant.

Recent studies identified NEO1 as a receptor for NTN1 on HSCs and that NTN1-NEO1 signaling likely promotes HSC dormancy in young mice[13,14]. Gulati et al reported that HSCs with surface expression of NEO1 (Hoxb5+NEO1+) represent HSCs with an impaired regenerative capacity and a myeloid-biased output[14]. Renders et al demonstrated that HSCs derived from NEO1 mutant mice perform poorly in serial transplantation assays, indicating an essential role for NEO1 in maintaining HSC repopulating activity[13]. Collectively, both studies suggest that NTN1-NEO1 signaling is essential to preserve HSC fitness in young mice wherein HSCs that do not receive adequate NTN1 signals exhibit increased NEO1 expression and manifest a decline in their fitness. While these studies utilized HSCs derived from young mice for functional transplantation analyses to illustrate the role of NTN1-NEO1 signaling in the hematopoietic system, the critical role of NTN1 signaling in maintaining HSC fitness during physiological aging had not been evaluated. This represented a crucial unanswered question in light of the observations that HSC NEO1 expression is primarily detected in aged animals, that NTN1 overexpression in young mice did not improve HSC functionality reflecting sufficient NTN1 signaling in the young hematopoietic system, and that aging is associated with a deficiency of NTN1[13,14]. Furthermore, given that NTN1 has been shown to regulate endothelial function, it raised the possibility that NTN1 could regulate the fitness and functionality of the BM niche during aging. While significant progress has been made in unraveling the role of BM niche derived extrinsic signals in regulating HSC function, the signaling mechanisms between cellular constituents comprising the niche that regulate vascular integrity, oxygenation, and adiposity within the aging BM remain poorly understood. Here, we identify NTN1 as a linchpin molecule that regulates BMEC-LepR+ MSC niche interactions reciprocally by preserving the integrity of the BM vascular niche and preventing adipocyte accumulation. Periarteriolar smooth muscle cells (SMCs) were initially described as a putative niche cell that provides NTN1 to HSCs within the BM[13]. However, conditional deletion of NTN1 within SMCs did not recapitulate HSC defects observed upon ubiquitous deletion of NTN1[13]. Here, we define LepR+ MSCs and BMECs as the bona fide sources of niche-derived NTN1 that not only maintain HSC fitness, but also preserve niche integrity within the BM. We demonstrate that niche-derived NTN1 plays an essential role in preventing DNA damage within MSCs and BMECs. We identify that aging is associated with a downregulation of DDR within both BMECs and MSCs, identifying a conserved mechanism underlying age-related defects within the BM niche. Notably, diminished DDR within aged niche cells likely explains the higher chemo-sensitivity observed during aging. Importantly, our findings demonstrate that niche-derived NTN1 is essential to prevent DNA damage within HSCs, illuminating a role for BM niche cells in regulating HSC DDR. By performing transcriptomic analysis of aged HSCs, we confirm that DDR downregulation is a conserved attribute of HSC aging as described previously[28,29,41] and our findings indicate that age-related decline in

niche-derived NTN1 likely underlie DDR downregulation within aged HSCs and BM niche cells.

It has become increasingly evident that the HSC pool within the aged BM is heterogeneous[42]. Elegant label-retention experiments have shown that only a small fraction of phenotypically defined HSCs within the aged BM represent true stem cells with serial repopulating activity[43]. This subset of bona fide stem cells were shown to reside in the H2B-GFP label retaining fraction. However, despite differences in their repopulating activity, both the GFP+ (label-retaining) and the GFP- fractions of aged HSCs exhibit a myeloid-biased differentiation when compared to young HSCs, indicating that all HSCs in the aged BM are functionally defective to some degree[43]. In this regard, the reversal of aged HSC myeloid bias observed in both ex vivo HSC expansion-transplantation assays (Fig. 10i) as well as following in vivo treatment of aged mice with NTN1 (Fig. 10d, e) demonstrates a true rejuvenation of aged HSC functionality. It remains plausible that a small-subset of yet to be identified 'young-like' HSCs exists within the aged BM that retain the functional potential equivalent to young HSCs. In this context, the rejuvenating effects of NTN1 on aged HSCs could potentially arise from an expansion of these 'young-like' HSCs within the aged BM. Alternatively, NTN1 might selectively rejuvenate specific subsets within the heterogeneous pool of aged HSCs that are not irreversibly damaged, and are therefore amenable for rejuvenation. Both of these possibilities are supported by the limiting-dilution transplantation analyses which demonstrates a ~5-fold increase in numbers of bona fide long-term repopulating HSCs within the aged BM following NTN1 treatment (Fig. 10b, c). These possibilities remain to be investigated further to obtain a better understanding of HSC heterogeneity within the aged BM, and to identify the HSC subsets that respond to NTN1 treatment.

Observations from human genetic disorders and studies employing diverse array of genetic model systems have unequivocally established that DDR insufficiency causes premature aging phenotypes and that aging results in generalized impairment of DDR capacity[3,44–46]. However, it remained unresolved whether DDR reactivation is sufficient to rejuvenate stem cell regenerative potential during aging, particularly within the mammalian system[3,44,45]. Indeed, accumulation of DNA damage is a hallmark feature of aging and widely considered one of the principal causes of the aging process[47]. However, the identification of master regulators of DNA repair that affects multiple DDR pathways and the development of strategies to reactivate the DDR in aging tissues are lacking[3,44,47]. Here, we demonstrate that restoration of DDR by NTN1 is sufficient to rejuvenate an aged BM niche and restore HSC function to youthful levels. The findings raise the possibility that DDR downregulation could underlie a fundamentally conserved attribute of tissue specific stem cells and their supportive niches. Supporting this, a recent study comparing the transcriptomes of young and aged muscle stem cells (MuSCs) identified that E2F_TARGETS

and G2M_CHECKPOINT were amongst the most significantly downregulated pathways during aging[48]. Importantly, the same pathways were upregulated following exercise in rejuvenated aged MuSCs[48].

While activities of specific proteins mediating DNA repair in aged HSCs and niche cells were not explored in this study owing to the limitations associated with the study of rare cell populations, NTN1 mediated upregulation of diverse DDR pathways at the transcriptional level, coupled with resolution of DNA damage in aged niche cells and HSCs are indicative of global restitution of DDR pathways. In this context, it remains plausible that DDR upregulation and DNA damage resolution within HSCs and BM niche cells of aged mice treated with NTN1 could result from an overall improvement in vascular function within the BM. While the effects of NTN1 infusion on other systemic aging parameters were not investigated in this study, the beneficial effects of NTN1 on diverse cell types including BM MSCs, ECs and HSCs, the premature hair greying observed in *LepR-NTN1* mice implicating altered melanocyte stem cell function, along with the ability of NTN1 to preserve body weight during serial chemotherapy are suggestive of therapeutic potential beyond the BM. Since DDR plays a critical role during diverse processes including clonal disorders, regeneration following myelosuppressive therapies, neurodegenerative disorders, and the overall systemic aging process[49], the receptors/signal transducers/effectors in HSCs and niche cells that are involved in NTN1 mediated activation of the DDR need to be explored further.

## Methods

### Animals
All murine experiments were conducted in accordance with the Association for Assessment and Accreditation of Laboratory Animal Care, Intl. (AAALAC) and National Institutes of Health (NIH) Office of Laboratory Animal Welfare (OLAW) guidelines, and under the approval of the Hackensack Meridian Health and Center for Discovery and Innovation Institutional Animal Care and Use Committee (IACUC). Young and aged C57BL/6 (CD45.2, JAX Stock No: 000664) mice were purchased from the National Institutes on Aging, and the Jackson Laboratory (Bar Harbor, ME). B6.129(Cg)-*Lepr*$^{tm2(cre)Rck}$/J (JAX Stock No. 008320) mice, B6.129(SJL)-$^{NTN1tm1.1Tek}$/J (JAX Stock No. 028038) mice and B6.SJL-*Ptprc*$^a$ *Pepc*$^b$/BoyJ (CD45.1; JAX stock No. 002014) mice were purchased from the Jackson Laboratory. *Cdh5(PAC)-creERT2* mice were obtained from Ralf H. Adams at The Max Planck Institute for Molecular Biomedicine[21,50]. Mice were acclimatized in the vivarium for at least 2 weeks prior to experimental use. B6.129(Cg)-*Lepr*$^{tm2(cre)Rck}$/J, *Cdh5(PAC)-creERT2* and B6.129(SJL)-$^{NTN1tm1.1Tek}$/J mice were maintained on a C57BL/6J (CD45.2) genetic background. To induce *Cdh5(PAC)-creERT2*-mediated recombination, mice were maintained on Custom Teklad 2020 Feed supplemented with 0.025% w/w tamoxifen (Envigo) *ad libitum* starting at 6-10 weeks of age for four consecutive weeks. Age matched *cre*-negative littermate mice underwent the same tamoxifen induction regimen, and were utilized as controls. Mice were allowed to recover for at least 4 weeks post-tamoxifen induction prior to experimental analysis. All mice were housed in NexGen Individually Ventilated Cages (IVC) with HEPA-filtered air exchange (Allentown, Inc.) and maintained on PicoLab Rodent Diet 20 (Lab Diet 5053) and water *ad libitum*. All mice were maintained in specific-pathogen-free housing.

### Functional Hematopoietic Assays
For competitive bone marrow transplants, CD45.1 transplant recipient mice (12-16 week old; B6.SJL-*Ptprc*$^a$ *Pepc*$^b$/BoyJ) were pre-conditioned with 950 Rads (split-dose; 2 hours apart) total body X-Ray irradiation (RS 2000 Small Animal Irradiator) 4 hours prior to transplantation. Hematopoietic stem cells, defined as DAPI-lineage-SCA1+cKIT+CD150+CD48-, were FACS sorted to purity. For primary transplantations, pre-conditioned CD45.1 recipients were injected via retro-orbital sinus with the indicated numbers of CD45.2+ donor HSCs along with CD45.1+ WBM competitor cells. For secondary WBM transplantations, BM cells from primary recipients were isolated and 2×10⁶ donor cells were injected into preconditioned CD45.1 recipient mice. For secondary HSC transplantations, CD45.2+ HSCs from long-term engrafted primary transplant recipients were FACS purified, and injected along with freshly isolated CD45.1+ WBM competitor cells into preconditioned CD45.1 recipients. Multi-lineage engraftment was monitored post-transplantation by flow cytometry analysis of red blood cell (RBC)-lysed peripheral blood (PB) stained with antibodies raised against CD45.2, CD45.1, TER119, GR1, CD11B, B220, and CD3. For myelosuppression studies, 5-Fluorouracil (Fresenius Kabi 101710) was diluted in PBS to achieve a dose of 150 mg/kg body weight, and administered intraperitoneally in a volume of 150 µL.

### Comet assays
Comet assays were performed utilizing the CometAssay® Electrophoresis System II (R&D Systems, 4250-050-ES) as per manufacturer's recommendations. Briefly, 5000-25000 MSCs, BMECs or HSCs were FACS sorted into 1.5 mL micro-centrifuge tubes containing 0.75 mL ice-cold PBS. Cells were transferred to pre-warmed 1.5 mL micro-centrifuge tubes (37 °C) placed on a heating block, containing 300 µL molten low-melting agarose. Cells were gently mixed with agarose and 30 µL of agarose-cell suspension was transferred onto each well of the pre-warmed Comet Slide. Slides were placed at 4 °C for 30 minutes after which they were immersed in CometAssay Lysis Solution overnight at 4 °C. After draining excess lysis solution, slides were placed in alkaline unwinding solution (200 mM NaOH, 1 mM EDTA, pH>13) for 1 hour at 4 °C, following which electrophoresis was performed in 850 mL chilled alkaline electrophoresis solution (200 mM NaOH, 1 mM EDTA, pH>13) at 21 V constant voltage for 30 minutes. Slides were sequentially washed in water and 70% ethanol, air dried, and counterstained with SYBR Gold (ThermoFisher Scientific) solution as per manufacturer's recommendations. Images were acquired on a Ti2 epifluorescence microscope (Nikon). Image analysis was performed utilizing the CometAssay® Analysis Software with default parameters, and all identified comets were manually reviewed for accuracy.

### NTN1 Treatment
Recombinant murine NTN1 protein (R&D Systems 1109-N1/CF) was reconstituted in PBS at 100 µg/mL and stored at −20 °C in single-use aliquots. 4 µg NTN1 was diluted to a final volume of 100 µL in PBS, and injected subcutaneously. PBS injections served as vehicle controls.

### 5-FU Treatment
5-Fluorouracil (Fresenius Kabi 101710) was diluted in PBS to achieve a dose of 150 mg/kg body weight, and administered intraperitoneally in a volume of 150 µL.

### Buffers and Media
Phosphate Buffered Saline (PBS pH 7.4, Corning 21-040 CV). Magnetic Activated Cell Sorting (MACS) buffer: PBS without Ca⁺⁺/Mg⁺⁺ (pH 7.4) (Corning 21-040-CV) containing 0.5% bovine serum albumin (BSA; Fisher Scientific BP1605) and 2 mM EDTA (Corning 46-034-CI). Digestion buffer: Hanks Balanced Salt Solution (Life Technologies 14065) containing 20 mM HEPES (Corning 25-060-CI), 2.5 mg/mL Collagenase A (Roche 11088793001), and 1 unit/mL Dispase II (Roche 04942078001). Endothelial Growth Medium. 1:1 ratio of Low-glucose DMEM (ThermoFisher Scientific 11885-084) and Ham's F-12 (Corning 10-080-CV), supplemented with 20% heat-inactivated FBS (Denville Scientific FB5002-H), 1% antibiotic-antimycotic (Corning 30-004-CI), 1% non-essential amino acids (Corning 25-025-CI), 10 mM HEPES (Corning 25-060-CI), 100 µg/mL heparin (Sigma-

Aldrich H3149), and 50 μg/mL endothelial cell growth supplement (Corning 356006).

## MSC cultures for NTN1 immunohistochemistry

MSC cultures were established as follows[51]. Femurs and tibiae were gently crushed using a mortar and pestle, enzymatically dissociated with Digestion buffer for 15 minutes at 37 °C, filtered (40 μm; Corning 352340), and washed in MACS buffer. WBM was depleted of terminally differentiated hematopoietic cells using a murine Lineage Cell Depletion Kit (Miltenyi Biotech 130-090-858) according to the manufacturer's recommendations. BMECs were immunopurified from cell suspensions using sheep anti-rat IgG Dynabeads (ThermoFisher Scientific 11035) pre-captured with a CD31 antibody (MEC13.3; Biolegend) in MACS buffer, according to the manufacturer's suggestions. CD31-depleted stromal cells were cultured in fibronectin-coated tissue culture plates in endothelial growth media. Stromal cells were selected for seven days in serum- and cytokine-free StemSpan SFEM (StemCell Technologies, Inc. 09650) media. Stromal cells were stained with antibodies against VECAD (BV13; Biolegend), CD31 (390; Biolegend), and CD45 (30-F11; Biolegend) and FACS sorted (FACS ARIA III, BD Biosciences) for purity. BMECs were defined as CD45⁻CD31⁺VEcadherin⁺, and stromal cells were defined as CD45⁻CD31⁻VEcadherin⁻. Cells were cultured in endothelial growth medium at 37 °C, 5% $CO_2$, and 20% $O_2$ in 70% relative humidity. Growth media was replaced every two days and cells were passaged 1:2 at 95% confluency with Accutase Cell Detachment Solution (Biolegend 423201) according to the manufacturer's suggestions.

## Hematopoietic cell quantification

To quantify total hematopoietic cells, femurs were gently crushed with a mortar and pestle, enzymatically disassociated with Digestion buffer for 15 minutes at 37 °C, filtered, and washed in MACS buffer. Viable cell numbers were quantified using a hemocytometer with Trypan Blue (Life Technologies 15-250-061) dye exclusion. To quantify hematopoietic stem and progenitor cells (HSPCs) in the BM, femurs and tibiae were flushed using a 26 G x 1/2 needle with MACS buffer. Gating strategy described under flow cytometry.

## Hematopoietic progenitor activity

Colony-forming units (CFUs) in semi-solid methylcellulose were quantified to assess hematopoietic progenitor activity. WBM was flushed from femurs and tibiae using a 26G x 1/2 needle with MACS buffer. Viable cell counts were determined with a hemocytometer using Trypan Blue. WBM cells were plated in duplicate in Methocult GF M3434 methylcellulose (StemCell Technologies 03444) according to the manufacturer's recommendations. Colonies were scored for phenotypic CFU-GEMM, CFU-GM, CFU-G, CFU-M, and BFU-E colonies using a SMZ1270 Stereo-Microscope (Nikon).

## Flow cytometry

Prior to cell surface staining, $F_c$ receptors were blocked using an antibody against CD16/32 (93; Biolegend) in MACS buffer for 10 minutes at 4 °C. Blocked samples were subsequently stained with fluorochrome-conjugated antibodies in MACS buffer for 30 minutes at 4 °C. Samples stained with biotinylated anti-LEPR antibody were washed and stained with Streptavidin-conjugated fluorochromes for 30 minutes at 4 °C. Stained cells were washed in MACS buffer and fixed in 1% paraformaldehyde (PFA, MP Bio 0219998380) in PBS with 2 mM EDTA (Corning 46-034-CI). Sample data was collected and analyzed using a flow cytometer (Fortessa, BD Biosciences) with FACS DIVA 8.0.1 software (BD Biosciences). Fluorescence compensation was performed utilizing single-stained controls of BM cells. Gates were established using unstained controls and standard fluorescence minus one strategies. List of antibody clones utilized for Flow Cytometry are included in Supplementary Data 13. Gating strategy for flow cytometry

are defined as follows[52]. KLS: Lineage (TER119/CD11B/GR1/B220/CD3)⁻ cKIT⁺ SCA1⁺, HSC: Lineage⁻ cKIT⁺ SCA1⁺CD48⁻ CD150⁺, MPP: Lineage⁻ cKIT⁺ SCA1⁺ CD48⁻ CD150⁻, Myeloid: CD45⁺ CD11B⁺ GR1⁺, B cells: CD45⁺ B220⁺, T cells: CD45⁺ CD3⁺, BMECs: CD45⁻ TER119⁻ CD31⁺ VEcadherin⁺, BM LEPR+ cells: CD45⁻ TER119⁻ CD31⁻ LEPR⁺.

## Peripheral blood analysis

Peripheral blood (PB) was collected using 75 mm heparinized glass capillary tubes (Kimble-Chase 41B2501) via retro-orbital sinus bleeds into micro-centrifuge tubes containing PBS with 10 mM EDTA. Blood indices were analyzed using an automated hematology analyzer (Element HT5, Heska). To quantify steady state lineage⁺ hematopoietic cells and mutli-lineage HSC engraftment, PB was depleted of red blood cells (RBC Lysis Buffer; Biolegend 420301) according to the manufacturer's recommendations, stained with indicated fluorophore-conjugated antibodies, and analyzed using flow cytometry.

## Mitochondrial membrane potential estimation

Mitochondrial membrane potential within HSCs was quantified by flow cytometry as follows[27]. Briefly, WBM cells from femurs and tibiae were depleted of lineage committed hematopoietic progenitors and surface stained for 30 minutes at 4⁰C with antibodies raised against SCA1 (D7; Biolegend), cKIT (2B8; Biolegend), CD150 (TC15-12F12.2; Biolegend), and CD48 (HM48-1; Biolegend). Following surface staining, cells were washed with MACS buffer and re-suspended in micro-centrifuge tubes containing 0.5 mL DMEM with TMRE (100 nM, ThermoFisher Scientific T669) and Verapamil (50 μM, Sigma-Aldrich V4629). Cells were incubated in a 37 °C $CO_2$ incubator for 25 minutes with caps open. Following incubation, cells were washed twice with ice-cold MACS buffer. Washed cells were re-suspended in ice-cold MACS buffer and maintained at 4 °C under low-light conditions until flow cytometry acquisition. Cells incubated in DMEM without TMRE served as gating controls. TMRE sensitivity was confirmed by collapse of TMRE intensity to background levels after mitochondrial uncoupling induced by 20 μM FCCP.

## Reactive oxygen species estimation

WBM cells from femurs and tibiae were depleted of lineage committed hematopoietic progenitors and surface stained for 30 minutes at 4 °C with antibodies raised against SCA1 (D7; Biolegend), cKIT (2B8; Biolegend), CD150 (TC15-12F12.2; Biolegend), and CD48 (HM48-1; Biolegend). Following surface staining, cells were washed with MACS buffer and re-suspended in micro-centrifuge tubes containing 0.5 mL DMEM with CM-H2DCFDA (1 μM, ThermoFisher Scientific C6827). Cells were incubated in a 37 °C $CO_2$ incubator for 25 minutes with caps open. Following incubation, cells were washed twice with ice-cold MACS buffer. Washed cells were re-suspended in ice-cold MACS buffer and kept at 4 °C under low-light conditions until flow cytometry acquisition.

## Cell cycle analysis

For HSC cell cycle analysis, WBM cells from femurs and tibiae were depleted of lineage committed hematopoietic progenitors and surface stained for 30 minutes at 4 °C with antibodies raised against SCA1 (D7; Biolegend), cKIT (2B8; Biolegend), CD150 (TC15-12F12.2; Biolegend), and CD48 (HM48-1; Biolegend). Following surface staining, cells were washed with MACS buffer, fixed and permeabilized using the BD Cytofix/Cytoperm Kit (BD Biosciences 554714) as per manufacturer's recommendations. Fixed and permeabilized cells were subsequently stained with an antibody raised against Ki67 (B56, BD 561165) and counterstained with Hoechst 33342 (BD Biosciences 561908). Cells were analyzed using Flow Cytometry with a low acquisition rate (~350 events/second). Cell cycle phases were defined as follows: G0 (Ki-67^negative; 2N DNA), G1 (Ki-67+; 2N DNA), and S/G2/M (Ki-67+; >2N DNA).

For cell cycle analysis of BMECs and LepR+ cells, femurs and tibiae were gently crushed using a mortar and pestle, enzymatically disassociated with Digestion Buffer for 15 minutes at 37 °C, filtered and washed in MACS buffer and surface stained with antibodies raised against CD45 (30-F11; Biolegend), TER119 (TER119; Biolegend), CD31 (390; Biolegend), and LEPR (R&D BAF497) for 30 minutes at 4 °C. Following surface staining, cells were fixed, permeabilized, and stained with an antibody raised against Ki67 and counterstained with Hoechst 33342 as described above for HSCs. BMECs were defined as CD45⁻TER119⁻CD31⁺, while LepR+ cells were defined as CD45⁻TER119⁻LEPR⁺. Cell cycle phases were defined as follows: G0 (Ki-67$^{negative}$; 2N DNA), G1 (Ki-67+; 2N DNA), and S/G2/M (Ki-67+; >2N DNA).

## Vascular permeability (Evans Blue Dye extravasation)

Bone marrow vascular integrity was examined as follows: 0.5% w/v Evans Blue Dye (Sigma-Aldrich E2129) in PBS was injected via tail vein at 25 mg dye/kg total body weight. Three hours post-injection, mice were sacrificed via cervical dislocation and cardiac perfused with 10 mL PBS. Femurs denuded of tissue were crushed in a mortar and pestle with 600 μL formamide (Millipore-Sigma S4117) and incubated at 55 °C overnight. Extractions were briefly vortexed and centrifuged at 16,000 x $g$ for 5 minutes at room temperature. Supernatant was removed and absorbance (Abs) was measured at 620 nm and 740 nm. Sample Abs was corrected for heme-containing proteins [$Abs_{620}$ - $(1.426 \times Abs_{740} + 0.03)$] and blanked using non-injected controls [corrected sample $Abs_{620}$ −corrected non-injected control $Abs_{620}$]. Evan's Blue Dye extravasation was calculated using a standard curve and normalized to femur weight.

## BMEC and BM LepR+ cell isolation

For niche cell RNA-Seq and stromal differentiation analyses, total hematopoietic cells were depleted from dissociated whole bone marrow using the EasyEights EasySep Magnet (StemCell Technologies 18103) with sheep anti-rat IgG Dynabeads (ThermoFisher Scientific 11035) pre-captured with CD45 (30-F11; Biolegend) and TER119 (TER119; Biolegend) antibodies according to the manufacturer's protocol. Briefly, femurs and tibiae were gently crushed with a mortar and pestle and enzymatically dissociated with Digestion buffer for 15 minutes at 37 °C, filtered (40 μm; Corning 352340), washed in MACS buffer, and resuspended in 4 mL MACS buffer. For bead/antibody capture, Dynabeads were prewashed with MACS Buffer and incubated for 30 minutes at 4 °C with either αCD45 or αTER119 antibodies. To remove unbound antibody, Dynabeads were washed four times with MACS Buffer via magnetic separation and resuspended to their initial volume in MACS Buffer. 200 μL CD45-Dynabeads plus 200 μL TER119-Dynabeads were added to the digested WBM cells and incubated for 30 minutes at 4 °C with agitation. To deplete hematopoietic cells, samples were placed on the magnet for 2 minutes and resulting flow through was retained. Dynabeads were washed two additional times with 4 mL MACS Buffer for complete recovery of niche cells; flow through was collected after each wash, combined, and centrifuged at 500 x g for 5 minutes at 4 °C to pellet hematopoietic-depleted bone marrow niche cells.

## LepR+ cell differentiation

To isolate LEPR+ cells for culture, WBM was isolated and depleted of hematopoietic cells as described above. Resulting cells were stained with CD45 (30-F11; Biolegend), TER119 (TER119; Biolegend), CD31 (390; Biolegend), and Biotinylated-LEPR (R&D BAF497, 4 mg/mL) for 30 minutes at 4 °C. Stained cells were washed with MACS Buffer and incubated with Streptavidin-BV421 (Biolegend 405226, 0.2 mg/mL) for 30 minutes at 4 °C. Samples were washed with MACS Buffer and FACS sorted for DAPI⁻CD45⁻TER119⁻CD31⁻LEPR⁺ cells. 15,000 LEPR+ cells were plated in individual wells of a 96 well plate with Adipogenic Differentiation Media (StemCell Tech 05507) and incubated at 37 °C 5% CO₂

5% O₂; media was changed according to the manufacturer's recommendations. Following 8 days of culture, adipogenic differentiation was assessed using BODIPY (ThermoFisher Scientific D3922). BODIPY was resuspended in DMSO according to the manufacturer's recommendations, added to cultured cells at a final concentration of 10 μM, and incubated at 37 °C 5% CO₂ 5% O₂ for 10 minutes. Following adipocyte labeling, total cells were washed with PBS, dissociated using 35 μL Accutase Cell Detachment Solution (Biolegend 423201), and brought to a final volume of 335 μL with MACS Buffer + 1 μg/mL DAPI (Biolegend 422801). BODIPY⁺ cells were quantified on a BD FACS Aria III using a 100 μm nozzle and 20 PSI sheath pressure.

## NTN1 immunofluorescence

B6.129(Cg)-$Lepr^{tm2(cre)Rck}$/J mice were crossed with B6.Cg-$Gt(ROSA)$$26Sor^{tm14(CAG-tdTomato)Hze}$/J (JAX Stock No. 007914) mice to achieve LepR-Cre+/tdTomato+ reporter mice. To label the vasculature, mice were anesthetized and intravenously administered 10 μg anti-mouse CD144 (α-VEcadherin; Clone BV13, Biolegend) antibody. After 10 minutes, mice were euthanized and femurs isolated, stripped of muscle and connective tissue, and fixed in 4% paraformaldehyde overnight at 4 °C. Bones were washed in PBS 3 × 5 minutes and cryo-protected in 30% Sucrose in PBS for 48 hours at 4 °C. Bones were then embedded in a 1:1 mixture of O.C.T. (Tissue-Tek) and 30% Sucrose solution, and snap-frozen under liquid nitrogen. Bones were sectioned on a Leica CM 3050 S cryostat at 12 μm slices, and collected on CFSA 0.5x Slides (Leica) using the Leica CryoJane tape-transfer method. Slides were brought to room temperature and washed in PBS 3 × 5 minutes to remove the O.C.T. Sections were delineated with an ImmEdge Pen (Vector Laboratories), blocked and permeabilized in blocking buffer (PBS containing 10% Normal Donkey Serum with 0.1% Triton X-100) for 2 hours at room temperature. Sections were stained with an antibody raised against NTN1 (R&D Systems, AF1109, 1:100 dilution), and antibodies raised against mature hematopoietic cells (αGR1/CD11B/B220/CD3/CD41/TER119) in blocking buffer and incubated overnight at 4 °C. Sections were washed in PBS 3 × 5 minutes, and stained with donkey anti-goat secondary antibody (Thermo Fisher Scientific) in blocking buffer for 1 hour at room temperature. Slides were washed 3 × 5 minutes in PBS, mounted with Prolong Gold Antifade solution (Thermo Fisher Scientific P36930). Forty μm Z stack images were acquired with a Nikon C2 confocal microscope, denoised with Denoise.ai, and rendered into a maximum intensity projection (MIP) in NIS Elements.

For immunofluorescence analysis of cultured BM MSCs, cells were plated in 8-well chamber slides (Nunc Lab-Tek II CC2, 154941). Cells were washed in PBS and fixed in 4% PFA in PBS for 15 minutes at room temperature. Cells were then permeabilized in blocking buffer for 1 hour at room temperature. Cells were stained with an antibody raised against NTN1 (R&D Systems, AF1109, 1:100 dilution) in blocking buffer for 1 hour at room temperature. Cells were washed 3 times with PBS and stained with donkey anti-goat secondary antibody (ThermoFisher Scientific) in blocking buffer for 30 minutes at room temperature. Cells were washed 3 times with PBS and counterstained with DAPI at 1 μg/mL, and mounted using Prolong Gold Antifade solution (Life Technologies). 40 μm Z stack images were acquired with a Nikon C2 confocal laser scanning microscope, denoised with Denoise.ai, and rendered into a maximum intensity projection (MIP) in NIS Elements.

## Whole-Mount Immunofluorescence

To assess oxygenation and vascular leakiness, mice were injected with 10 μg anti-VEcadherin antibody (BV13, Biolegend), 100 mg/kg Pimonidazole HCl (Hypoxyprobe™−1, Hypoxyprobe Inc.), and 10 μg of 10,000 MW Dextran (ThermoFisher Scientific D-22910) via retro-orbital injections. 10 minutes following injections, mice were euthanized, and femurs were isolated and fixed in 4% paraformaldehyde overnight at 4 °C. Bones were washed in PBS 3 × 5 minutes, cryo-protected in 15% Sucrose in PBS for 24 hours at 4 °C, and further cryo-

protected in 30% Sucrose in PBS for 24 hours at 4 °C. Bones were then embedded in a 1:1 mixture of O.C.T. (Tissue-Tek) and 30% Sucrose solution and snap-frozen under liquid nitrogen. Bones were then shaved longitudinally on a Leica CM 3050S cryostat to expose the bone marrow cavity. Shaved bones were washed in PBS 3 × 5 minutes to remove the O.C.T. Exposed bones were permeabilized in blocking buffer (PBS containing 20% Normal Goat Serum with 0.5% Triton X-100) for 2 hours at room temperature. Bones were stained with anti-bodies raised against Perilipin1 (Sigma P1998, 1:100), and hypoxyprobe (HP-Red549, Hypoxyprobe Inc., 1:100 dilution) in blocking buffer, and incubated for 48 hours at 4 °C. Bones were washed 3 × 10 minutes in PBS and stained with a goat-anti-rabbit secondary antibody (Invitrogen) in blocking buffer overnight at 4 °C. Bones were washed 3 × 10 minutes in PBS, and 40 μm Z stack images were acquired with a Nikon C2 confocal laser scanning microscope, denoised with Denoise.ai, and rendered into a maximum intensity projection (MIP) in NIS Elements. Images were analyzed using Image J software[53]. Briefly, raw images for Hypoxyprobe or Dextran were converted into 8-bit grayscale images, uniformly thresholded, and % of total area for each parameter was exported for analysis. Vessel diameter was measured in Image J by utilizing scale bars to convert pixels into distance measurements. Vessel diameters were then estimated by manually drawing lines across the width of vessels that were intravitally labeled with an antibody targeting VEcadherin (BV13, Biolegend).

## HSC Cytospin

Following lineage cell depletion, HSCs were FACS sorted into microcentrifuge tubes containing MACS buffer. Cytospins were carried out by applying 100–150 μL cell suspensions to slides using pre-wet Shandon Filter Cards, Cytoclip and Sample Chambers assembly (ThermoScientific), according to the manufacturer's suggestions. Cytospins were performed using a Shandon Cytospin 4 centrifuge (ThermoFisher Scientific) with low acceleration at 800 rpm for 3 minutes. Slides were subsequently air dried for 20 minutes, fixed for 10 minutes in 4% paraformaldehyde in PBS, rinsed three times for 5 minutes in PBS, and incubated in blocking buffer (PBS with 10% Normal Goat Serum and 0.2% Triton X-100) for 30 minutes at room temperature. Slides were then incubated in primary antibody raised against phospho-H2AX (Ser139) (JBW301; Millipore) in antibody dilution buffer (PBS with 1% BSA and 0.2% Triton X-100) overnight at 4 °C. Cells were washed three times in PBS and incubated in antibody dilution buffer for 1 hour with goat anti-mouse IgG (Invitrogen A-11029, 1:1000 dilution). Slides were washed three times in PBS, stained with DAPI (Biolegend) at 1 μg/mL in PBS for 5 minutes at room temperature. Slides were washed three times and mounted in ProLong Gold Antifade (ThermoFisher Scientific Scientific). Cells were imaged on a Nikon C2 confocal microscope. Images were analyzed in Image J[54]. A minimum of 500 HSCs were analyzed in aged mice treated with PBS or NTN1.

## PVA HSC expansion

Ex vivo HSC expansion and transplantation assays were performed as follows[25]. Briefly, HSCs (defined as DAPI⁻lineage⁻SCA1⁺ cKIT⁺ CD150⁺ CD48⁻) were FACS sorted into fibronectin-coated 96-well tissue culture plates (Corning, 354409) containing PVA-based expansion media with the following composition: 485 mL F12 Medium (11765054, LifeTech), 5 mL ITSX Supplement 100X (51500056, LifeTech), 5 mL Pen-Strep-Glutamax 100X (10378016, LifeTech), 5 mL 1 M HEPES (25-060-CI, Corning) and 500 mg PVA Powder (P8136, Sigma). After dissolution, PVA-base media was filtered (0.22 μm) and stored at 4 °C. Recombinant murine SCF (PeproTech 250-03) to achieve a final concentration of 10 ng/mL and recombinant murine TPO (PeproTech 315-14) to achieve a final concentration of 100 ng/mL were added to the PVA-base media for HSC expansions. Recombinant murine NTN1 (R&D Systems 1109-N1/CF) was added for HSC expansions with NTN1 (200 ng/mL final concentration), while a corresponding volume of PBS served as vehicle

controls for expansions. Complete media exchanges were performed on Days 6, 8 and 10, and expansions were harvested on Day 11 for hematopoietic analysis and transplantations. Expansions cells were collected by gentle mixing with a P200 pipette, followed by two serial washes of each well with ice-cold PBS to ensure complete recovery of expansion cells. Harvested cells were centrifuged in a swinging-bucket rotor at 300 x *g* for 10 minutes at 4 °C, and supernatants were gently aspirated under low suction pressure with a P10 aspirating tip. Expansion cells were stained with antibody targeting CD45, and equal numbers of DAPI-CD45+ expansion cells were FACS sorted into microcentrifuge tubes for transplantations.

## BM Cytokine analysis

BM was flushed from 4 long bones (femurs/tibiae) using 1.1 mL cold PBS. Flushed samples were centrifuged at 500 *g* for 5 minutes at 4 °C. The supernatants were transferred to new tubes and centrifuged again at 1000 g for 5 minutes at 4 °C. 1 mL supernatant was collected and stored at −80 °C. Samples were shipped on dry ice to Raybiotech for BM cytokine quantitation utilizing their sandwich ELSIA based Quantibody platform. Samples were not diluted prior to quantification.

## RNA isolation

For HSC RNA isolation, lineage cell depletion was performed as described above and stained with antibodies raised against SCA1 (D7; Biolegend), cKIT (2B8; Biolegend), CD150 (TC15-12F12.2; Biolegend), and CD48 (HM48-1; Biolegend) and HSCs (defined as DAPI⁻lineage⁻SCA1⁺cKIT⁺CD150⁺CD48⁻) were FACS sorted directly into TRIzol LS Reagent (ThermoFisher Scientific 10296010). For BMEC and LepR+ cell RNA isolation, mice were intravitally labeled with an antibody raised against VEcadherin (BV13; Biolegend) 10 minutes prior to sacrifice. Femurs and tibiae were gently crushed using a mortar and pestle and enzymatically dissociated with Digestion buffer for 15 minutes at 37 °C, filtered, washed in MACS buffer, and depleted of terminally differentiated hematopoietic cells as described under 'BMEC and BM LepR+ cell isolation'. Depleted cell suspensions were stained with antibodies raised against CD45 (30-F11; Biolegend), TER119 (TER119; Biolegend), CD31 (390; Biolegend), and LEPR (R&D BAF497). Stained cells were washed in MACS buffer. BMECs (defined as CD45⁻TER119⁻Vecad⁺CD31⁺) and LepR+ cells (defined as CD45⁻TER119⁻VEcadherin⁻LEPR⁺) were sorted directly into TRIzol LS Reagent. RNA was purified from TRIzol LS according to manufacturer's recommendations.

## RT-PCR

Total RNA was reverse transcribed using SuperScript™ III First-Strand Synthesis system (ThermoFisher 18080400). cDNA equivalent to RNA content of 500 cells was utilized for RT-PCR analysis. NTN1 mRNA expression within BM niche cells was confirmed by RT-PCR analysis utilizing primers targeting murine Ntn1 (Qiagen Geneglobe ID PPM04909F-200). NTN1 deletion efficiency in BM niche cells was quantified by RT-qPCR analysis utilizing primers specific for the deleted exon (Forward Primer: 5′ −3′ GTTGCAAGCCCTTCCACTAC, Reverse Primer 5′−3′: TCCATGTTGAATCTGCAGCG).

## RNA Sequencing and Mapping

RNA concentration was estimated using the Agilent High Sensitivity RNA Screen Tape System, and libraries were prepared using the SMART-Seq® v4 Ultra® Low Input Kit (Takara Bio) with 5 ng total RNA input per sample, as per manufacturer recommendations. cDNA libraries were subject to high-throughput sequencing (Illumina Nova-Seq) and ~50 million paired-end reads were generated per sample. Reads were checked for quality (FastQC v0.11.5) and processed using the Digital Expression Explorer 2 (DEE2) workflow[55]. Adapter trimming was performed with Skewer (v0.2.2)[56]. Further quality control was performed with Minion, part of the Kraken package[57]. Filtered reads

were mapped to the mouse reference genome GRCm38 using STAR aligner[58] and gene-wise expression counts were generated using the "-quantMode GeneCounts" parameter. The R package edgeR was used to calculate FPKM[59].

## RNA-Seq analysis

Normalized read counts were uploaded and analyzed using the integrated Differential Expression and Pathway (iDEP) (http://bioinformatics.sdstate.edu/idep94/) analysis pipeline[60]. For BMEC and LEPR+ cell RNA-Seq analysis in *LepR-NTN1* mice (*N* = 5 samples per group), normalized read counts were uploaded to iDEP, annotated with mouse ENSEMBL Gene IDs, and genes with low expression values (FPKM < 0.5 in 5 samples) were filtered out with default settings for normalization (Constant c for started log: log(x + c) = 1). 12121 genes (LepR+ cells) and 11397 genes (BMECs) passed the filter. Filtered and processed datasets were downloaded and utilized for performing GSEA. Heatmaps depicting unsupervised hierarchical clustering of the Top 500 differentially expressed genes were generated with default parameters (Distance: Correlation, Linkage: Average, Cutoff Z score: 4, Center Genes: Subtract Mean). The numbers of differentially expressed genes were estimated with the cutoffs for Log2 Fold Change > 1.25 and False Discovery Rate (FDR) < 0.1. For HSC RNA-Seq analysis (*N* = 3 Samples per group, *N* = 3 Groups), genes with low expression values were filtered out with a more stringent cut-off (FPKM < 1 in 3 samples), owing to smaller number of samples per group, with default settings for normalization. Heatmaps depicting unsupervised hierarchical clustering of the Top 100 differentially expressed genes were generated with default parameters. For pairwise comparisons (Young vs Aged PBS and Aged PBS vs Aged NTN1), similar FPKM cutoffs (FPKM < 1 in 3 samples) was utilized to filter out low-abundant genes, and the processed datasets (*N* = 12717 genes for Young vs Aged PBS, *N* = 12256 genes for Aged PBS vs Aged NTN1) that passed filter were downloaded and utilized for GSEA.

## GSEA

For performing GSEA of niche cells and HSCs, GSEA input files were created using filtered and processed datasets downloaded from iDEP. Briefly, genes that were not annotated with ENSEMBL IDs (~50 genes/dataset) were excluded. For the remaining genes, Log 2 FPKM were converted to FPKM, and expression dataset files (.GCT) and phenotype label files (.CLS) were generated in Microsoft Excel. GSEA was performed with the following default parameters. Number of permutations: 1000, Collapse to Gene Symbols, Permutation Type: Gene Set (Less than 7 samples per group), Chip Platform: Human_ENSEMBL_GENE_ID.MSigDB_V7.4.Chip, Gene Set Size: 15-500. GSEA was performed utilizing the HALLMARK (V7.4), REACTOME (V7.4) collections and DNA Repair pathways in the KEGG (V7.4) database.

## HSC aging signature

Genes identified as differentially expressed in the 'Aging List Reanalysis' (Consistency Score ≥ 2) were obtained from Supplemental Table 2 associated with the *Svendsen et al* manuscript[26]. The analysis identified 1131 genes that were consistently altered with age (reported in at least 2 published datasets), of which 749 were upregulated and 382 were down regulated genes. After removing unannotated genes, 717 uniquely upregulated genes and 366 uniquely down regulated genes were identified. The Top 500 upregulated genes (based on Consistency Score and Fold Change), and 366 down regulated genes were utilized to create the 'Svendsen Aged' and 'Svendsen Young' comprehensive HSC signatures, respectively.

## Pre-ranked GSEA

To identify pathways that are consistently altered during HSC aging, Pre-Ranked GSEA analysis was performed on the comprehensive HSC aging signature (Svendsen et al.). The 1131 genes that were consistently altered with age, were ranked by multiplying Log2 Fold Change with their Consistency Score. Gene names were converted Human Gene symbols. Unannotated genes were filtered out, and the remaining ranked list of genes (1079 genes) was analyzed by performing Pre-Ranked GSEA with the following default parameters. Number of permutations: 1000, No Collapse/Remap, Permutation Type: Gene Set, Chip Platform: NA, Gene Set Size: 15-500. GSEA was performed utilizing the HALLMARK (V7.4), REACTOME (V7.4) collections and DNA Repair pathways in the KEGG (V7.4) database.

## Statistics

Sample sizes for phenotypic and functional analysis of mouse hematopoietic parameters were determined based on estimates of variance and effect sizes determined in previous experiments. Number of animals needed were calculated based on the ability to detect a two-fold change in the Mean with 80% power, with the threshold for significance (α) set at 0.05. Both male and female mice were utilized for experiments, and a similar proportion of genders across experimental groups was maintained in all experiments. All experimental findings including transplantations were confirmed in at least 2 independent cohorts of mice, and the data presented represent pooled data from independent experiments. Statistical comparisons between two groups were performed using two-tailed Student's t-test. Multiple comparisons were performed using One-way ANOVA analysis with a Tukey's Correction. Data is presented as the mean ± standard error of the mean (SEM), unless otherwise noted. Statistical significance is indicated as *($p < 0.05$), **($p < 0.01$), ***($p < 0.001$), and n.s. (not significant). Statistical analysis was performed using Prism 6 (GraphPad Software).

## Reporting summary

Further information on research design is available in the Nature Portfolio Reporting Summary linked to this article.

## Data availability

Data supporting the findings of this study are available within the manuscript and its supplementary information files. Source data are provided with this paper. RNA-sequencing datasets generated during this study have been deposited in Gene Expression Omnibus (GEO) under accession number GSE227148. Source data are provided with this paper.

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

## Acknowledgements

We thank Wen-Shan Tsao at the Center for Discovery and Innovation's Flow Cytometry Core. We thank Erika Shor for providing a critical evaluation of the manuscript. We thank Prashant Singh and the Roswell Park Genomics Core for RNA sequencing services. Our work is supported by the American Federation of Aging (PR), National Institutes of Health (1R01CA204308-JMB and 1R01AG065436-JMB), and Leukemia and Lymphoma Society (JMB). PR is supported by the Glenn Foundation for Medical Research Postdoctoral Fellowship in Aging Research. JMB is a Scholar of the Leukemia and Lymphoma Society.

## Author contributions

Conceptualization: P.R., M.C.G., M.G.P., J.M.B.; Methodology: P.R., M.C.G., MGP, T.T., CD, A.W., DR, JMB; Investigation: P.R., M.C.G., MGP, T.T., J.M.B.; Visualization: P.R., M.C.G., M.G.P., A.G.F., J.M.B; Funding acquisition: P.R., J.M.B.; Project administration: J.M.B.; Supervision: P.R., M.C.G., M.G.P., J.M.B.; Writing – original draft: P.R., J.M.B.; Writing – review & editing: P.R., M.C.G., M.G.P., T.T., CD, A.W., A.G.F., S.R., D.R., J.M.B.

## Competing interests

The authors declare no competing interests.
