## [Peer Review File · Nature Communications]

Restoring bone marrow niche function rejuvenates aged hematopoietic stem cells by reactivating the DNA Damage ResponseReviewer #1 (Remarks to the Author):

In this manuscript by Ramalingam et al., the authors describe the role of Netrin 1 in regulating BM Niche cell DDR, maintaining HSC content and function during aging, and how Netrin1 administration can restore HSC function and DDR capacities. The paper is well written. I have some concerns, which if addressed, can strengthen the manuscript:

Figure 1a-c. What is the expression of Netrin1 in BM ECs, overall or sinusoidal versus arteriolar? The merged imaging in 1c suggests co-localization/co-expression by ECs. Other investigators have described netrin 1 expression by arteriolar ECs (Renders et al. Nat Comm 2021). What is the EC marker shown in red? State the marker in the figure.

Figure 2. The vascularity shown in Figure 2a, in both the controls and the LepR-NTN1 mice, appears different than the vascularity shown in Figure 2c, despite the scale apparently being the same, 100 microns. Were different sections of the long bone evaluated between Fig 2a and 2c?

Figure 3. Please include percentages of cells in the key gates, LSK, 150+48-LSK populations, shown in Fig 3c. The decrease in % MPPs is more pronounced than that of the % HSCs. Is the decrease in % HSCs simply a derivative of less ckit+sca1+lin- cells in the NTN1 deficient mice? Meaning, are the percentages of HSCs within the LSK population the same within the 2 groups of mice?

The data in Figure 3i suggest an important decrease in total engraftment capacity of HSCs from the NTN1 deficient donor mice. For Fig 3j, the authors should please show the absolute percentages of the 3 donor lineage populations from each group.

Figure 4, 5. The effects of NTN1 deletion in LepR+ stromal cells on DDR appears to be more pronounced in BM ECs compared to BM lepR+ stromal cells themselves. This is a very interesting finding, suggesting that LepR-derived Ntn1 is essential for BM EC maintenance and function in vivo.

Figure 6. The results of the NTN1 administration to aged mice are interesting. It appears that the effect of NTN1 administration on DDR is of higher magnitude in BM ECs compared to lepR MSCs, although the numbers of replicates are different. It is unclear mechanistically how NTN1 administration systemically is correcting or improving DDR in BM ECs or LepR+ stromal cells, via which receptor and via direct or indirect effects?

Figure 7. The images and quantification shown in this figure suggest a dramatic effect of 2 weeks of NTN1 treatment on adipogenesis in the BM of aged mice. How do the authors propose that 2 weeks of NTN1 mediates such a shift in the BM architecture from the mechanistic standpoint? Is there macrophage activation in response to NTN1 administration?

Figure 8. In panel 8e, it appears that the NTN1 treatment group shows correction of T cell engraftment and myeloid skewing in primary recipient mice that is otherwise observed in mice transplanted with aged control HSCs. However, in the secondary transplanted mice, the corrections of myeloid skewing and T cell reconstitution are not demonstrated between the NTN1 treatment group vs. the control group. Does this mean that NTN1 treatment does not affect/correct the effects of aging on long term HSCs with self renewal capacity?

Supp Figure 4D. The effects of NTN1 administration on hematologic recovery following 5FU myelosuppression are very clear/evident; does the significant decrease in HSC and MPP percentages in the NTN1/5FU treated mice indicate that NTN1 mediates effects on myeloid progenitor cells primarily following injury, or is deleterious to HSC recovery following myelosuppressive injury?

What is the effect of the NTN1 administration on DNA damage measurements in HSCs and MPPs over time after 5FU in the mice shown in Fig 9?

Figure 10. The gene expression data and gamma H2AX and Comet assay data shown here are

consistent with the hypothesis that NTN1 promotes rejuvenation of aged hematopoietic stem/progenitor cells via DNA damage regulation pathways. However, the mechanistic results could be strengthened by additional experiments to show that the functional effects of NTN1 on aged HSCs or aged MPPs are dependent on a particular DDR pathway. Or does augmentation of particular DDR pathways rejuvenate aged HSC or MPP functions in manner redundant to NTN1 administration?

Recent studies by Renders et al. (Nat Comm 2021;12:608) showed that ubiquitous deletion of Ntn1 expression caused HSC loss with associated effects on HSC proliferation/quiescence. The authors should address how their results importantly add to/distinguish from the prior studies by Renders et al. relating to understanding the biology of Ntn1 signaling in hematopoiesis.

In the gain of function Netrin 1 administration studies, it remains unclear whether HSCs are the direct target or whether the beneficial effects on hematopoiesis are observed via indirect action on BM niche cells. Since Ntn1 deletion in LepR+ stromal cells caused substantially increased DNA damage in stromal cells and BM ECs, is it possible that Netrin 1 administration restores niche cell activities as the basis for HSC rejuvenation in aged mice; or a combination of direct and indirect effects?

Reviewer #2 (Remarks to the Author):

In this study, the authors identified netrin-1 (NTN1) as a functional molecule that preserves BM niche homeostasis by maintaining BM vascular integrity and preventing adipocyte accumulation. While periarteriolar smooth muscle cells (SMCs) were described as a putative niche cell that provides NTN1 to HSCs, conditional deletion of NTN1 in SMCs did not recapitulate HSC defects observed upon ubiquitous deletion of NTN1 by CAG-Cre (Renders et al., Nat Commun 2021). Here, the authors showed that LepR+ MSCs is the major source of niche derived NTN1 that not only maintain HSC fitness, but also preserve niche integrity within the BM. Furthermore, the authors demonstrated that niche-derived NTN1 is essential to prevent DNA damage in HSCs, BM endothelial cells (BMECs), and LepR+ MSCs and showed transcriptional downregulation of DDR-related genes. Finally, the authors showed age-related decline in niche derived NTN1 and rejuvenation of aged niche and HSCs by administration of NTN1 into aged mice. Collectively, they proposed that age-related decline in niche derived NTN1 underlies compromised function of aged HSCs.

The important function of niche produced NTN1 in maintaining functional HSC via its receptor neogenin-1 (Neo1) has already been demonstrated by Renders et al. (Nat Commun 2021). Renders et al. also demonstrated that decline of NTN1 production during aging leads to the gradual decrease of Neo1 mediated HSC self-renewal. In this study, the authors clearly showed that LepR+ MSCs is the major source of niche derived NTN1 and that NTN1 maintains niche integrity and functional HSCs. Furthermore, the authors successfully restored the HSC function and the niche integrity in aged mice by administration of recombinant NTN1 into aged mice. These findings are novel and support the role of NTN1 in not only the homeostatic hematopoiesis but also in alterations in hematopoiesis during aging. However, there are several concerns or issues to be clarified as follows.

1. The relationship between NTN1 and its receptor Neo1 in niche cells and HSCs are not clear. Since BMECs are defective upon NTN1 deletion in LepR+ MSCs, BMECs should express Neo1. In contrast, Renders et al. postulated that NTN1 is expressed in arteriole ECs. Please clarify these points by RT-PCR of NTN1 and Neo1, and immunostaining of NTN1 in ECs and MSCs. In addition, which LepR+ MSCs express NTN1, peri-arteriole or sinusoid MSCs, or other MSCs? It looks that not all LepR+ MSCs produce NTN1 in Figure 1C.
2. If NTN1 from LepR+ MSCs is critical for BMECs, deletion of its receptor Neo-1 in BMECs should have impacts on HSCs and niches like the deletion of NTN1 in LepR+ MSCs. Can the authors test this? What happens if NTN1 is deleted in BMECs?
3. Figure 8. The restoration of aged HSC function is impressive. But more careful analysis is required to validate the rejuvenation of HSCs. Long-term repopulation of rejuvenated HSCs should

be confirmed by secondary transplantation. How long the NTN1 effects on HSCs and niche cells last after the cessation of administration? Is it a transient effect? It has been shown that ageing leads to the decline of netrin-1 expression in niches and a compensatory but reversible upregulation of Neo1 on HSCs. This restoration might be a direct effect of NTN1 not only on niche cells, but also on aged HSCs. Can the authors address this question, for example using HSC specific Neo1-null mice? Gulati et al., (Ref 13) also showed Neo1+HoxB5+ LT-HSCs are myeloid-biased and show attenuated repopulating capacity compared to Neo1^{-/-}HoxB5+ LT-HSCs. If NTN1 directly acts on aged HSCs, they are assumed to show similar phenotype rather than rejuvenation. Please discuss this point.

4. The molecular mechanism underlying NTN1-mediated regulation of DDR remains obscure. The authors claim that attenuated expression of DDR related genes by NTN1 deletion or age-related downregulation is one of the mechanisms. But specific DNA repair pathways affected in aged HSCs and niche cells are not demonstrated. Is this a direct effect? This reviewer wonders it may be a consequence of compromised niche integrity. What kind of signals do NTN1/Neo-1 transmit in HSCs and niche cells? Do Neo1-NTN1 signals regulate other cellular functions, such as cell to cell interaction in ECs by regulating tight junctions or integrins?

5. Figure 9, does NTN1 treatment improve the recovery of hematopoiesis after 5-FU myelosuppression also in young mice?

Minor:

1. Figure 2a and suppl Fig 2, please quantitate the vessel dilatation upon NTN1 deletion and young vs. aged BM.

2. Figure 4-6, HALLMARK_E2F_TARGETS and HALLMARK_DNA_REPAIR are negatively enriched in young control MSCs and BMECs compared with young LepR-NTN1-MSCs and BMECs while they are positively enriched in young MSCs and BMECs compared with aged MSCs and BMECs. Are these results consistent with each other?

3. Suppl Fig 3, Does NTN1 treatment alter the lineage composition of the PB in aged mice?

4. Fig 8C, what is the threshold (% chimerism) of engraftment at 16 weeks post-BMT? Do NTN1 injection change the levels of BM cytokines, such as SCF, TPO, and CXCL12 (HSC supportive) and IL-1, IL-6, and TNF α (inflammatory) in aged mice?

5. Figure 8, please check the cell cycle of HSCs after 10 injections?

Reviewer #3 (Remarks to the Author):

In this study, Ramalingam et al. suggested a putative role of NTN1 in regulating BM niche and HSC aging. Deletion of NTN1 from LepR+ BMSCs induced accelerated aging of the BM niche, along with increased DNA damage in BMSCs and HSCs. Importantly, administration of recombinant NTN1 reactivates DDR and resolves the DNA damage within niche cells and HSCs in aged mice. Finally, they showed that NTN1 administration enables the aged hematopoietic system to survive serial chemotherapy regimens.

Whereas the therapeutic effects of recombinant NTN1 seems very interesting, this study has serious flaws, as listed below. Most importantly, NTN1 seems to be expressed in periosteal cells but NOT in LepR+ BMSCs of adult mice, according to several public databases. This strongly challenges the idea that BM niche-derived NTN1 regulates HSC aging. Considering that LepR+ stromal cells are also found in the intestine and skin, it could be that systemic NTN1, but not BM-derived NTN1, plays a more important role in regulating HSC function.

Major comments:

1. NTN1 is basically NOT expressed in LepR+ BMSCs, according to several public database (Baryawno et al., Cell, 2019; Mo et al., EMBO J, 2022). Therefore, it is very surprising to observe the BM niche and HSC phenotypes in LepR-Cre; NTN1-flox mice. In contrast, NTN1 is expressed to some extent in periosteal fibroblasts (Baryawno et al, Cell, 2019; Mo et al, EMBO J, 2022). This can explain why the RNA-seq data in Figure 1a showed expression of NTN1, because in that study, they digested the BM and bone fragments, which contain a large fraction of periosteal cells (Helbling et al., Cell Reports, 2019). The authors did not analyze the deletion efficiency of NTN1 in LepR+ BMSCs by western-blot or qPCR, and the immunostaining data in Figure 1c is not

convincing because of the high background signals that include a lot of hematopoietic cells. The validity of the antibody should be proved by staining the bone sections of Lepr-Cre; NTN1-flox mice.

2. In Figure 2a, the number of arterioles is significantly increased, which is in sharp contrast to the current notion that BM aging is accompanied with diminished arterioles. This strongly argues against premature aging of the BM niche as claimed by the authors. In Figure 2c, magnification of control and cKO pictures is obviously different. In Figure 2d, pictures are not typical adipogenic differentiation results (low density plating precludes efficient differentiation). A larger view should be provided with unbiased quantification of adipocyte numbers.

3. In Figure 3d, the decrease of HSC number is largely due to decreased BM cellularity (Figure 2b), but not decreased HSC frequency. When stringently gated, there are about 70-80 HSCs per 10^6 whole bone marrow cells. Therefore, when transplanting 10^6 CD45.1 WBM competitors with 250 CD45.2 HSCs, as indicated in Figure 3h, the resulting chimerism should be around 75%, not 30% as indicated in Figure 3i. This could be caused by low purity of sorted HSCs, or inefficient irradiation of the recipient CD45.1 mice. In either case, it strongly compromises the reliability of all transplantation data.

Minor points:

1. Little information is provided in the Introduction section. The authors simply reiterate their findings in this study.
2. For detecting DNA damage (Figure 6), gH2AX, 53BP1, RAD51 or P-CHK1 staining should be provided.
3. NES score and FDR should be provided for all GSEA analyses.
4. In Figure S2c and d, no quantification of adipogenesis were provided. The representative pictures showed extremely small views.
5. The authors should detect other aging hallmarks, such as cell polarity (Cdc42/Tubulin) and mitochondrial membrane potential, in aged NTN1-HSCs to compare with young and aged PBS-HSC.

NCOMMS-22-33471: Restoring bone marrow niche function rejuvenates aged hematopoietic stem cells by reactivating the DNA Damage Response.

Ramalingam et. al.,

We would like to thank the Reviewers' for their time reviewing our manuscript and we appreciate the constructive criticisms. Following the Reviewers' guidance, we believe we have adequately addressed the Reviewers' comments and our revised manuscript is much improved as a result. Below is a summary of the major new data we have included:

- New data demonstrating that NTN1 is also expressed in bone marrow endothelial cells in addition to LepR⁺ MSCs, and that endothelial derived NTN1 also regulates BM niche and HSC homeostasis (**New Figures 2, 3g-l, 4d-j, 5d-j, Sup. Fig. 1f-j, Sup. Fig. 2b, g-j, & Sup. Fig. 6f-h**). In all transparency, we were preparing this data for another manuscript since we did not have some of the phenotypic and functional analyses that were performed in our LEPR-NTN1 model. However, based on the Reviewers' suggestions, we felt that bringing this important data to this manuscript would strengthen the message and address some of the major concerns. We have now performed and included all of the phenotypic, transcriptional and functional analysis of HSCs and BM niche cells (competitive HSC transplantations, BM niche analysis, RNA Seq analysis, comet assays etc) in our endothelial specific NTN1 deletion model (CDH5-NTN1). Our new data demonstrates that endothelial specific deletion of NTN1 also results in increased DNA damage within BM niche cells and HSCs that are associated with an impairment of HSC functional potential. These data further consolidate the conclusions of our manuscript by demonstrating that niche derived NTN1 regulates the DNA Damage Response (DDR) within HSCs and the BM niche.
- New data showing that supplementation with NTN1 in an *ex vivo* HSC expansion assay directly rejuvenates aged HSC function assessed by serial transplantation assays (**New Figures 8h-j & Sup. Figure 4d, e**).
- New RNA sequencing data of BM niche cells in aged mice treated with NTN1 that clarifies the direct vs indirect effects of Ntn1 on the BM niche (**New Figure 6h-k**).
- Additional characterization of HSCs in aged mice following 5FU injury and NTN1 treatment, including comet assays (**New Figure 10p, q**) and cell cycle analysis (**New Figure 10p, r**).

Referee #1 (Remarks to the Author):

In this manuscript by Ramalingam et al., the authors describe the role of Netrin 1 in regulating BM Niche cell DDR, maintaining HSC content and function during aging, and how Netrin1 administration can restore HSC function and DDR capacities. The paper is well written. I have some concerns, which if addressed, can strengthen the manuscript:

Figure 1a-c. What is the expression of Netrin1 in BM ECs, overall or sinusoidal versus arteriolar? The merged imaging in 1c suggests co-localization/co-expression by ECs. Other investigators have described netrin 1 expression by arteriolar ECs (Renders et al. Nat Comm 2021). What is the EC marker shown in red? State the marker in the figure.

Response: The EC marker shown in Red is CDH5 (VE Cadherin), and we have included this information in the image as suggested. Analysis of published RNA Seq analysis (Helbling et al PMID: 31801092, Valletta et al PMID: 32796847)^{1,2} performed on BM stromal subpopulations indeed confirms BMEC NTN1 expression including both arteriole and sinusoid BMECs as indicated by the Reviewer (**Reviewer Figure R1a & b**). We have validated these datasets by performing RNA-Seq analysis on BM-derived Arteriole and Sinusoidal ECs, and have confirmed that both EC subsets express Ntn1 (**Figure R1c**). Additionally, we have confirmed the expression of Ntn1 in BM MSCs and total BMECs by performing RT-PCR (**Figure R1d**) and immunostaining (**Figure R1e**). Note that while Renders et al also reported Ntn1 expression within ECs, they did not generate EC-specific Ntn1 knockouts to analyze its role in HSC biology. To this end we are including new data demonstrating that EC-derived Ntn1 also regulates HSC function, in addition to LepR⁺ MSCs (**New Figures 1-6 and New Figures S1 & S2, please see attached figures below following the References**). Our new data demonstrates that similar to MSC-derived Ntn1, EC-derived Ntn1 also regulates DDR within HSCs and BM

niche cells, further reaffirming the idea that Ntn1 regulates DDR within the BM niche. Importantly, EC-specific deletion of Ntn1 does not result in loss of vascular integrity that is observed in LepR-NTN1 mice indicating that MSC derived Ntn1 regulates vascular integrity but not EC-derived Ntn1. While we were planning to submit the EC-Ntn1 story as a separate follow-up publication, in light of the interest in this avenue raised by both Reviewers 1 and 2, we have merged the data from both deletion models in the current manuscript. Note that we are including a comprehensive analysis of all assays performed in the LepR-NTN1 model (HSC transplants, RNA Seq, comet assays etc) for the EC-Ntn1 knockout model as well. We believe these new data provide novel insights into the role of MSC and EC derived Ntn1 in regulating niche function and HSC aging, and further strengthens the conclusions of our manuscript.

Figure 2. The vascularity shown in Figure 2a, in both the controls and the LepR-NTN1 mice, appears different than the vascularity shown in Figure 2c, despite the scale apparently being the same, 100 microns. Were different sections of the long bone evaluated between Fig 2a and 2c?

Response: The Reviewer is correct. A representative diaphyseal field is shown in Fig 2a and a trabecular field is shown in Fig 2c (**New Fig 2i**). A larger uncropped field (**Figure R2**) better illustrates the gross vascular disruption and adiposity in LepR-NTN1 mice. While these IF images are indicative of niche dysfunction, the functional assays (Evans Blue dye extravasation and MSC adipocyte differentiation) provide experimental confirmation of these niche cell defects.

Figure 3. Please include percentages of cells in the key gates, LSK, 150+48-LSK populations, shown in Fig 3c. The decrease in % MPPs is more pronounced than that of the % HSCs. Is the decrease in % HSCs simply a derivative of less ckit+sca1+lin- cells in the NTN1 deficient mice? Meaning, are the percentages of HSCs within the LSK population the same within the 2 groups of mice?

Response: We have included the percentages of these populations as suggested by the reviewer (**Figure R3a**). Note that despite the changes in frequency of HSC/HSPC subsets within the LSK population, there is a

significant decline of overall numbers of HSCs and HSPCs within the BM of LepR-NTN1 mice. Given that HSCs give rise to hematopoietic progenitors, we interpret the changes in relative percentage of HSPCs as well the overall significant decrease in BM cellularity, ckit+sca1+lin- (LKS) cells, MPPs and HSCs as a reflection of compromised HSC functionality in LepR-NTN1 mice. Supporting this idea, Gulati et al (PMID: 31754028)³ and Renders et al (PMID: 33504783)⁴ showed that Ntn1-Neol signaling is relatively specific for HSCs as compared to

hematopoietic progenitors. While we cannot rule out the direct effects of Ntn1 deficiency on MPP function, our cell-cycle analysis reveals a significant increase in HSC quiescence (**Figure R3b**), with no significant cell-cycle changes within MPPs implying a defect at the HSC level. The increased HSC quiescence is indicative of an impaired differentiation of HSCs towards MPPs and explains the significant decrease in MPPs in LepR-NTN1 mice. Our HSC transplantation data (**New Figure 3d**) confirms that HSC functionality is indeed compromised within LepR-NTN1 mice. Additionally, our new data utilizing PVA-based HSC expansion transplantation assays confirm that Ntn1 directly impacts HSC function (**New Figure 8h-j, please see below following the References**).

Figure R3. HSPC alterations in LepR-NTN1 mice.

The data in Figure 3i suggest an important decrease in total engraftment capacity of HSCs from the NTN1 deficient donor mice. For Fig 3j, the authors should please show the absolute percentages of the 3 donor lineage populations from each group.

Response: We have included the absolute percentages of each lineage (% of CD45.2+ Myeloid/B/T cells of the total engrafted cells). In line with the decreased total CD45.2+ engraftment observed in HSCs derived from LepR-NTN1 mice as compared to control mice (**New Figure 3d**), the absolute lineage engraftment levels demonstrate a significantly decreased engraftment of all 3 lineages in HSCs from LepR-NTN1 mice (**Figure R4, New Figure S1e**). We have also included total engraftment for the presented EC-NTN1 conditional knockout mice (**New Figure S1j**). While absolute percentages of lineage engraftment reflect the overall CD45.2+ engraftment levels, we typically utilize relative engraftment of lineage cells (lineage distribution within the engrafted CD45.2+ population) as it is able to provide information regarding lineage-biased differentiation of our test population⁵.

Figure R4. Absolute lineage+ engraftment.

Figure 4, 5. The effects of NTN1 deletion in LepR+ stromal cells on DDR appears to be more pronounced in BM ECs compared to BM lepR+ stromal cells themselves. This is a very interesting finding, suggesting that LepR-derived Ntn1 is essential for BM EC maintenance and function in vivo.

Response: This is indeed an important observation wherein we not only observe a higher DNA damage, but also observe pronounced transcriptional alterations within BMECs of LepR-NTN1 mice as compared to MSCs (**Table S1**). BMECs of LepR-NTN1 mice manifest a downregulation of the ANGIOGENESIS pathway (**Table S3**) that correlates with the vascular leakiness observed in these mice demonstrating that LepR-derived Ntn1 regulates vascular integrity within the BM. We have included these observations in our revised manuscript. Notably, we have previously demonstrated that selective ablation of LepR+ cells also induces vascular disruption confirming that LepR+ cells play a critical role in preserving BM vascular integrity (Ramalingam et al, PMID: 28594660)⁶. These findings illustrate the critical importance of characterizing niche defects in model

systems studying HSC biology to delineate the indirect effects of vascular disruption on HSC activity. Of note, niche analysis was not provided by Renders et al in their global Ntn1 knockouts. We hope the reviewer appreciates the comprehensive transcriptional, phenotypic and functional characterization of niche cells (BMECs, LepR+ MSCs) as well as HSCs provided in our manuscript. Our new data demonstrates that while EC-derived Ntn1 directly regulates HSC activity, the HSC defects observed in LepR-NTN1 mice, and possibly following global knockout of Ntn1, arise as a result of both direct effects of Ntn1 deficiency on HSCs and indirect effects of BM vascular disruption. Collectively, our findings definitively demonstrate for the first time that vascular niche-derived Ntn1 directly regulates HSC function.

Figure 6. The results of the NTN1 administration to aged mice are interesting. It appears that the effect of NTN1 administration on DDR is of higher magnitude in BM ECs compared to lepR MSCs, although the numbers of replicates are different.

Response: The replicates reflect the differences in recovery of these BM niche cells by FACS wherein we consistently recover ~3-4 fold higher numbers of LepR+ MSCs per mouse as compared to BMECs owing to the higher *in vivo* abundance of LepR+ MSCs. We agree with the reviewer that the apparently better DNA damage recovery in aged BMECs as compared to aged MSCs could reflect either a difference in the replicates or due to differential responses of these niche cells to systemically administered Ntn1. To assess this, we performed RNA Seq analysis of BM niche cells in aged mice treated with NTN1 (**Figure R5 and New Figure 6h-k**) which showed that, unlike LepR+ MSCs, BMECs demonstrate a robust upregulation of DDR pathways following NTN1 treatment. These findings suggest that NTN1 directly regulates DDR within BMECs and HSCs whereas the DNA damage improvements in LepR+ MSCs likely results from indirect effects of NTN1 on vascular restoration. It is plausible that higher doses/longer duration of treatment could potentially further improve DNA damage recovery in MSCs. Nonetheless, the observed response to Ntn1 within both aged BMECs and MSCs is clearly sufficient for their functional rejuvenation as indicated by the reversal of BM vascular leakiness, hypoxia and adiposity (**New Figure 7**).

It is unclear mechanistically how NTN1 administration systemically is correcting or improving DDR in BM ECs or LepR+ stromal cells, via which receptor and via direct or indirect effects?

Response: Our transcriptional analysis of aged HSCs and BM niche cells reveals a global downregulation of a number of DDR related pathways (**Tables S5, S7 & S11 REACTOME**, DDR related pathways highlighted in Red text). Note that despite the well-established concept that aging is associated with generalized decline in DDR within most tissues, the underlying mechanisms remain very poorly understood. Our data demonstrates for the first time that aging-associated decline in niche derived Ntn1 likely plays a key role in age-related DDR downregulation within the BM niche and HSCs. This is supported by the restoration of essentially all DDR pathways within aged HSCs following Ntn1 treatment (**New Figure S5b & Table S11**), While we presumed that Ntn1 mediated DNA damage resolution within aged niche cells (BMECs and MSCs) also results from a similar upregulation of these DDR networks as observed in aged HSCs, it could also be an indirect consequence of vascular restoration mediated by Ntn1 as suggested by the Reviewer. This is indeed supported by the increased vascular leakiness and DNA damage observed within niche cells of both LepR-NTN1 mice and aged mice. To formally test whether Ntn1 mediated DNA damage resolution in niche cells of aged mice result from direct effects of Ntn1 on DDR upregulation or indirect effects on improving BM niche

integrity, we have performed transcriptomic analysis on niche cells of aged mice treated with PBS or NTN1 *in vivo* (**Figure R5, New Figure 6h-k**). Our new data demonstrates that Ntn1 treatment does not upregulate DDR pathways within LepR+ MSCs of aged mice (**Figure R5a**), indicating that the DNA damage resolution within MSCs results from indirect effects of Ntn1 on improving vascular function. On the other hand, BMECs of aged mice treated with Ntn1 demonstrate a striking upregulation of the DDR networks that are downregulated during aging (**Figure R5b**), similar to the observation in aged HSCs (**New Figure S5b**). Collectively, our new data demonstrates that Ntn1 directly upregulates DDR responses within both aged HSCs and BMECs that likely reflect their shared developmental ontogeny, whereas DNA damage resolution within MSCs is an indirect consequence of vascular restoration.

Regarding receptors, both BMECs and MSCs express at least 3 Ntn1 receptors: Unc5a, Unc5b and Neo1 (**Figure R6**). While it would be important to understand the relevant receptors in BM niche cells that regulate the response to Ntn1, and to determine whether there are functional redundancies in receptor usage, this would require the generation of conditional niche-cell specific knockouts of all 3 receptors in both BMECs and MSCs. We do not believe generation of these 6 niche specific conditional knockouts and analysis of their HSC/niche function is feasible within this review time frame, given that a comprehensive analysis of these parameters in our LepR-NTN1 mice and CDH5-NTN1 mice took ~5 years to accomplish. However, we believe that the lack of the receptor data does not take away the novel findings of our manuscript wherein we demonstrate for the first time that:

Figure R6. Ntn1 receptor expression in BM niche cells.

1. MSCs and ECs are the critical BM niches for providing Ntn1 to HSCs
2. Ntn1 is a key regulator of BM niche cell function including vascular integrity and adiposity
3. Ntn1 regulates DDR within the BM niche and Ntn1 treatment is sufficient to resolve the aging-associated physical DNA damage within BMECs, MSCs and HSCs
4. Ntn1 treatment restores the functionality of an aged BM niche including improved vascular integrity and suppressing BM adiposity
5. Ntn1 treatment rejuvenates the functionality of aged HSCs (assessed by stringent assays for HSC self-renewal including serial transplantation and serial 5FU challenge) that ensures 100% survival during serial myelosuppressive chemotherapy.

Figure 7. The images and quantification shown in this figure suggest a dramatic effect of 2 weeks of NTN1 treatment on adipogenesis in the BM of aged mice. How do the authors propose that 2 weeks of NTN1 mediates such a shift in the BM architecture from the mechanistic standpoint? Is there macrophage activation in response to NTN1 administration?

Response: While the kinetics were initially surprising to us, it has been reported that unlike human adipocytes, murine adipocytes have a rapid turnover. In young healthy mice, ~4% adipocytes are turned over each day, and the turnover is faster in conditions of excessive adipocyte differentiation including obesity, diabetes and aging (Rigamonti et al, PMID: 21407813, Sakers et al, PMID: 35120662)^{7,8}. We believe that the resolution of BM adiposity following a 2 week treatment with Ntn1 likely reflects the accelerated adipocyte turnover during aging as well as the potency of Ntn1 in suppressing adipocyte differentiation.

Figure 8. In panel 8e, it appears that the NTN1 treatment group shows correction of T cell engraftment and myeloid skewing in primary recipient mice that is otherwise observed in mice transplanted with aged control HSCs. However, in the secondary transplanted mice, the corrections of myeloid skewing and T cell reconstitution are not demonstrated between the NTN1 treatment group vs. the control group. Does this mean that NTN1 treatment does not affect/correct the effects of aging on long term HSCs with self renewal capacity?

Response: A hallmark defect of aging is a loss of HSC self-renewal ability that is reflected as decreased engraftment in HSC transplantation assays. This aspect is critically discussed in a recent publication by Emmanuelle Passegue's group (Ho et al, PMID: 34032859)⁹. To quote from this study, "...*in vivo* restoration of defective repopulation ability should be the key benchmark for bona fide rejuvenation of old HSC function in

transplantation assays". Our engraftment analysis confirms that Ntn1 is able to restore the self-renewal ability of aged HSCs as indicated by their robust engraftment levels in both primary and secondary transplants. With regard to the lineage distribution, while aged HSCs exhibit the typical myeloid bias during primary HSC transplantation (and is completely resolved by Ntn1 treatment), this lineage bias is not always apparent during serial transplantation. This is because during secondary transplantation, aged HSCs are close to functional exhaustion that is reflected by the increased variance in their lineage distribution as compared to young HSCs and Ntn1 treated aged HSCs (**New Figures 8e, g**). Note that many aged PBS HSC recipients show a near complete loss of myeloid or T cell lineages, reflecting a loss of tri-lineage differentiation capacity that is another defining feature of stem cell exhaustion (these caveats regarding lineage bias are discussed in PMID: 34032859)⁹. On the other hand, note that the lineage distribution and its variance in HSCs derived from Ntn1 treated aged mice is very similar to young HSCs. These findings demonstrate that Ntn1 treatment indeed rejuvenates the self-renewal capacity of aged HSCs. The findings are further confirmed by the 100% survival of Ntn1 treated aged mice undergoing serial 5FU treatments which is another stringent assay for evaluating HSC self-renewal (**New Figure 9e, f**). Additionally, our new data demonstrates that *ex vivo* treatment of aged HSCs with Ntn1 is sufficient to restore their self-renewal and balanced lineage reconstitution abilities to levels observed in young HSCs during serial HSC transplantation assays (**New Figure 8h-j, Please see below following the References**). Collectively, these independent assays (limiting dilution transplantations, competitive HSC transplantations, *ex vivo* expansion transplantations and serial 5-FU challenge) demonstrate that Ntn1 rejuvenates the self-renewal ability of aged HSCs. Please note that while several studies have demonstrated partial correction of aged HSC lineage bias in primary transplants (caveats associated with published rejuvenation studies reviewed in Ho et al, PMID: 34032859)⁹, our study is the first to demonstrate true restoration of aged HSC self-renewal and complete reversal of lineage bias.

Supp Figure 4D. The effects of NTN1 administration on hematologic recovery following 5FU myelosuppression are very clear/evident; does the significant decrease in HSC and MPP percentages in the NTN1/5FU treated mice indicate that NTN1 mediates effects on myeloid progenitor cells primarily following injury, or is deleterious to HSC recovery following myelosuppressive injury?

Response: 5FU treatment results in a significant decline of BM cellularity and decreased PB counts in the first week. This is followed by a massive expansion of the HSPC compartment to replenish BM cellularity and blood counts, followed by a return to homeostatic levels. This HSPC expansion is indicated by the ~50 fold higher frequency of HSCs per million WBM cells in both PBS and Ntn1 treated aged mice following 5FU administration (**New Figure S4i**), as compared to their steady state levels (**New Figure S4a**). While there is a ~2 fold decrease in HSPC frequency (HSCs/million WBM cells) in Ntn1 treated aged mice at Day 10 post 5FU, there is a ~3 fold increase in BM cellularity (**New Figure S4h**) demonstrating that absolute numbers of HSPCs per femur are either unchanged or modestly increased in Ntn1 treated aged mice. These findings demonstrate an accelerated recovery of HSPCs and the hematopoietic system in Ntn1 treated aged mice towards homeostatic levels. This is supported by the increased BM cellularity and increased peripheral blood counts observed in Ntn1 treated aged mice at Day 10, with platelets starting to recover as early as Day 7 post 5FU (**New Figure 9a-d**). This accelerated recovery mediated by Ntn1 treatment explains the survival benefit observed during the serial 5FU challenge (**New Figure 9e-g**). We have clarified these observations regarding HSPC frequency in our revised manuscript.

What is the effect of the NTN1 administration on DNA damage measurements in HSCs and MPPs over time after 5FU in the mice shown in Fig 9?

Response: Prior studies have shown that cell cycle activation within aged HSCs following 5FU treatment is able to resolve their DNA damage over time (Beerman et al, PMID: 24813857)¹⁰. To test whether NTN1 treatment can accelerate DNA damage recovery, we performed comet assays following the first injection of NTN1. Our new data demonstrates that NTN1 is able to significantly accelerate DNA damage recovery within HSCs of aged mice treated with 5-FU (**Figure R7, New Figure 10p, q**). However, we could not assess DNA damage within MPPs as we observed a near complete loss of MPPs at this time point.

Figure R7. NTN1 accelerates DNA damage recovery in HSCs following myelosuppressive injury.

Figure 10. The gene expression data and gamma H2AX and Comet assay data shown here are consistent with the hypothesis that NTN1 promotes rejuvenation of aged hematopoietic stem/progenitor cells via DNA damage regulation pathways. However, the mechanistic results could be strengthened by additional experiments to show that the functional effects of NTN1 on aged HSCs or aged MPPs are dependent on a particular DDR pathway. Or does augmentation of particular DDR pathways rejuvenate aged HSC or MPP functions in manner redundant to NTN1 administration?

Response: While we agree with the reviewer that the mechanistic insights could be further strengthened by discovering if specific arms of the DDR are modulated by Ntn1, our transcriptional analysis aged HSCs reveals a global upregulation of not just the major DDR pathways including BER, NER, MMR and HR, (**New Figure S5b**), but over 50 pathways involved in DDR following Ntn1 treatment (**Table S12 REACTOME**, DDR related pathways in Red). Notably, a similar global DDR downregulation is also observed in aged MSCs (**Table S5 REACTOME**) and BMECs (**Table S7 REACTOME**). Importantly, Ntn1 treatment of aged mice restores the expression of the vast majority of these DDR pathways in aged HSCs and BMECs (**New Figure S5b, Table S12, Figure R5b**) arguing against the modulation of a specific DDR pathway by Ntn1. Rather, our data suggests that Ntn1 likely regulates upstream regulators of DDR pathways that are restored in aged HSCs following Ntn1 treatment (**New Figure 10c, d, Figure R5b**). The identification of these upstream regulators in HSCs, BMECs and MSCs represent an important avenue for future investigations, and we are trying to develop and optimize protocols for DDR analysis within these rare cell populations. Note that despite the well-established concept that aging is associated with generalized decline in DDR within most tissues, the underlying mechanisms remain very poorly understood. In this regard, our data demonstrates for the first time that aging-associated decline in niche derived Ntn1 plays a key role in age-related DDR downregulation within the BM. This is supported by the resolution of DNA damage within aged MSCs, BMECs and HSCs following Ntn1 treatment.

While our data does not rule out that Ntn1 might have differential impacts on different arms of the DDR, it is difficult to test this hypothesis experimentally as each pathway involves co-operation of multiple protein complexes. Furthermore, it has been suggested that DNA damage restoration during aging might not be resolved by simple overexpression of individual DDR genes, and indeed genetic overexpression of DDR genes has been shown to have detrimental consequences (Yousefzadeh et al, PMID: 33512317, Birbak et al, PMID: 29452344)^{11,12}. These limitations partly illustrate the reason for lack of scientific literature regarding activities of specific DDR pathways within rare cell populations including HSCs, and we have acknowledged these limitations in our manuscript's discussion. Nonetheless, we plan to investigate these aspects in our future studies utilizing genetic mouse models that are deficient in various arms of the DDR and assess their response to Ntn1 treatment. However, these studies are not feasible to be performed in the timeframe of this review and it is possible that Ntn1 treatment might not rescue the defects in murine models with DDR gene deletions/mutations as the barrier to overcome this would be much higher in genetic knockouts as compared to the physiological DDR downregulation observed during aging.

Recent studies by Renders et al. (Nat Comm 2021;12:608) showed that ubiquitous deletion of Ntn1 expression caused HSC loss with associated effects on HSC proliferation/quiescence. The authors should address how their results importantly add to/distinguish from the prior studies by Renders et al. relating to understanding the biology of Ntn1 signaling in hematopoiesis.

Response: We observe a significant decrease in HSC frequency following Ntn1 deletion in LepR+ MSCs, similar to the observation by Renders et al in their ubiquitous deletion model. However, we see a significant increase in HSC quiescence unlike the increased HSC proliferation reported by Renders et al. The increased quiescence in our LepR-NTN1 model is consistent with the observed decrease in HSC and MPP numbers. On the other hand, Renders et al reported increased HSC proliferation both at 2 months and 5 months following Ntn1 deletion which is difficult to reconcile with their observation of reduced HSC numbers at both time points. The differences in HSC cell cycle status between the two studies likely reflects varying degrees of hematopoietic and niche damage in the model systems utilized, wherein a ubiquitous deletion model of Renders et al would be expected to result in a much severe degree of BM niche damage as compared to our MSC-specific conditional knockouts. While we demonstrate a significant decrease of BM cellularity and BM niche integrity in LepR-NTN1 mice, *these analyses were not performed by Renders et al in their Ntn1 global knockouts precluding any direct comparisons with our study.* Importantly, our new data demonstrates that EC specific deletion of Ntn1 does not result in significant changes in BM cellularity, niche integrity, HSC numbers or cell cycle status. However, EC-specific deletion of Ntn1 results in decline in long-term engraftment potential and a myeloid-biased output that recapitulates features of an aged HSC. *Given that Renders et al did not provide data on lineage reconstitution for the HSC transplants in their ubiquitous Ntn1 knockouts, a direct comparison of this attribute with our findings is not possible.* Collectively, the findings of both our study and Renders et al clearly demonstrate that Ntn1 deletion results in loss of HSC functionality. However, our study confirms that Ntn1 does not regulate HSC cell cycle *per se*, and the observed alterations in the HSC cell-cycle status in LepR-NTN1 knockouts and the ubiquitous knockouts of Renders et al, but not in EC Ntn1 knockouts, likely reflect indirect effects on HSC cell cycle due to the varying degree of BM niche injury in these model systems. These findings are further confirmed by our Ntn1 infusion models wherein we demonstrate that Ntn1 treatment of aged mice rejuvenates the self-renewal potential of aged HSCs without changes in their cell cycle activity.

Another important distinguishing feature is that HSC analysis by Renders et al in NTN1 global knockout mice were performed in HSCs derived from young animals. However, the critical role of NTN1 signaling in maintaining HSC fitness during physiological aging had not been evaluated. This represented a crucial unanswered question in light of the observations that NTN1 overexpression in young mice did not improve HSC functionality reflecting sufficient NTN1 signaling in the young hematopoietic system, and that aging is associated with a deficiency of NTN1⁴. To evaluate the role of NTN1 signaling on the aged hematopoietic system, we physiologically aged our *LepR-NTN1* and *CDH5-NTN1* mouse models for 16 months and assessed their HSC activity. Our results show that niche derived NTN1 plays a critical role in preserving HSC fitness during aging. More importantly, we show that NTN1 treatment of aged mice is sufficient to rejuvenate their HSC functionality. Additionally, we demonstrate for the first time that NTN1 is a key regulator of the BM niche activity wherein deletion of NTN1 in young mice induces premature aging phenotypes within the BM niche including adiposity and vascular leakiness, whereas treatment of aged mice with NTN1 restores their BM niche defects. We believe that our novel findings summarized below clearly distinguish our study from prior reports and represent significant advancements to the fields studying HSC/BM niche aging and rejuvenation.

We demonstrate for the first time that:

1. MSCs and ECs are the critical BM niches for providing Ntn1 to HSCs.
2. Ntn1 is a key regulator of BM niche cell function including vascular integrity and adiposity.
3. Ntn1 regulates DDR within the BM niche and Ntn1 treatment is sufficient to resolve the aging-associated physical DNA damage within BMECs, MSCs and HSCs.
4. Ntn1 treatment restores the functionality of an aged BM niche including improved vascular integrity and suppressing BM adiposity.

5. Ntn1 treatment rejuvenates the functionality of aged HSCs (assessed by stringent assays for HSC self-renewal including serial transplantation and serial 5FU challenge) that ensures 100% survival during serial myelosuppressive chemotherapy.

In the gain of function Netrin 1 administration studies, it remains unclear whether HSCs are the direct target or whether the beneficial effects on hematopoiesis are observed via indirect action on BM niche cells. Since Ntn1 deletion in LepR⁺ stromal cells caused substantially increased DNA damage in stromal cells and BM ECs, is it possible that Netrin 1 administration restores niche cell activities as the basis for HSC rejuvenation in aged mice; or a combination of direct and indirect effects?

Response: We agree with the reviewer's assessment. To assess whether Ntn1 directly affects aged HSC function, we performed PVA based HSC expansion transplantation assays. Our new data (**New Figure 8h-j**) demonstrates that Ntn1 treatment directly restores aged HSC self-renewal following *ex vivo* culture wherein vehicle treated aged HSCs completely lose their engraftment potential while Ntn1 treated aged HSCs demonstrate robust long-term engraftment and multi-lineage reconstitution that are indistinguishable from young HSCs. Collectively, these findings demonstrate that HSC rejuvenation observed in aged mice treated with Ntn1 results from a combination of direct effects of Ntn1 on HSC self-renewal and indirect effects on improving vascular function, hypoxia and DNA damage within the aged BM niche. We have included these data and observations in our revised manuscript that further clarifies the mechanisms underlying Ntn1 mediated HSC rejuvenation.

Referee #2 (Remarks to the Author):

In this study, the authors identified netrin-1 (NTN1) as a functional molecule that preserves BM niche homeostasis by maintaining BM vascular integrity and preventing adipocyte accumulation. While periarteriolar smooth muscle cells (SMCs) were described as a putative niche cell that provides NTN1 to HSCs, conditional deletion of NTN1 in SMCs did not recapitulate HSC defects observed upon ubiquitous deletion of NTN1 by CAG-Cre (Renders et al., Nat Commun 2021). Here, the authors showed that LepR+ MSCs is the major source of niche derived NTN1 that not only maintain HSC fitness, but also preserve niche integrity within the BM. Furthermore, the authors demonstrated that niche-derived NTN1 is essential to prevent DNA damage in HSCs, BM endothelial cells (BMECs), and LepR+ MSCs and showed transcriptional downregulation of DDR-related genes. Finally, the authors showed age-related decline in niche derived NTN1 and rejuvenation of aged niche and HSCs by administration of NTN1 into aged mice. Collectively, they proposed that age-related decline in niche derived NTN1 underlies compromised function of aged HSCs.

The important function of niche produced NTN1 in maintaining functional HSC via its receptor neogenin-1 (Neo1) has already been demonstrated by Renders et al. (Nat Commun 2021). Renders et al. also demonstrated that decline of NTN1 production during aging leads to the gradual decrease of Neo1 mediated HSC self-renewal. In this study, the authors clearly showed that LepR+ MSCs is the major source of niche derived NTN1 and that NTN1 maintains niche integrity and functional HSCs. Furthermore, the authors successfully restored the HSC function and the niche integrity in aged mice by administration of recombinant NTN1 into aged mice. These findings are novel and support the role of NTN1 in not only the homeostatic hematopoiesis but also in alterations in hematopoiesis during aging. However, there are several concerns or issues to be clarified as follows.

The relationship between NTN1 and its receptor Neo1 in niche cells and HSCs are not clear. Since BMECs are defective upon NTN1 deletion in LepR+ MSCs, BMECs should express Neo1. In contrast, Renders et al. postulated that NTN1 is expressed in arteriole ECs. Please clarify these points by RT-PCR of NTN1 and Neo1, and immunostaining of NTN1 in ECs and MSCs. In addition, which LepR+ MSCs express NTN1, peri-arteriole or sinusoid MSCs, or other MSCs? It looks that not all LepR+ MSCs produce NTN1 in Figure 1C.

Response:

We agree with the Reviewer's assessment. Analysis of two published BM niche cell RNA Seq datasets confirms that Ntn1 is indeed expressed in BMECs and a diverse array of BM stromal cells that are marked by the LepR-cre including Pdgfra+ Sca+ (PaS) cells, CAR cells, Osteoblasts, LepR+ Pdgfra+ and LepR+ Pdgfra- cells (Figure R8a, b). Additionally, our RNA Seq analysis confirms the expression of Ntn1 in both arteriole

Figure R8. Ntn1 expression in BM niche cells.

and sinusoid BMECs (**Figure R8c**). We have also confirmed Ntn1 expression in BMECs and MSCs by RT-PCR (**Figure R8d**), and immunostaining (**Figure R8e**). As noted by the reviewer, Ntn1 expression is indeed heterogeneous within LepR⁺ MSCs and BMECs within the BM as indicated by our immunofluorescence analysis (**New Figure 1c**). While identification of the MSC subsets (periarterial vs perisinusoidal etc) and BMEC subsets (arteriole vs sinusoid) that express HSC regulatory Ntn1 represents important future avenues of investigation, we are including new data demonstrating that EC-derived Ntn1 also regulates HSC function (**New Figures 2-6 and New Figures S1&S2, please see below following the References**). Please note that while Renders et al demonstrated Ntn1 expression within BMECs by qPCR, they did not generate EC-specific conditional knockouts of NTN1 to investigate the role of EC-derived NTN1 in regulating HSC activity. Our new data demonstrates that similar to MSC-derived Ntn1, EC-derived Ntn1 also regulates DDR within HSCs and BM niche cells, further confirming the idea that Ntn1 regulates DDR within the BM niche. Importantly, EC-specific deletion of Ntn1 does not result in loss of vascular integrity that is observed in LepR-NTN1 mice indicating that MSC derived Ntn1 regulates vascular integrity but not EC-derived Ntn1. While we were planning to submit the EC-Ntn1 story as a separate follow-up publication, in light of the interest in this avenue raised by both Reviewers 1 and 2, we have merged the findings from both knockout models in the current manuscript. Note that we have included a comprehensive analysis of all assays performed in the LepR-NTN1 model (HSC transplants, RNA Seq, comet assays etc) for the EC-Ntn1 knockout model as well. We believe these new data provide novel insights into the role of MSC and EC derived Ntn1 in regulating niche function and HSC aging, and further strengthens the conclusions of our manuscript.

Regarding receptors, both BMECs and MSCs express at least 3 Ntn1 receptors: Unc5a, Unc5b and Neo1 (**Figure R9**). While it would be important to understand the relevant receptor in BM niche cells that regulate the response to Ntn1, and to determine whether there are functional redundancies in receptor usage, this would require the generation of conditional niche-cell specific knockouts of all 3 receptors in both BMECs and MSCs. We do not believe generation of these 6 niche specific conditional knockouts and analysis of their HSC/niche function is feasible within this review time frame, given that a comprehensive analysis of these parameters in our LepR-NTN1 mice and CDH5-NTN1 mice took ~5 years to accomplish. But these studies are certainly the focus of our future investigations. Nonetheless, the lack of the receptor data does not take away the novel findings of our manuscript wherein we demonstrate for the first time that:

Figure R9. Ntn1 receptor expression in BM niche cells.

1. MSCs and ECs are the critical BM niches for providing Ntn1 to HSCs
2. Ntn1 is a key regulator of BM niche cell function including vascular integrity and adiposity
3. Ntn1 regulates DDR within the BM niche and Ntn1 treatment is sufficient to resolve the aging-associated DNA damage within BMECs, MSCs and HSCs
4. Ntn1 treatment restores the functionality of an aged BM niche including improved vascular integrity and suppressing BM adiposity
5. Ntn1 treatment rejuvenates the functionality of aged HSCs (assessed by stringent assays for HSC self-renewal including serial transplantation and serial 5FU challenge) that ensures 100% survival during serial myelosuppressive chemotherapy.

2. If NTN1 from LepR⁺ MSCs is critical for BMECs, deletion of its receptor Neo-1 in BMECs should have impacts on HSCs and niches like the deletion of NTN1 in LepR⁺ MSCs. Can the authors test this? What happens if NTN1 is deleted in BMECs?

Response: While it is not feasible to test all the Ntn1 receptors in BM niche cells which would require the generation of 6 conditional knockout mouse models as stated above (**Figure R9**), we are including a comprehensive analysis of EC-specific deletion of Ntn1 (**New Figures 2-6 and New Figures S1&S2, please see below following the References**). Our new data demonstrates that similar to MSC-derived Ntn1, EC-derived Ntn1 also regulates DDR within HSCs and BM niche cells, further confirming the idea that Ntn1 regulates DDR within the BM niche. Importantly, the HSC defects in EC Ntn1 knockout mice arise in the

absence of vascular disruption that is observed in LepR-NTN1 mice (and presumably in global Ntn1 knockouts in Renders et al) demonstrating for the first time that EC niche derived Ntn1 directly regulates HSC function.

3. Figure 8. The restoration of aged HSC function is impressive. But more careful analysis is required to validate the rejuvenation of HSCs. Long-term repopulation of rejuvenated HSCs should be confirmed by secondary transplantation. How long the NTN1 effects on HSCs and niche cells last after the cessation of administration? Is it a transient effect?

Response: Please note that the secondary transplantation data was provided in **Figure 8f-g** of our original and revised submissions demonstrating that HSCs derived from aged mice treated with Ntn1 maintain robust engraftment and balanced lineage output during secondary transplantation. These findings coupled with the complete survival observed during serial 5FU challenge (**New Figure 9**), a more stringent assay for HSC self-renewal, confirms that Ntn1 indeed rejuvenates aged HSC function. Additionally, we are including new data demonstrating that *ex vivo* treatment of aged HSCs with Ntn1 is sufficient to restore their long-term serial repopulation ability in serial HSC transplantation assays (**New Figure 8h-j, please see below following the References**). Our new data in **New Figure 8** demonstrates that Ntn1 treatment restores aged HSC self-renewal following *ex vivo* culture wherein vehicle treated aged HSCs completely lose their long term engraftment potential while Ntn1 treated aged HSCs demonstrate robust long-term engraftment and multi-lineage reconstitution that are indistinguishable from young HSCs. While we have not undertaken a time course analysis of how long the effects of Ntn1 last, our serial 5FU challenge experiment indicates that the effects should last at least a few weeks (**New Figure 9**). Note that Ntn1 was administered to aged mice only following the first dose of 5FU and not following subsequent 3 doses of 5FU. However, this initial Ntn1 regimen ensured a 100% survival benefit over the course of 4 rounds of 5FU administration over a period of ~4 months.

It has been shown that ageing leads to the decline of netrin-1 expression in niches and a compensatory but reversible upregulation of Neo1 on HSCs. This restoration might be a direct effect of NTN1 not only on niche cells, but also on aged HSCs. Can the authors address this question, for example using HSC specific Neo1-null mice?

Response: While we do not have access to Neo1-null mice, we have indeed confirmed the direct effects of Ntn1 treatment on aged HSC function by performing *ex vivo* HSC expansion-transplantation assays (**New Figure 8h-j, Please see below following the References**). Our new data demonstrates that *ex vivo* treatment of aged HSCs with Ntn1 is sufficient to restore their long-term repopulation ability in serial HSC transplantation assays, confirming that Ntn1 directly affects HSC function. Collectively, our findings demonstrate that HSC rejuvenation observed in aged mice treated with Ntn1 results from a combination of direct effects of Ntn1 on HSC self-renewal, and indirect effects on improving vascular function, hypoxia and DNA damage within the aged BM niche.

Gulati et al., (Ref 13) also showed Neo1+HoxB5+ LT-HSCs are myeloid-biased and show attenuated repopulating capacity compared to Neo1⁻HoxB5+ LT-HSCs. If NTN1 directly acts on aged HSCs, they are assumed to show similar phenotype rather than rejuvenation. Please discuss this point.

Response: Gulati et al reported that HSCs with surface expression of NEO1 (Hoxb5+NEO1+) represent HSCs with an impaired regenerative capacity and a myeloid-biased output. It is presumed that the NEO+ HSC fraction represents the HSC subset that do not receive adequate NTN1 signals and consequently upregulate NEO1, and exhibit functional defects due to inadequate NTN1-NEO1 signaling. Supporting this, Renders et al showed that aged HSCs demonstrate increased NEO1 expression likely owing to a deficiency of niche derived NTN1. Notably, Renders et al also showed that NEO1 mutant HSCs perform poorly in serial transplantation assays indicating an essential role for NEO1 to receive NTN1 signals for maintaining HSC repopulating activity. Collectively, both studies suggest that NTN1-NEO1 signaling is essential to preserve HSC fitness in young mice wherein HSCs that do not receive adequate NTN1 signals upregulate NEO1 and manifest a decline in their fitness. This phenotype is exacerbated in aging wherein HSC NEO1 expression increases significantly due to deficiency of niche-derived NTN1. Hence, treatment of aged mice with NTN1 *in vivo* (or aged HSCs *ex vivo*) is able to provide adequate NTN1 signals to aged HSCs and restore their functional capacity. We have included these clarifications in our revised manuscript.

4. The molecular mechanism underlying NTN1-mediated regulation of DDR remains obscure. The authors claim that attenuated expression of DDR related genes by NTN1 deletion or age-related downregulation is one of the mechanisms. But specific DNA repair pathways affected in aged HSCs and niche cells are not demonstrated. Is this a direct effect? This reviewer wonders it may be a consequence of compromised niche integrity.

Response: Please note that the specific DDR pathways altered within aged HSCs and BM niche cells are provided in the supplemental tables. Our transcriptional analysis of aged HSCs reveals a global downregulation of not just the major DDR pathways (BER, NER, MMR and HR, (**New Figure S5b**)), but a number of pathways involved in DDR (**Table S11 REACTOME**, DDR related pathways highlighted in Red text). Notably, a similar global DDR downregulation is also observed in aged MSCs (**Table S5 REACTOME**) and BMECs (**Table S7 REACTOME**). Importantly, Ntn1 treatment of aged mice restores the expression of the vast majority of these DDR pathways in aged HSCs (**New Figure S5b, Table S11**) arguing against the modulation of specific DDR pathways by Ntn1. Rather, our data suggests that Ntn1 modulates the key upstream regulators of DDR pathways as indicated by their restoration in aged HSCs following Ntn1 treatment (**New Figure 10c, d**). The identification of these upstream regulators in HSCs, BMECs and MSCs represent an important avenue for future investigations, and we are trying to develop and optimize protocols for DDR analysis within these rare cell populations. Despite the well-established concept that aging is associated with generalized decline in DDR within most tissues, the underlying mechanisms remain very poorly understood. In this context, our data demonstrates for the first time that aging-associated decline in niche derived Ntn1 likely plays a key role in age-related DDR downregulation within the BM niche. This is supported by the resolution of DNA damage within aged MSCs, BMECs and HSCs following Ntn1 treatment. It is becoming increasingly evident that Ntn1 has profound effects on transcriptional networks and biological function as illustrated by its ability to preserve the naïve state of pluripotency in embryonic stem cells (Huyghe et al, PMID: 32231305)¹³. This is supported by our transcriptional analysis wherein Ntn1 treatment not only restores DDR pathways, but essentially the vast majority of transcriptional networks that are altered during HSC aging (**New Figure 10 a-d**).

We agree with the reviewer that it is quite plausible that Ntn1 mediated improvement in DNA damage within the aged BM niche and HSCs could result in part from indirect effects on improving vascular integrity in aged mice. This is indeed supported by the increased vascular leakiness and DNA damage observed within niche cells of LepR-NTN1 mice and PBS-treated aged mice. To formally test whether Ntn1 mediated DNA damage resolution in niche cells of aged mice result from direct effects of Ntn1 on DDR upregulation as observed in aged HSCs or indirect effects on improving BM niche integrity, we have performed transcriptomic analysis on niche cells of aged mice treated with PBS or NTN1 *in vivo* (**Figure R10, New Figure 6h-k**). Our new data demonstrates that Ntn1 does not restore the dampened DDR within LepR+ MSCs of aged mice (**Figure R10a**),

indicating that the DNA damage resolution within MSCs results from indirect effects of Ntn1 on improving vascular function. On the other hand, BMECs of aged mice treated with Ntn1 demonstrate a striking upregulation of the DDR networks that are downregulated during aging (**Figure R10b**), similar to the observation in aged HSCs. Collectively, our data demonstrates that Ntn1 directly upregulates DDR responses within both aged HSCs and BMECs that likely reflect their shared developmental ontogeny, whereas DNA

Figure R10. RNA Seq analysis of BM niche cells in aged mice treated with Ntn1.

damage resolution within MSCs is an indirect consequence of vascular restoration. We have included these observations in our revised manuscript.

What kind of signals do NTN1/Neo-1 transmit in HSCs and niche cells? Do Neo1-NTN1 signals regulate other cellular functions, such as cell to cell interaction in ECs by regulating tight junctions or integrins?

Response: While Ntn1 mediated aged HSC and niche rejuvenation are in part mediated by resolving their DNA damage, our data clearly demonstrates that Ntn1 signaling has multifaceted effects on cellular function as illustrated by the upregulation of Adipogenesis pathway in MSCs (**New Figure 5c**) and downregulation of Angiogenesis pathway in BMECs (**Table S3**) following deletion of Ntn1 in MSCs of *LepR-NTN1* mice. These transcriptional alterations correlate with the increased adipogenesis and vascular leakiness observed within the BM of *LepR-NTN1* mice. Indeed, beyond its first discovered role in axon guidance, the multifunctional role of Ntn1 has also been well established in several recent studies wherein it has been shown to regulate a diverse array of cellular functions including blood-brain barrier integrity (Boye et al, PMID: 35246514)¹⁴ and embryonic stem cell pluripotency (Huyghe et al, PMID: 32231305)¹³. We have included the impact of NTN1 on angiogenesis pathways within BMECs and Adipogenesis pathways within MSCs in our revised manuscript.

Figure 9, does NTN1 treatment improve the recovery of hematopoiesis after 5-FU myelosuppression also in young mice?

Response: We only assessed the effects of Ntn1 treatment on hematopoietic recovery in aged mice as our hypothesis was based on the premise that aging is associated with Ntn1 deficiency, and hence would benefit from Ntn1 supplementation. While we have not assessed this experimentally, it is plausible that Ntn1 treatment might impart modest improvements in hematopoietic recovery in young mice.

Minor:

Figure 2a and suppl Fig 2, please quantitate the vessel dilatation upon NTN1 deletion and young vs. aged BM.

Response: We have included these measurements as suggested by the reviewer (**Figure R11, New Figure 2d, e, & New Figure S3e**). Note that deletion of NTN1 within *LepR+* MSCs results in a significant increase in vessel diameter (**Figure R11a**), while NTN1 deletion within ECs does not result in significant changes (**Figure R11b**). Aging is associated with an increase in vessel diameter that is restored following Ntn1 treatment (**Figure R11c**).

Figure 4-6, HALLMARK_E2F_TARGETS and HALLMARK_DNA_REPAIR are negatively enriched in young control MSCs and BMECs compared with young *LepR-NTN1*-MSCs and BMECs while they are positively enriched in young MSCs and BMECs compared with aged MSCs and BMECs. Are these results consistent with each other?

Response: The reviewer is correct. We believe that the upregulation of DDR responses in Ntn1 knockout niche cells of young mice reflects an acute cellular response in an attempt to correct the DNA damage induced by Ntn1 deletion. While this reflects the capacity of young niche cells to upregulate DDR responses, comet assays indicate that this DDR activation is insufficient to resolve their DNA damage due to the constitutive nature of Ntn1 deletion in these genetic models. On the other hand, age related DDR downregulation and DNA damage within the BM niche likely reflects a consequence of chronic Ntn1 deficiency and vascular dysfunction, and these defects are resolved following Ntn1 supplementation. Please note that our interpretations on DDR are based on the combined analysis of our transcriptional data and functional analysis utilizing comet assays that are able to explain the discordance in DDR directionality in young Ntn1 knockout niche cells and physiologically aged niche cells. The functional data is essential because transcriptional changes in DDR by itself does not reflect the status of physical DNA damage.

Suppl Fig 3, Does NTN1 treatment alter the lineage composition of the PB in aged mice?

Response: We do not see significant changes in PB lineage composition following the NTN1 treatment of aged mice (**Figure R12**). In our experience with steady state analyses of aged BL6 mice that we obtain from Jackson Laboratories and Charles River, BM lineage composition reveals a more consistent myeloid bias as compared to peripheral blood wherein we do not observe a consistent myeloid bias in all cohorts of aged mice (as shown by the relatively normal lineage in the aged PBS mice in **Figure R12**). This observation was also confirmed by Jennifer Trowbridge at Jackson Laboratories. This is unlike the post HSC transplantation setting wherein we consistently observe a peripheral blood myeloid bias from aged HSCs, irrespective of their steady state peripheral blood lineage status. The reasons underlying these remain unclear to us but could possibly reflect the differences in environmental exposures of these mice during their 18 month aged period.

Fig 8C, what is the threshold (% chimerism) of engraftment at 16 weeks post-BMT? Do NTN1 injection change the levels of BM cytokines, such as SCF, TPO, and CXCL12 (HSC supportive) and IL-1, IL-6, and TNF α (inflammatory) in aged mice?

Response: Our threshold for chimerism is >1% engraftment at 16 weeks following BMT. To determine whether NTN1 treatment modulates the expression of HSC regulatory angiocrine and inflammatory factors, we have performed transcriptional analysis of BM niche cells in aged mice treated with 10 injections of Ntn1 (**Figure R13a, b**). Our analysis does not reveal significant changes in the expression of these factors within BMECs and MSCs following NTN1 treatment, except for CXCL12. However, quantification of these cytokines in the BM supernatant did not reveal significant changes in their protein expression (**Figure R13c**). THPO, IL-6 and TNF α were below the limit of detection. We can include these findings as Supplemental Data if deemed necessary.

5. Figure 8, please check the cell cycle of HSCs after 10 injections?

Response: We analyzed cell cycle of HSCs in Figure 10h of the original submission and did not observe any significant differences in cell cycle status. However, we presume that the Reviewer was referring to cell cycle status of HSCs following 5-FU. To this end, we have performed cell cycle analysis of HSCs in 5-FU challenged aged mice treated with PBS/NTN1 which did not reveal significant changes in cell cycle status following Ntn1 treatment (**Figure R14, New Figure 10p, r & New Figure S6a, b**).

Referee #3 (Remarks to the Author):

In this study, Ramalingam et al. suggested a putative role of NTN1 in regulating BM niche and HSC aging. Deletion of NTN1 from LepR+ BMSCs induced accelerated aging of the BM niche, along with increased DNA damage in BMSCs and HSCs. Importantly, administration of recombinant NTN1 reactivates DDR and resolves the DNA damage within niche cells and HSCs in aged mice. Finally, they showed that NTN1 administration enables the aged hematopoietic system to survive serial chemotherapy regimens.

Whereas the therapeutic effects of recombinant NTN1 seems very interesting, this study has serious flaws, as listed below. Most importantly, NTN1 seems to be expressed in periosteal cells but NOT in LepR+ BMSCs of adult mice, according to several public databases. This strongly challenges the idea that BM niche-derived NTN1 regulates HSC aging. Considering that LepR+ stromal cells are also found in the intestine and skin, it could be that systemic NTN1, but not BM-derived NTN1, plays a more important role in regulating HSC function.

Major comments:

1. NTN1 is basically NOT expressed in LepR+ BMSCs, according to several public database (Baryawno et al., Cell, 2019; Mo et al., EMBO J, 2022). Therefore, it is very surprising to observe the BM niche and HSC phenotypes in LepR-Cre; NTN1-flox mice. In contrast, NTN1 is expressed to some extent in periosteal fibroblasts (Baryawno et al, Cell, 2019; Mo et al, EMBO J, 2022). This can explain why the RNA-seq data in Figure 1a showed expression of NTN1, because in that study, they digested the BM and bone fragments, which contain a large fraction of periosteal cells (Helbling et al., Cell Reports, 2019). The authors did not analyze the deletion efficiency of NTN1 in LepR+ BMSCs by western-blot or qPCR, and the immunostaining data in Figure 1c is not convincing because of the high background signals that include a lot of hematopoietic cells. The validity of the antibody should be proved by staining the bone sections of LepR-Cre; NTN1-flox mice.

Response:

We respectfully disagree with the reviewer's comments. The Reviewer's conclusions are derived solely on the basis of single cell RNA Seq (scRNA Seq) studies. It is well known that RNA Seq in general, and scRNA seq in particular, has a poor coverage for low abundant transcripts like Ntn1. Note that Renders et al⁴ utilized RT-qPCR and not RNA-Seq to demonstrate Ntn1 expression within BM niche cells throughout their manuscript, owing to its low expression within the BM niche. The caveats of scRNA Seq are explicitly acknowledged in the reference provided by the reviewer themselves (Mo et al)¹⁵, wherein the authors were unable to recapitulate the known overlap of LepR+ MSCs and Nestin+ MSCs established by genetic lineage tracing studies. This highlights the importance of genetic studies to understand niche biology rather than relying solely on scRNA Seq. The deficiencies associated with scRNA analysis of BM niche cells have been critically discussed in a recent Review which found large variations in both gene expression and niche populations across 6 different BM niche scRNA Seq studies owing to differences in sample preparation and sequencing depth¹⁶. To clarify the comments regarding the Helbling et al dataset, note that MSCs (defined and FACS purified CD45-Ter119-CD31-CD140b+Sca1+) presented in our original **Figure 1a** do express LepR (**Figure R13c**). Additionally, for

an independent validation, we also confirmed Ntn1 expression in LepR⁺ MSCs in another RNA Seq study of BM stromal cell sub-populations published recently in *Nat Comm* (**Figure R13d**, Valletta et al)². Note that both these studies (Helbling et al and Valletta et al) by experts in the field performed bulk RNA Seq on *FACS purified BM stromal cell subpopulations* utilizing well-established cell surface markers in the field of BM niche biology (not bone fragments or periosteal cells as alluded by the reviewer). Nonetheless, we had independently confirmed Ntn1 mRNA expression in FACS purified LepR⁺ MSCs by RT-PCR (**Figure R13a**), and Ntn1 protein expression in BM MSCs by immunofluorescence analysis (**Figure R13b**), prior to generating LepR-NTN1 mice. As for Ntn1 staining observed in hematopoietic cells, it represents the known expression of Ntn1 in hematopoietic subsets including neutrophils, monocytes and macrophages (Ramkhalawon et al, *Nat. Med.*, PMID 24584118, Li et al, *J Ex Med.*, PMID: 33891683, Schlegel et al, *Circ. Res.*, PMID 34289717)¹⁷⁻¹⁹. Additionally, as suggested by the Reviewer, we have now performed qPCR analysis to assess NTN1 deletion efficiency in FACS purified LepR⁺ MSCs that confirms a significant downregulation of Ntn1 expression within MSCs derived from *LepR-NTN1* mice (**New Figure 3f, please see below following the References**).

The other critical aspect that the Reviewer appears to have overlooked is that LepR-cre marks not only LepR⁺ mesenchymal stem cells, but also all of its BM stromal cell derivatives^{20,21} (**Figure R13e**). Indeed, Ntn1 is expressed within a diverse array of BM stromal cells including LepR⁺ cells, Pdgfra⁺ cells, CAR cells and osteoblasts all of which are marked by the LepR-cre (**Figure R13d**), making this cre line a logical first choice to investigate whether MSC derived Ntn1 regulates HSC activity. We have included these new data and clarifications regarding Ntn1 expression within BM MSCs in our revised manuscript.

Another point to note is that the level of gene expression has poor correlation with biological function and this is well known fact to experts working in niche biology. For instance, Sean Morrison's group showed that LepR⁺ MSCs have ~100 fold higher expression of Kit ligand, as compared to endothelial cells (Ding et al, PMID: 22281595)²². However, despite this log-scale difference in mRNA expression, deletion of Kit ligand in either MSCs or endothelial cells resulted in largely similar HSC engraftment defects²². On a similar note, Renders et al deleted Ntn1 in SMA cells as they observed a higher Ntn1 mRNA expression in SMA cells as compared to endothelial cells⁴. However, SMA-specific Ntn1 deletion did not result in HSC engraftment defects¹⁵. Note that while Renders et al also reported Ntn1 expression within BMECs as well, they did not generate EC-specific Ntn1 knockouts to analyze its role in HSC biology. To this end we are including new data demonstrating that EC-derived Ntn1 also regulates HSC function, in addition to LepR⁺ MSCs (**New Figures 2-6 and New Figures S1 & S2, please see below following the References**). Our new data demonstrates that similar to MSC-derived Ntn1, EC-derived Ntn1 also regulates DDR within HSCs and BM niche cells, further reaffirming the idea that Ntn1 regulates DDR within the BM niche. Importantly, EC-specific deletion of Ntn1 does not result in loss of vascular integrity that is observed in LepR-NTN1 mice indicating that MSC derived Ntn1 regulates vascular integrity but not EC-derived Ntn1. While were planning to submit the EC-Ntn1 story as a separate follow-up publication, in light of the interest in this avenue raised by both Reviewers 1 and 2, we have merged the data from both deletion models in the current manuscript. Note that we are including a comprehensive analysis of all assays performed in the LepR-NTN1 model (HSC transplants, RNA Seq, comet assays etc) for the EC-Ntn1 knockout model as well. We believe these new data provide novel insights into the role of MSC and EC derived Ntn1 in regulating niche function and HSC aging, and further strengthens the conclusions of our manuscript. Collectively, our findings emphasize the importance of genetic knockouts to identify the critical niche sources of HSC regulatory factors. Our genetic knockouts reveal that Ntn1 derived from MSCs and ECs are the critical sources of Ntn1 for HSCs within the BM.

Regarding LepR⁺ MSC derived Ntn1 expression within skin and intestine potentially regulating BM HSC function, the idea is a possibility that cannot be experimentally verified as there are no cre lines available that exclusively mark BM niche cells. This caveat not only applies to Ntn1 and LepR-cre but essentially all HSC regulatory molecules identified to date (Kit ligand, Cxcl12, Jag1 etc) utilizing the currently available cre lines (LepR-cre, CDH5-cre, Tie2-cre etc).

2. In Figure 2a, the number of arterioles is significantly increased, which is in sharp contrast to the current notion that BM aging is accompanied with diminished arterioles. This strongly argues against premature aging of the BM niche as claimed by the authors. In Figure 2c, magnification of control and cKO pictures is obviously different. In Figure 2d, pictures are not typical adipogenic differentiation results (low density plating precludes efficient differentiation). A larger view should be provided with unbiased quantification of adipocyte numbers.

Response: Figure 2a is a representative image demonstrating features of gross vascular disruption in LepR-NTN1 mice that is confirmed by the gold standard Evans Blue assay in

Figure 2b. Note that while we have not undertaken quantification of arteriole numbers which is a morphological assessment, our statement regarding premature niche aging in LepR-NTN1 mice is based on functional assessments of hallmark features of BM niche aging (vascular leakiness, MSC adipogenic differentiation, physical DNA damage by comet assays, and overall niche functionality evaluated by HSC transplantation assays). While no premature aging genetic model including the diverse array of DDR mutant mice described till date recapitulates all aspects of physiological aging, we believe the aging-associated functional niche defects observed in LepR-NTN1 mice justifies its characterization as a model of premature niche aging. Nonetheless, we have no issues in describing these findings as ‘aging phenotypes’ and have changed this accordingly in our revised manuscript. As regards Figure 2c, the magnification is the same in both control and LEPR-NTN1. The uncropped images in **Figure R14** demonstrate vascular disruption and adiposity in LepR-NTN1 mice.

As for Figure 2d, this is a representative magnified image demonstrating increased adipocytes in MSCs derived from LepR-NTN1 mice. As described in our methodology, these differentiation assays were performed on FACS purified LepR+ MSCs utilizing plating densities recommended by the manufacturer. Additionally, note that same number of LepR+ MSCs were FACS sorted into 96 well plates from both control and LepR-NTN1 mice to control for the effects of density. FACS sorting equal numbers of LepR+

MSCs directly in Adipocyte differentiation medium without pre-culture in MSC expansion medium allows for unbiased evaluation of LepR+ MSC differentiation potential. These adipocytes have the hallmark lipid droplets observed in adipocytes, and are labeled by BODIPY (**Figure R15a**). Representative uncropped images

Figure R14. Vascular disruption and adiposity in LepR-NTN1 mice. Scale bar: 25 μ M.

Figure R15. Adipogenic differentiation of LepR+ MSCs.

demonstrating larger fields of view are shown in **Figure R15b**. Of note, all quantification and analysis provided in the manuscript are unbiased.

3. In Figure 3d, the decrease of HSC number is largely due to decreased BM cellularity (Figure 2b), but not decreased HSC frequency. When stringently gated, there are about 70-80 HSCs per 10^6 whole bone marrow cells. Therefore, when transplanting 10^6 CD45.1 WBM competitors with 250 CD45.2 HSCs, as indicated in Figure 3h, the resulting chimerism should be around 75%, not 30% as indicated in Figure 3i. This could be caused by low purity of sorted HSCs, or inefficient irradiation of the recipient CD45.1 mice. In either case, it strongly compromises the reliability of all transplantation data.

Response: Given that HSCs give rise to the vast majority of HSPCs and hematopoietic cells within the BM, we believe that the decline in BM cellularity (**New Figure 3a**) results from a decline in HSC frequency (**New Figure 3b**) and functionality (**New Figure 3d**).

As for the comments regarding transplantation data, note that the *average* HSC frequency (defined and stringently gated as Lineage^{Neg} cKIT⁺SCA1⁺ CD150⁺CD48^{Neg}; LKS-SLAM) within the BM of young mice is ~200 HSCs per million WBM cells (not 70-80 as suggested by the reviewer). To cite a few examples illustrating this, please refer to recent publications by the late Paul Frenette's group (Xu et al, *Nat Comm*, PMID: 29934585)²³, Ralf Adams' group (Kusumbe et al, *Nature*, PMID: 27074508)²⁴, Jennifer Trowbridge's group (Young et al, *Cell Stem Cell*, PMID: 33848471)²⁵ and our group (Ramalingam et al, *Nat Comm*, PMID: 32015345)²⁶. More importantly, the *variance* in HSC frequency ranges anywhere from 50 to 500 HSCs per million WBM (illustrated in the publications cited above that utilize dot-plots to demonstrate BM HSC frequency). Based on these numbers, the observed % engraftment in Figure 3i is well within the expected levels. Additionally, the ~6 month old CD45.2 HSC donors (both Control and LepR-NTN1 mice) are Tamoxifen treated that is known to impact HSPC function, unlike the 16-week old unperturbed CD45.1 competitors. In our experience with over 2000 HSC transplants to date, it is the norm, rather than an exception, to observe modest differences in average % engraftment from experiment to experiment despite using the same ratio of HSC donors/competitors for the reasons listed above (e.g., experimental treatments of donor mice, variance in competitor HSC frequency etc).

However, despite these caveats, competitive HSC transplantation is a robust assay for comparing HSC activity between experimental groups because of its well-controlled nature of experimental variables (e.g., utilization of the same source of competitors for all HSC donor cohorts, FACS sorting HSCs of all donor cohorts on the same day using the same gates etc). The stringent HSC gating strategy that we utilize (**shown in Figure 3c**) coupled with the 'Purity' mode during FACS sorting ensures a high level of purity for HSCs. A testament to our HSC sort purity can be observed from our RNA Seq analysis of young versus aged HSCs (**Figure R16, New Figure 10a and Raw Data in Table S10**) wherein the differentially expressed transcripts exhibit a near complete overlap with a meta-analysis of 10 previously published HSC aging RNA Seq datasets²⁷, both for genes downregulated in aged HSCs (Young HSC signature genes, **Figure R16a**) and genes upregulated in aging (Aged HSC signature genes, **Figure R16b**).

As for the reliability of our transplantation data, note the ~4 fold decrease in engraftment potential and myeloid-biased reconstitution in aged HSCs as compared to young HSCs (**Figure 8**) that has been reproduced by multiple groups with expertise in the field of HSC aging and transplant biology.

Moving forward, we kindly ask that the reviewer refrain from casting doubt on others' work with statements such as "low purity of sorted HSCs or inefficient irradiation". We believe these statements are unwarranted based on numerous published manuscripts from our own group and our well respected colleagues.

As for the reliability of our transplantation data, note the ~4 fold decrease in engraftment potential and myeloid-biased reconstitution in aged HSCs as compared to young HSCs (**Figure 8**) that has been reproduced by multiple groups with expertise in the field of HSC aging and transplant biology.

Minor points:

Little information is provided in the Introduction section. The authors simply reiterate their findings in this study.

Response: There is only 1 prior publication on the role of NTN1 signaling in HSC biology (Renders et al), and no prior publications on the role of NTN1 signaling within the BM niche. Renders et al demonstrated that while global deletion of Ntn1 in young mice results in HSC defects, conditional deletion of Ntn1 in smooth muscle cells did not recapitulate these HSC defects. While Renders et al proposed that ECs represent the likely source for HSC regulatory Ntn1 within the BM based on qPCR expression, this was not experimentally addressed by generating mice with conditional deletion of Ntn1 within ECs. Furthermore, the potential roles of Ntn1 in regulating BM niche activity or physiological aging of HSCs and the BM niche was not explored. Here, we demonstrate for the first time that:

1. MSCs and ECs are the critical BM niches for providing Ntn1 to HSCs.
2. Ntn1 is a key regulator of BM niche cell function including vascular integrity and adiposity.
3. Ntn1 regulates DDR within the BM niche and Ntn1 treatment is sufficient to resolve the aging-associated physical DNA damage within BMECs, MSCs and HSCs.
4. Ntn1 treatment restores the functionality of an aged BM niche including improved vascular integrity and suppressing BM adiposity.
5. Ntn1 treatment rejuvenates the functionality of aged HSCs (assessed by stringent assays for HSC self-renewal including serial transplantation and serial 5FU challenge) that ensures 100% survival during serial myelosuppressive chemotherapy.

The principal findings from the previous studies by Renders et al and Gulati et al are included in the Introduction and Discussion sections of our manuscript that provides adequate background.

For detecting DNA damage (Figure 6), gH2AX, 53BP1, RAD51 or P-CBK1 staining should be provided.

Response: While these markers have utility as surrogate markers for DNA damage, their presence or absence *per se* does not confirm or rule out the presence of physical DNA damage. They are also not exclusive for DNA damage as they accumulate under situations of replicative stress. As our objective was to measure the levels of physical DNA damage, we performed the gold standard comet assays in HSCs and BM niche cells. Given the non-informative nature of these additional markers as regards to physical DNA damage, we do not believe that inclusion of these markers will add any insights.

NES score and FDR should be provided for all GSEA analyses.

Response: Note that NES and FDR were already included in our original manuscript within the Table Headers of figures with GSEA analysis (Figures 4b, 5b, 6b, 6d, 10c & 10d, Supplemental Figures 5a, 5c, 5d & 5e). Also note that the NES & FDR values for each pathway were already provided in Tables S2, S3, S5, S7, S9 & S10 in the individual tabs of each excel sheet. Additionally, we have now included NES and p-values within the GSEA plots themselves in all the figures for readers' convenience.

In Figure S2c and d, no quantification of adipogenesis were provided. The representative pictures showed extremely small views.

Response: Note that adipocyte quantification for S2c & d was already provided in Figures 7b & 7c of our original manuscript. As described in the Figure legend, the figures in S2c & d represent thresholded images (Image J) of femoral sections of all mice utilized for quantification of adipocytes, Dextran and Hypoxyprobe. We included these images in the interest of transparency and also to help appreciate the stark differences observed in these parameters between PBS and NTN1 treated aged mice. Note that each square in S2c & d denotes a 100 x 100 x 40 μ m confocal section (90 images in total), and enlarged representative images for these sections was provided in Figure 7a and S2a & b.

The authors should detect other aging hallmarks, such as cell polarity (Cdc42/Tubulin) and mitochondrial membrane potential, in aged NTN1-HSCs to compare with young and aged PBS-HSC.

Response: Note that we had already included HSC aging hallmarks in our original manuscript including mitochondrial membrane potential assessment (**Figure 10i & j**) and cell-cycle status (**Figure 10g & h**) which revealed no significant changes following NTN1 treatment of aged mice. As described in our manuscript, our transcriptional analysis revealed that NTN1 mediated HSC rejuvenation potentially arises from alterations in aging hallmarks (cell cycle/mitochondrial membrane potential/DNA damage). We assessed all of these HSC aging hallmarks in NTN1 treated aged mice which revealed that NTN1 mediated HSC rejuvenation was associated with resolution of DNA damage but without gross changes in cell-cycle and mitochondrial membrane potential.

As regard to cell polarity, while treatment of aged HSCs with Casein (PMID: 22560076)²⁸ or Osteopontin fragments (PMID: 28254837)²⁹ have been shown to completely resolve polarity changes in aged HSCs, it has minimal impact on improving aged HSC functionality (partial improvement in myeloid-bias with no improvement in aged HSC engraftment defects). The caveats of these previous publications are critically discussed in a recent publication by Emmanuelle Passegue's group (PMID: 34032859)⁹. Given that polarity changes in aged HSCs has a poor correlation with HSC functionality/rejuvenation, we do not believe that polarity analysis will provide any significant insights to our findings. Also of note, none of these *phenotypic* hallmarks can predict HSC *functionality*. The only way to assess true rejuvenation of an aged HSC is by performing functional analysis to evaluate the most important hallmarks of HSC aging: *reduced self-renewal* and a *myeloid-biased differentiation*. We believe we have included all the gold standard assays to support our claim of HSC rejuvenation (competitive HSC transplants, limiting dilution transplants, hematopoietic recovery following myelosuppression and serial 5FU challenge). Additionally, we have now included *ex vivo* HSC expansion transplantation assays where our new data demonstrates that *ex vivo* treatment of aged HSCs with Ntn1 is sufficient to restore their self-renewal and balanced lineage reconstitution abilities to levels observed in young HSCs during serial HSC transplantation assays.

References

- 1 Helbling, P. M. *et al.* Global Transcriptomic Profiling of the Bone Marrow Stromal Microenvironment during Postnatal Development, Aging, and Inflammation. *Cell reports* **29**, 3313-3330.e3314, doi:10.1016/j.celrep.2019.11.004 (2019).
- 2 Valletta, S. *et al.* Micro-environmental sensing by bone marrow stroma identifies IL-6 and TGFβ1 as regulators of hematopoietic ageing. *Nat Commun* **11**, 4075, doi:10.1038/s41467-020-17942-7 (2020).
- 3 Gulati, G. S. *et al.* Neogenin-1 distinguishes between myeloid-biased and balanced Hoxb5(+) mouse long-term hematopoietic stem cells. *Proc Natl Acad Sci U S A* **116**, 25115-25125, doi:10.1073/pnas.1911024116 (2019).
- 4 Renders, S. *et al.* Niche derived netrin-1 regulates hematopoietic stem cell dormancy via its receptor neogenin-1. *Nat Commun* **12**, 608, doi:10.1038/s41467-020-20801-0 (2021).
- 5 Rossi, D. J. *et al.* Cell intrinsic alterations underlie hematopoietic stem cell aging. *Proc Natl Acad Sci U S A* **102**, 9194-9199, doi:10.1073/pnas.0503280102 (2005).
- 6 Ramalingam, P., Poulos, M. G. & Butler, J. M. Regulation of the hematopoietic stem cell lifecycle by the endothelial niche. *Current opinion in hematology* **24**, 289-299, doi:10.1097/moh.0000000000000350 (2017).
- 7 Rigamonti, A., Brennand, K., Lau, F. & Cowan, C. A. Rapid cellular turnover in adipose tissue. *PloS one* **6**, e17637, doi:10.1371/journal.pone.0017637 (2011).
- 8 Sakers, A., De Siqueira, M. K., Seale, P. & Villanueva, C. J. Adipose-tissue plasticity in health and disease. *Cell* **185**, 419-446, doi:10.1016/j.cell.2021.12.016 (2022).
- 9 Ho, T. T. *et al.* Aged hematopoietic stem cells are refractory to bloodborne systemic rejuvenation interventions. *J Exp Med* **218**, doi:10.1084/jem.20210223 (2021).
- 10 Beerman, I., Seita, J., Inlay, M. A., Weissman, I. L. & Rossi, D. J. Quiescent hematopoietic stem cells accumulate DNA damage during aging that is repaired upon entry into cell cycle. *Cell stem cell* **15**, 37-50, doi:10.1016/j.stem.2014.04.016 (2014).

- 11 Yousefzadeh, M. *et al.* DNA damage-how and why we age? *Elife* **10**, doi:10.7554/eLife.62852 (2021).
- 12 Birkbak, N. J. *et al.* Overexpression of BLM promotes DNA damage and increased sensitivity to platinum salts in triple-negative breast and serous ovarian cancers. *Ann Oncol* **29**, 903-909, doi:10.1093/annonc/mdy049 (2018).
- 13 Huyghe, A. *et al.* Netrin-1 promotes naive pluripotency through Neo1 and Unc5b co-regulation of Wnt and MAPK signalling. *Nature cell biology* **22**, 389-400, doi:10.1038/s41556-020-0483-2 (2020).
- 14 Boyé, K. *et al.* Endothelial Unc5B controls blood-brain barrier integrity. *Nat Commun* **13**, 1169, doi:10.1038/s41467-022-28785-9 (2022).
- 15 Mo, C. *et al.* Single-cell transcriptomics of LepR-positive skeletal cells reveals heterogeneous stress-dependent stem and progenitor pools. *Embo j* **41**, e108415, doi:10.15252/embj.2021108415 (2022).
- 16 Woods, K. & Guezguez, B. Dynamic Changes of the Bone Marrow Niche: Mesenchymal Stromal Cells and Their Progeny During Aging and Leukemia. *Front Cell Dev Biol* **9**, 714716, doi:10.3389/fcell.2021.714716 (2021).
- 17 Ramkhelawon, B. *et al.* Netrin-1 promotes adipose tissue macrophage retention and insulin resistance in obesity. *Nature medicine* **20**, 377-384, doi:10.1038/nm.3467 (2014).
- 18 Li, J. *et al.* PMN-derived netrin-1 attenuates cardiac ischemia-reperfusion injury via myeloid ADORA2B signaling. *J Exp Med* **218**, doi:10.1084/jem.20210008 (2021).
- 19 Schlegel, M. *et al.* Silencing Myeloid Netrin-1 Induces Inflammation Resolution and Plaque Regression. *Circ Res* **129**, 530-546, doi:10.1161/circresaha.121.319313 (2021).
- 20 Matsuzaki, Y., Mabuchi, Y. & Okano, H. Leptin receptor makes its mark on MSCs. *Cell stem cell* **15**, 112-114, doi:10.1016/j.stem.2014.07.001 (2014).
- 21 Zhou, B. O., Yue, R., Murphy, M. M., Peyer, J. G. & Morrison, S. J. Leptin-receptor-expressing mesenchymal stromal cells represent the main source of bone formed by adult bone marrow. *Cell stem cell* **15**, 154-168, doi:10.1016/j.stem.2014.06.008 (2014).
- 22 Ding, L., Saunders, T. L., Enikolopov, G. & Morrison, S. J. Endothelial and perivascular cells maintain haematopoietic stem cells. *Nature* **481**, 457-462, doi:10.1038/nature10783 (2012).
- 23 Xu, C. *et al.* Stem cell factor is selectively secreted by arterial endothelial cells in bone marrow. *Nat Commun* **9**, 2449, doi:10.1038/s41467-018-04726-3 (2018).
- 24 Kusumbe, A. P. *et al.* Age-dependent modulation of vascular niches for haematopoietic stem cells. *Nature* **532**, 380-384, doi:10.1038/nature17638 (2016).
- 25 Young, K. *et al.* Decline in IGF1 in the bone marrow microenvironment initiates hematopoietic stem cell aging. *Cell stem cell* **28**, 1473-1482.e1477, doi:10.1016/j.stem.2021.03.017 (2021).
- 26 Ramalingam, P. *et al.* Chronic activation of endothelial MAPK disrupts hematopoiesis via NFkB dependent inflammatory stress reversible by SCGF. *Nat Commun* **11**, 666, doi:10.1038/s41467-020-14478-8 (2020).
- 27 Arthur Flohr, S. *et al.* A comprehensive transcriptomesignature of murine hematopoietic stem cell aging. *Blood*, doi:<https://doi.org/10.1182/blood.2020009729> (2021).
- 28 Florian, M. C. *et al.* Cdc42 activity regulates hematopoietic stem cell aging and rejuvenation. *Cell stem cell* **10**, 520-530, doi:10.1016/j.stem.2012.04.007 (2012).
- 29 Guidi, N. *et al.* Osteopontin attenuates aging-associated phenotypes of hematopoietic stem cells. *Embo j* **36**, 840-853, doi:10.15252/embj.201694969 (2017).

Figure 1

Figure 1. NTN1 is expressed by LepR+ MSCs and BMECs within the BM niche. (a, b) RNA-Seq analyses of the BM niche demonstrating NTN1 expression within MSCs and BMECs (Helbling et al, Valletta et al). PaS: PDGFRa+Sca1+, CAR: CXCL12 Abundant Reticular cells. (c) Representative immunofluorescence (IF) images of femoral sections showing NTN1 expression in LepR+ MSCs tightly associated with BMECs (Yellow arrows) and perivascular LepR+ MSCs (Green arrows). Blue: Lineage+ (TER119/CD11B/GR1/B220/CD3) hematopoietic cells. Red: BMECs intravitaly labeled with anti-VE-CADHERIN antibody. White: LepR+ cells marked by LepR-Cre+/tdTomato+. Green: NTN1 expression detected by an anti-NTN1 antibody. (d) RT-PCR analysis demonstrating NTN1 expression within FACS purified LepR+ MSCs (defined as CD45⁻TER119⁻VECAD⁻CD31⁻LepR⁺) within the BM. (e) Representative IF images of cultured BM MSCs (defined as CD45⁻TER119⁻VECAD⁻CD31⁻) showing NTN1 expression. (f) RT-PCR analysis demonstrating NTN1 expression within FACS purified BMECs (defined as CD45⁻TER119⁻LepR⁻VECAD⁺CD31⁺) within the BM. (g) Representative IF images of cultured BMECs (defined as CD45⁻TER119⁻VECAD⁺CD31⁺) showing NTN1 expression.

Figure 2

Figure 2. Loss of NTN1 induces premature aging phenotypes within the BM niche. (a) Representative immunofluorescence (IF) images of femurs intravitaly-labeled with a vascular-specific CD144/VEcadherin antibody (red) demonstrating vascular disruption in *LepR-NTN1* mice. **(b, c)** Representative images of femurs isolated from mice injected with Evans Blue Dye (EBD) and quantification of vascular leakiness by EBD extravasation in **(b)** *CDH5-NTN1* mice, and **(c)** *LepR-NTN1* mice (n=6 mice/group). **(d, e)** Quantification of

vessel diameter within the BM of **(d)** *CDH5-NTN1* mice and **(e)** *LepR-NTN1* mice (n=3 mice/group). **(f, i)** Representative IF images of femurs stained with α -Perilipin1 (PLIN1) antibody demonstrating an increase in PLIN1+ adipocytes in **(f)** *CDH5-NTN1* mice, and **(i)** *LepR-NTN1* mice, as compared to their littermate controls (n=3 mice/group). **(g, h)** Representative IF images **(g)**, and quantification **(h)**, of LepR+ MSC adipogenic differentiation *ex vivo* in *CDH5-NTN1* mice (N=3 mice/group). **(j, k)** Representative IF images **(j)**, and quantification **(k)**, of LepR+ MSC adipogenic differentiation *ex vivo* in *LepR-NTN1* mice (N=3 mice/group). Statistical significance determined using two-tailed unpaired t-test. * P<0.05; ** P<0.01; *** P<0.001. Error bars represent sample mean \pm standard error of the mean (SEM).

Figure 3

Figure 3. BM niche derived NTN1 is essential for maintaining HSC homeostasis. (a-c) BM analysis in young (5-6 month old) *LepR-NTN1* mice demonstrating a decrease in BM cellularity (a), and HSC frequency (b, c), as compared to littermate controls (N=7-8 mice/group). (d, e) Competitive HSC transplantations (250 CD45.2+ donor HSCs with 10⁶ CD45.1 WBM competitor/recipient) demonstrating a loss of long-term (>6 months post-transplant) HSC engraftment potential (d) without lineage alterations (e) in HSCs derived from *LepR-NTN1* mice (N=4-6 donors/group; N=14-18 recipients/group). (f) RT-qPCR analysis of FACS purified LepR+ MSCs demonstrating decreased NTN1 expression in *LepR-NTN1* mice (N=3 mice/group). (g-i) BM analysis of *CDH5-NTN1* mice demonstrating no gross changes in BM cellularity (g), and HSC frequency (h, i), as compared to littermate controls (N=5/group). (j, k) Competitive HSC transplantations (250 CD45.2+ donor HSCs with 10⁶ CD45.1 WBM competitor/recipient) demonstrating a loss of long-term engraftment potential (j), along with a myeloid-biased output (k), in HSCs derived from *CDH5-NTN1* mice (N=4-6 donors/group; N=14 recipients/group). (l) RT-qPCR analysis of FACS purified BMECs demonstrating decreased NTN1 expression in *CDH5-NTN1* mice (N=3 mice/group). Error bars represent sample mean \pm standard error of the mean (SEM). Statistical significance determined using two-tailed unpaired t-test. * P<0.05; ** P<0.01; *** P<0.001. ns denotes statistically not significant.

Figure 4

Figure 4. Loss of NTN1 causes DNA damage within LepR+ MSCs. (a-c) RNA-Seq analysis of LepR+ MSCs derived from young (5-6 month-old) *LepR-NTN1* mice. (a) Heatmap depicting hierarchical clustering of differentially expressed genes. (b) GSEA demonstrating activation of DDR pathways in LepR+ cells of *LepR-NTN1* mice. (c) GSEA Enrichment plots demonstrating activation of ADIPOGENESIS and DNA REPAIR pathways in LepR+ cells of *LepR-NTN1* mice. (d-f) RNA-Seq analysis of LepR+ MSCs derived from *CDH5-NTN1* mice. (d) Heatmap depicting hierarchical clustering of differentially expressed genes. (e) GSEA demonstrating activation of DDR pathways in LepR+ cells of *CDH5-NTN1* mice. (f) GSEA Enrichment plots demonstrating activation of ADIPOGENESIS and DNA REPAIR pathways in LepR+ cells of *CDH5-NTN1* mice. (g) Cell-cycle analysis of LepR+ MSCs derived from *LepR-NTN1* mice and *CDH5-NTN1* mice (N=5 mice/group). (h) Representative IF images of single cell gel electrophoresis (alkaline comet assays) performed on BM LepR+ MSCs. (i, j) Alkaline comet analysis demonstrating an increase in average tail-moment and % Tail DNA in MSCs derived from both *LepR-NTN1* mice (i), and *CDH5-NTN1* mice (j), as compared to their littermate controls (N=3 mice/group). Note that deletion of NTN1 within either MSCs or BMECs results in increased DNA damage within MSCs. Statistical significance determined using two-tailed unpaired t-test. * P<0.05; ** P<0.01; *** P<0.001. ns denotes statistically not significant.

Figure 5

Figure 5. Loss of NTN1 causes DNA damage within BMECs. (a-c) RNA-Seq analysis of BMECs derived from young (5-6 month-old) *LepR-NTN1* mice. (a) Heatmap depicting hierarchical clustering of differentially expressed genes. (b) GSEA demonstrating activation of DDR pathways in BMECs of *LepR-NTN1* mice. (c) GSEA Enrichment plots demonstrating activation of E2F_TARGETS and G2M_CHECKPOINT pathways in BMECs cells of *LepR-NTN1* mice. (d-f) RNA-Seq analysis of BMECs derived from *CDH5-NTN1* mice. (d) Heatmap depicting hierarchical clustering of differentially expressed genes. (e) GSEA demonstrating activation of DDR pathways in BMECs of *CDH5-NTN1* mice. (f) GSEA Enrichment plots demonstrating activation of MYC_TARGETS and MITOTIC_SPINDLE pathways in BMECs of *CDH5-NTN1* mice. (g) Cell-cycle analysis of BMECs derived from *LepR-NTN1* mice and *CDH5-NTN1* mice (N=5 mice/group). (h) Representative IF images of single cell gel electrophoresis (alkaline comet assays) performed on BMECs. (I, j) Alkaline comet analysis demonstrating an increase in average tail-moment and % Tail DNA in BMECs derived from both *LepR-NTN1* mice (i), and *CDH5-NTN1* mice (j), as compared to their littermate controls (N=3 mice/group). Note that deletion of NTN1 within either MSCs or BMECs results in increased DNA damage within BMECs. Statistical significance determined using two-tailed unpaired t-test. * P<0.05; ** P<0.01; *** P<0.001. ns denotes statistically not significant.

Figure 6

Figure 6. NTN1 supplementation restores DNA damage within the aged BM niche. (a, b) RNA-Seq analysis of LepR+ MSCs derived from young (3 month) and aged (18 month) mice. **(a)** Heatmap depicting hierarchical clustering of differentially expressed genes. **(b)** GSEA demonstrating downregulation of DDR pathways in LepR+ cells of aged mice. **(c, d)** RNA-Seq analysis of BMECs derived from young (3 month) and aged (18 month) mice. **(c)** Heatmap depicting hierarchical clustering of differentially expressed genes. **(d)** GSEA demonstrating downregulation of DDR pathways in BMECs of aged mice. **(e)** Experimental design for NTN1 treatment. **(f, g)** Alkaline comet assays demonstrating an increase in average tail-moment and % Tail DNA in both LepR+ MSCs **(f)**, and BMECs **(g)**, of aged mice as compared to young mice (N=3/group). Note that NTN1 treatment results in a significant reduction of DNA damage within aged MSCs and BMECs. **(h-k)** RNA-Seq analysis of LepR+ MSCs and BMECs derived from aged (18 month) mice treated with PBS or NTN1. **(h, i)** GSEA demonstrating no significant upregulation of DDR pathways in LepR+ cells of aged mice treated with NTN1. **(j, k)** GSEA demonstrating an upregulation of DDR pathways in BMECs of aged mice treated with NTN1. Statistical significance determined using One-Way ANOVA with Tukey's correction for multiple comparisons. * P<0.05; ** P<0.01; *** P<0.001. ns denotes statistically not significant.

Figure 8

Figure 8. NTN1 rejuvenates aged HSC functionality. (a) Experimental design for NTN1 treatment in steady state aged mice. (b, c) Limiting dilution WBM transplantation analysis demonstrating ~4 fold increase in HSC numbers in aged mice treated with NTN1 as compared to PBS treated aged mice, evaluated by number of recipient mice demonstrating long-term multi-lineage reconstitution (LTMR) in their peripheral blood after 16 weeks following transplantation (N=3 donors/group; N=20 recipients/group per cell dose). Line graph displaying estimates of HSC frequency in the indicated groups with dashed lines representing 95% confidence intervals. Stem cell frequency and significance were determined using Extreme Limiting Dilution Analysis (ELDA). (d, e) Competitive HSC transplantation demonstrating that NTN1 treatment of aged mice restores long-term (>4 months) HSC engraftment potential (d) and myeloid-biased output (e), similar to HSCs from young donors (N=10-12 donors/group; N=20-50 recipients/group). (f, g) Secondary transplantation

demonstrating a preservation of serial repopulation **(f)** and balanced lineage reconstitution abilities **(g)** in donor cells derived from NTN1 treated aged mice (N=10 donors/group, N=10-20 recipients/group). **(h)** Experimental design describing HSC transplantation strategy to assess the direct effects of NTN1 treatment on aged HSC function (Supp. Fig. 4d). Following a 11 day *ex vivo* expansion, 10,000 FACS sorted expansion cells (DAPI-CD45.2+) from each donor were transplanted along with 10^6 WBM competitor (CD45.1) into preconditioned CD45.1 recipients (N=2 donors/group, N=9-10 recipients/group for each donor). Following long-term (>6 month) engraftment, CD45.2+ HSCs were FACS purified from primary recipients, and transplanted along with fresh CD45.1 WBM competitor into preconditioned CD45.1 secondary recipients (N=12-20 recipients/group). **(i)** Bar graphs demonstrating long-term engraftment potential in *ex vivo* cultured aged HSCs, as compared to HSCs derived from young mice. Note that NTN1 treatment restores LTMR ability of aged HSCs (N=2 donors/group, N=18-20 recipients/group). **(j)** Bar graphs demonstrating serial repopulation and LTMR (>4 months) (N=12-20 recipients/group). Statistical significance determined using two-tailed unpaired t-test (for pairwise comparisons), and One-Way ANOVA with Tukey's correction for multiple comparisons. * P<0.05; ** P<0.01; *** P<0.001. ns denotes statistically not significant.

Figure 10

Figure 10. NTN1 reactivates the dampened DDR within aged HSCs. (a-f) RNA-Seq analysis of HSCs derived from young (3 month), and aged (18 month) mice treated with PBS or NTN1. **(a)** Comparison with an HSC aging meta-analysis (Svendsen et al) demonstrating alignment of young HSCs **(a)**, and aged NTN1 HSCs **(b)**, with the 'young HSC gene signature'. **(c, d)** GSEA showing upregulation of DDR pathways in both young HSCs **(c)** and aged NTN1 HSCs **(d)**, when compared with aged PBS HSCs. **(e, f)** Heatmaps of E2F-Targets **(e)**, and DNA repair genes **(f)**, downregulated in aged HSCs, and restored after NTN1 treatment. **(g)** Representative contour plots (flow cytometry) for HSC cell-cycle analysis. **(h)** Bar graphs showing HSC cell-cycle distribution (N=5 mice/group). **(i, j)** Assessment of mitochondrial membrane potential (MMP) by flow cytometric quantification of Tetramethyl rhodamine ethyl ester (TMRE) in HSCs of aged mice following PBS/NTN1 treatment. Bar graphs demonstrating no significant changes in % of HSCs with high MMP **(i)**, or average TMRE per HSC **(j)**, following NTN1 treatment (N=5 mice/group). **(k)** Assessment of ROS by flow cytometric quantification of dichlorodihydrofluorescein diacetate (DCFDA) in HSCs of aged mice following PBS/NTN1 treatment (N=5 mice/group). **(l, m)** Representative IF images for γ H2AX staining within HSCs **(l)**, showing a decrease in γ H2AX immunoreactivity **(m)**, following NTN1 treatment of aged mice (N=4/group). HSCs from irradiated young mice were utilized as positive controls. **(n, o)** Representative IF images of alkaline comet analysis of HSCs **(n)**, showing decrease in average Tail-moment **(o)**, and a reduction in % Tail DNA within aged HSCs following NTN1 treatment (N=5/group). **(p)** Experimental design for assessment of DNA damage and cell cycle status within HSCs following a single dose of 5-FU. **(q)** Alkaline comet analysis showing significantly reduced DNA damage within HSCs of aged mice treated with NTN1 (N=2/group). **(r)** HSC cell cycle analysis in aged mice following 5-FU treatment (N=5/group). Significance determined using two-tailed unpaired t-test (pairwise comparisons), and One-Way ANOVA (multiple comparisons). * P<0.05; ** P<0.01; *** P<0.001. ns denotes statistically not significant.

Supplemental Figure 1

Supplemental Figure 1. BM niche derived NTN1 regulates HSC activity. (a) Peripheral blood (PB) analysis demonstrating unaltered lineage composition in young (5-6 month old) *LepR-NTN1* mice, as compared to littermate controls (N=6-8 mice/group). (b) BM analysis of *LepR-NTN1* mice demonstrating a decrease in phenotypic BM MPPs (N=7-8 mice/group). (c) Cell-cycle analysis (Ki67/Hoechst) demonstrating an increase in quiescence (%G0) in HSCs derived from *LepR-NTN1* mice (N=5 mice/group). (d) Colony Forming Unit (CFU) assays demonstrating a decline in BM hematopoietic progenitor activity in *LepR-NTN1* mice, as compared to their littermate controls (N=5-8 mice/group). (e) Absolute engraftment levels of long term engrafted CD45.2+ myeloid, B and T cells following HSC transplantation (Figure 3e). (f) Peripheral blood analysis demonstrating unaltered lineage composition in young *CDH5-NTN1* (5-6 month old) mice, as compared to littermate controls (N=10 mice/group). (g) BM analysis of *CDH5-NTN1* mice demonstrating no changes in phenotypic BM MPPs (N=5 mice/group). (h) Cell-cycle analysis (Ki67/Hoechst) demonstrating a modest increase in quiescence (%G0) in HSCs derived from *CDH5-NTN1* mice (N=5 mice/group). (i) CFU assays demonstrating no gross changes in overall progenitor activity in *CDH5-NTN1* mice (N=5 mice/group). (j) Absolute engraftment levels of long term engrafted CD45.2+ myeloid, B and T cells following HSC transplantation (Figure 3k). Statistical significance determined using two-tailed unpaired t-test. * P<0.05; ** P<0.01; *** P<0.001. ns denotes statistically not significant.

Supplemental Figure 2

Supplemental Figure 2. Niche derived NTN1 preserves HSC fitness during aging. (a, b) Representative images demonstrating hair greying in aged *LepR-NTN1* mice (a), but not in aged *CDH5-NTN1* mice (b). (c, d) BM analysis of aged *LepR-NTN1* mice demonstrating a decrease in MPPs, as compared to littermate controls (N=7 mice/group). (e-f) Competitive HSC transplantation assays (1,000 CD45.2+ donor HSCs with 10⁶ CD45.1 WBM competitor/recipient) demonstrating diminished long-term (>6 months) engraftment (e), and a myeloid-biased reconstitution (f), in HSCs derived from aged *LepR-NTN1* mice as compared to littermate controls (N=4-6 donors/group, N=17-18 recipients/group). (g, h) BM analysis of aged *CDH5-NTN1* mice demonstrating an increase in phenotypic HSCs (g), as compared to littermate controls (N=6 mice/group). (i, j) Competitive HSC transplantation assays demonstrating diminished long-term engraftment (i), and altered lineage reconstitution (j), in HSCs derived from aged *CDH5-NTN1* mice as compared to littermate controls (N=4-6 donors/group, N=18-19 recipients/group). Note the increased variance in HSC frequency in both aged *LepR-NTN1* mice and *CDH5-NTN1* mice indicative of a stressed hematopoietic system. Statistical significance determined using two-tailed unpaired t-test. * P<0.05; ** P<0.01; *** P<0.001. ns denotes statistically not significant.

Supplemental Figure 6

Supplemental Figure 6. Loss of niche-derived NTN1 induces DNA damage within HSCs. (a, b) Representative flow cytometry contour plots for assessment of HSC cell cycle status following 5-FU administration and treatment with PBS or NTN1. (c) Representative IF images of alkaline comet assays in HSCs derived from *LepR-NTN1* mice. (d, e) Alkaline comet analysis demonstrating an increase in average Tail-Moment and % Tail DNA in HSCs derived from *LepR-NTN1* mice (N=3 mice/group). (f) Representative IF images of alkaline comet assays in HSCs derived from *CDH5-NTN1* mice. (g, h) Alkaline comet analysis demonstrating an increase in average Tail-Moment and % Tail DNA in HSCs derived from *CDH5-NTN1* mice (N=3 mice/group). Note that deletion of NTN1 within either MSCs or BMECs results in increased DNA damage within HSCs. Statistical significance determined using two-tailed unpaired t-test. * P<0.05; ** P<0.01; *** P<0.001.

Reviewer #1 (Remarks to the Author):

The authors have significantly improved their manuscript through additional analyses and the revisions have been very responsive to all of the concerns raised in the prior review. The revised manuscript adds significantly to the field and provides important new understanding of the role of NTN1 in the BM niche, the interactions of LepR⁺ stromal cells and BM endothelial cells, and their complementary regulation of hematopoiesis in steady state and in response to stress.

Reviewer #2 (Remarks to the Author):

The manuscript has improved very much after extensive revision. The data of EC-specific NTN1-deletion complemented the missing data. Now the paper has a strong impact for the readers. Although this may be out of the scope of this study, I wonder whether the improved function of aged HSCs during ex vivo culture with NTN1 is real rejuvenation or clonal selection of a minor HSC population. The following are very minor points.

1. Please include the BM cytokine data as a supplemental data.
2. Please describe the method for vessel diameter quantification.

Reviewer #3 (Remarks to the Author):

The authors have provided substantial amounts of arguments, instead of new or modified data, to address the concerns raised by this reviewer. Most of the suggestions have been dismissed, which is quite disappointing and precludes publication in NC.

REVIEWERS' COMMENTS

Reviewer #1 (Remarks to the Author):

The authors have significantly improved their manuscript through additional analyses and the revisions have been very responsive to all of the concerns raised in the prior review. The revised manuscript adds significantly to the field and provides important new understanding of the role of NTN1 in the BM niche, the interactions of LepR+ stromal cells and BM endothelial cells, and their complementary regulation of hematopoiesis in steady state and in response to stress.

Reviewer #2 (Remarks to the Author):

The manuscript has improved very much after extensive revision. The data of EC-specific NTN1-deletion complemented the missing data. Now the paper has a strong impact for the readers. Although this may be out of the scope of this study, I wonder whether the improved function of aged HSCs during *ex vivo* culture with NTN1 is real rejuvenation or clonal selection of a minor HSC population. The following are very minor points.

1. Please include the BM cytokine data as a supplemental data.
2. Please describe the method for vessel diameter quantification.

We thank Reviewers 1 & 2 for their time and efforts in reviewing our manuscript, and their constructive suggestions. We believe that the additional data provided have significantly strengthened the overall conclusions of the manuscript. We have included the BM cytokine data in Supplementary Figure 3f. We have included the methodology for vessel diameter quantitation in the Methods section. We agree with the Reviewer that a sub-population of aged HSCs could selectively respond to NTN1 treatment, and this indeed represents an important future direction that is the subject of our ongoing investigations. Elegant experiments performed by Kateri Moore's group (Bernitz JM et al., PMID: 27839867) have shown that only a small fraction of phenotypically defined aged HSCs represent true stem cells with serial repopulating activity. This subset of true stem cells were shown to reside in the H2B-GFP label retaining fraction. However, despite differences in their repopulating activity, both the GFP+ (label-retaining) and the GFP- fractions of aged HSCs exhibit a myeloid-biased differentiation when compared to young HSCs, suggesting that all HSCs in the aged BM are functionally defective when compared to young HSCs. In this regard, the reversal of aged HSC myeloid bias observed in both *ex vivo* HSC expansion-transplantation assays (Figure 10i) as well as following *in vivo* treatment of aged mice with NTN1 (Figure 10d, e) demonstrates that this indeed represents a true rejuvenation of aged HSCs. It has become increasingly evident that the HSC pool within the aged BM is highly heterogeneous, and it remains plausible that a small-subset of yet to be identified 'young-like' HSCs exists within the aged BM that retain the functional potential equivalent to young HSCs. In this context, the rejuvenating effects of NTN1 on aged HSC function could potentially arise from an expansion of these 'young-like' HSCs within the aged BM. Alternatively, NTN1 might selectively rejuvenate specific subsets within the heterogeneous pool of aged HSCs that are not irreversibly damaged, and are therefore amenable for rejuvenation. Both these possibilities are supported by our limiting-dilution transplantation analyses which demonstrates a ~5-fold increase in numbers of *bona fide* long-term repopulating HSCs within the aged BM following NTN1 treatment (Figure 10b, c). However, these possibilities remain to be investigated further to obtain a better understanding of HSC heterogeneity within the aged BM and to identify the HSC subsets that respond to NTN1 treatment. We have included these observations in the Discussion section of our revised manuscript.

Reviewer #3 (Remarks to the Author):

The authors have provided substantial amounts of arguments, instead of new or modified data, to address the concerns raised by this reviewer. Most of the suggestions have been dismissed, which is quite disappointing and precludes publication in NC.

We respectfully disagree with the reviewer's comments. We would like to briefly summarize the extensive data that we had included in our revised manuscript to address this Reviewer's concerns.

1. The Reviewer requested confirmation of NTN1 expression within LepR+ MSCs. To address this, we performed RT-PCR analysis on FACS purified LepR+ MSCs to demonstrate NTN1 expression within LepR+ MSCs *in vivo* (Figure 1d). We

performed immunofluorescence analysis for NTN1 protein expression in cultured BM derived MSCs (Figure 1e). Additionally, for an independent validation, we have also included data demonstrating NTN1 expression within *in vivo* LepR+ MSCs from two RNA Seq datasets published in *Nat Comms* and *Cell Reports* (Figure 1a, b). These new data unequivocally demonstrate that NTN1 is expressed by LepR+ MSCs within the BM.

2. The Reviewer requested quantification of NTN1 deletion efficiency in LepR+ MSCs of *LepR-NTN1* mice. To address this, as suggested by the Reviewer, we performed qPCR analysis for NTN1 mRNA expression on FACS purified LepR+ MSCs of *LepR-NTN1* mice (Figure 3f). Additionally, we have also included NTN1 deletion efficiency in our EC-specific NTN1 deletion model (Figure 3l).

3. The Reviewer requested images with larger fields of view for immunofluorescence analysis of BM femoral sections and MSC adipogenic differentiation assays. These images have been included in our rebuttal (Figures R14 and R15) and demonstrate the vascular disruption and increased adipogenic differentiation observed in *LepR-NTN1* mice.

4. The Reviewer requested data to demonstrate the purity of our FACS sorting and the robustness of our HSC transplant data. To demonstrate the purity of our FACS sorting, we have included data demonstrating that the differentially expressed genes in our RNA Seq analysis of young versus aged HSCs overlaps with a meta-analysis of 10 previously published HSC aging transcriptomic datasets (Figure 10a, Figure R16). Our HSC transplant data demonstrates a significant decrease in engraftment potential (Figure 8d), and a myeloid-biased reconstitution (Figure 8e), in aged HSCs as compared to young HSCs. These data faithfully recapitulates the HSC defects previously reported by multiple groups with expertise in the field of HSC aging and transplant biology. Collectively, these data demonstrate the purity of our FACS sorting and robustness of our transplant data.

5. The Reviewer requested data for quantification of adipocytes in the BM following NTN1 treatment of aged mice. We have included this data in our manuscript (Figures 7b, c).

6. The Reviewer requested data for assessment of HSC aging hallmarks in aged mice following NTN1 treatment. We have included an extensive analysis of the hallmark phenotypic and functional defects associated with HSC aging following NTN1 treatment including mitochondrial membrane potential (Figures 10i, j), cell cycle analysis (Figures 10g, h), gamma-H2AX immunoreactivity (Figures 10l, m), comet assays for DNA damage (Figures 10n, o), RNA Seq analysis (Figure 8d), HSC transplantation analyses (limiting dilution transplantations (Figure 8b, c), competitive HSC transplantations (Figure 8d-g), *ex vivo* HSC expansion transplantations (Figures 8i, j)), and myelosuppressive recovery studies (Figures 9a-g).

7. The Reviewer requested NES scores/p values for GSEA analyses to be included within the Figures, and suggested the Introduction to be expanded. We have included these data and expanded the Introduction as suggested.

In summary, we believe we have adequately addressed this Reviewer's concerns by including a substantial amount of new data that further strengthens the conclusions of our manuscript.